# Destabilizers of the thymidylate synthase homodimer accelerate its proteasomal degradation and inhibit cancer growth

Luca Costantino[1], Stefania Ferrari[1], Matteo Santucci[1], Outi MH Salo-Ahen[2†], Emanuele Carosati[3‡], Silvia Franchini[1], Angela Lauriola[4§], Cecilia Pozzi[5], Matteo Trande[1§], Gaia Gozzi[1], Puneet Saxena[1#], Giuseppe Cannazza[1], Lorena Losi[1], Daniela Cardinale[6], Alberto Venturelli[1], Antonio Quotadamo[1], Pasquale Linciano[1], Lorenzo Tagliazucchi[1], Maria Gaetana Moschella[1,7], Remo Guerrini[8], Salvatore Pacifico[8], Rosaria Luciani[1], Filippo Genovese[1], Stefan Henrich[2], Silvia Alboni[1], Nuno Santarem[9], Anabela da Silva Cordeiro[9,10], Elisa Giovannetti[11,12], Godefridus J Peters[11¶], Paolo Pinton[13], Alessandro Rimessi[13], Gabriele Cruciani[3], Robert M Stroud[14], Rebecca C Wade[2,15,16], Stefano Mangani[5], Gaetano Marverti[4], Domenico D'Arca[4*], Glauco Ponterini[1*], Maria Paola Costi[1*]

[1]Department of Life Sciences, University of Modena and Reggio Emilia, Modena, Italy; [2]Molecular and Cellular Modeling Group, Heidelberg Institute for Theoretical Studies, Heidelberg, Germany; [3]Department of Chemistry, Biology and Biotechnology, University of Perugia, Perugia, Italy; [4]Department of Biomedical, Metabolic and Neural Sciences, University of Modena and Reggio Emilia, Modena, Italy; [5]Department of Biotechnology, Chemistry and Pharmacy, University of Siena, Siena, Italy; [6]Respiratory, Critical Care & Anesthesia UCL Great Ormond Street Institute of Child Health, London, United Kingdom; [7]Clinical and Experimental Medicine PhD Program, University of Modena and Reggio Emilia, Modena, Italy, Modena, Italy; [8]Department of Chemical and Pharmaceutical Science, University of Ferrara, Ferrara, Italy; [9]IBMC I3S, Porto, Portugal; [10]Department of Biological Sciences, Faculty of Pharmacy, University of Porto, Porto, Portugal; [11]Department of Medical Oncology, Amsterdam University Medical Center, Cancer Center Amsterdam, 1081HV, Vrije Universiteit Amsterdam, Amsterdam, Netherlands; [12]CancerPharmacology Lab, Fondazione Pisana per la Scienza, Pisa, Italy; [13]Dept. of Medical Sciences and Laboratory for Technologies of Advanced Therapies (LTTA), University of Ferrara, Ferrara, Italy; [14]Biochemistry and Biophysics Department, University of California San Francisco, San Francisco, United States; [15]Interdisciplinary Center for Scientific Computing (IWR), Heidelberg University, Heidelberg, Germany; [16]Center for Molecular Biology (ZMBH), DKFZ-ZMBH Alliance, Heidelberg University, Heidelberg, Germany

*For correspondence:
domenico.darca@unimore.it (DD'A);
glauco.ponterini@unimore.it (GP);
mariapaola.costi@unimore.it (MPC)

Present address: [†]Pharmaceutical Sciences Laboratory/Structural Bioinformatics Laboratory, Faculty of Science and Engineering, Pharmacy/ Biochemistry, ÅboAkademi University, Tykistökatu, Finland; [‡]Department of Chemical and Pharmaceutical Sciences, Trieste, Italy; [§]Department of Biotechnology, University of Verona, Verona, Italy; [#]Excelra Knowledge Solutions, Nacharam, India; [¶]Department of Biochemistry, Medical University of Gdansk, Gdansk, Poland

**Abstract** Drugs that target human thymidylate synthase (hTS), a dimeric enzyme, are widely used in anticancer therapy. However, treatment with classical substrate-site-directed TS inhibitors induces over-expression of this protein and development of drug resistance. We thus pursued an alternative strategy that led us to the discovery of TS-dimer destabilizers. These compounds bind at the monomer-monomer interface and shift the dimerization equilibrium of both the recombinant and the intracellular protein toward the inactive monomers. A structural, spectroscopic, and kinetic investigation has provided evidence and quantitative information on the effects of the interaction

of these small molecules with hTS. Focusing on the best among them, **E7**, we have shown that it inhibits hTS in cancer cells and accelerates its proteasomal degradation, thus causing a decrease in the enzyme intracellular level. **E7** also showed a superior anticancer profile to fluorouracil in a mouse model of human pancreatic and ovarian cancer. Thus, over sixty years after the discovery of the first TS prodrug inhibitor, fluorouracil, **E7** breaks the link between TS inhibition and enhanced expression in response, providing a strategy to fight drug-resistant cancers.

## Editor's evaluation

Drugs that therapeutically target human thymidylate synthase are widely used to treat cancer. In this manuscript, the authors have discovered and validated an alternative strategy to target this enzyme using dimer destabilizers. By disrupting homodimerization of these proteins using a series of biophysical assays, this research team identified an analog E7 that inhibits hTS in cancer cells and demonstrates a superior profile to approved drugs targeting this pathway. The work is innovative and novel and could lead to the development of new classes of drugs targeting this key enzyme for cancer therapy.

## Introduction

Small molecules able to bind at specific pockets of the monomer/monomer interface of a dimeric protein may perturb the dimeric assembly to the limit of disrupting it, and thus alter metabolic pathways associated with the cellular functions of the monomeric and dimeric protein forms (*Taddia et al., 2015*). Such a drastic structural change of the protein may open the way to unexpected events, such as a higher liability to intracellular degradation of the protein monomers relative to the dimers.

Human thymidylate synthase (hTS) is an obligate, stable homodimer ($K_d$ = 80 nM *Genovese et al., 2010*) with two active sites, each including residues from both monomers (*Costi et al., 2005*). As a dimeric enzyme, it provides the sole de novo pathway to deoxythymidylate (dTMP) synthesis in human cells by catalyzing the reductive methylation of deoxyuridylate (dUMP) to dTMP using methylenetetrahydrofolate (mTHF) as the one-carbon methyl donor (*Costi et al., 2005*; *Carreras and Santi, 1995*). By interacting with its own and other mRNAs, this protein regulates its own levels and those of other proteins involved in apoptotic processes, including bcl2, c-myc, and p53 (*Brunn et al., 2014*; *Jennings and Willis, 2015*). Its inhibition is usually achieved with compounds that bind at the protein active-site, competing either with the dUMP substrate, such as 5-fluorodeoxyurdine 5'-monophosphate (FdUMP), or with the folate cofactor, such as raltitrexed (RTX) and pemetrexed (PMX) (*Carreras and Santi, 1995*; *Jennings and Willis, 2015*; *Figure 1A and B*). However, these drugs induce cells to develop drug resistance associated with increased hTS levels which eventually leads to therapy failure (*Peters, 2018*). A drastic change of strategy, based on the design of new compounds with different mechanisms of action, is thus necessary (*Voeller et al., 2002*; *Cardinale et al., 2011*). Here we report the discovery of molecules that bind at the hTS monomer/monomer interface and, despite their small size, markedly shift the monomer-dimer equilibrium of the enzyme towards the inactive monomeric form. Acting as destabilizers of the hTS dimer, they not only inhibit the activity of this obligate homodimeric enzyme but also favor its intracellular degradation and, hence, decrease its level.

Using a tethering approach in which sulfhydryl-containing fragments were initially identified by reaction with a cysteine residue inserted by mutation around the target site, we identified fragments anchored by disulfide bond formation at the interfacial mutant cysteine residue. We then employed molecular modeling and medicinal chemistry to modify the fragments and develop inhibitors able to bind WT hTS guided solely by their affinities for the interface region. For these small molecules, we propose a model able to quantitatively account both for the dissociation of the hTS dimer, observed spectroscopically, and for the corresponding activity inhibition. We have also shown that their ability to induce the TS-dimer dissociation in cells lies below their efficacy in inhibiting growth of colon, ovarian and pancreatic cancer cells. One of our hTS-dimer destabilizers, compound **E7**, induced apoptosis in cancer cells and a decrease of hTS levels due to enhanced proteasomal degradation of the enzyme monomers with respect to the dimers. Remarkably, in a mouse model of orthotopic pancreatic cancer, this dimer disrupter caused a higher reduction of cancer growth and had lower toxicity than 5-fluorouracil (**5-FU**), the prodrug of 5-FdUMP. Like what we observed in vitro, the molecular analysis

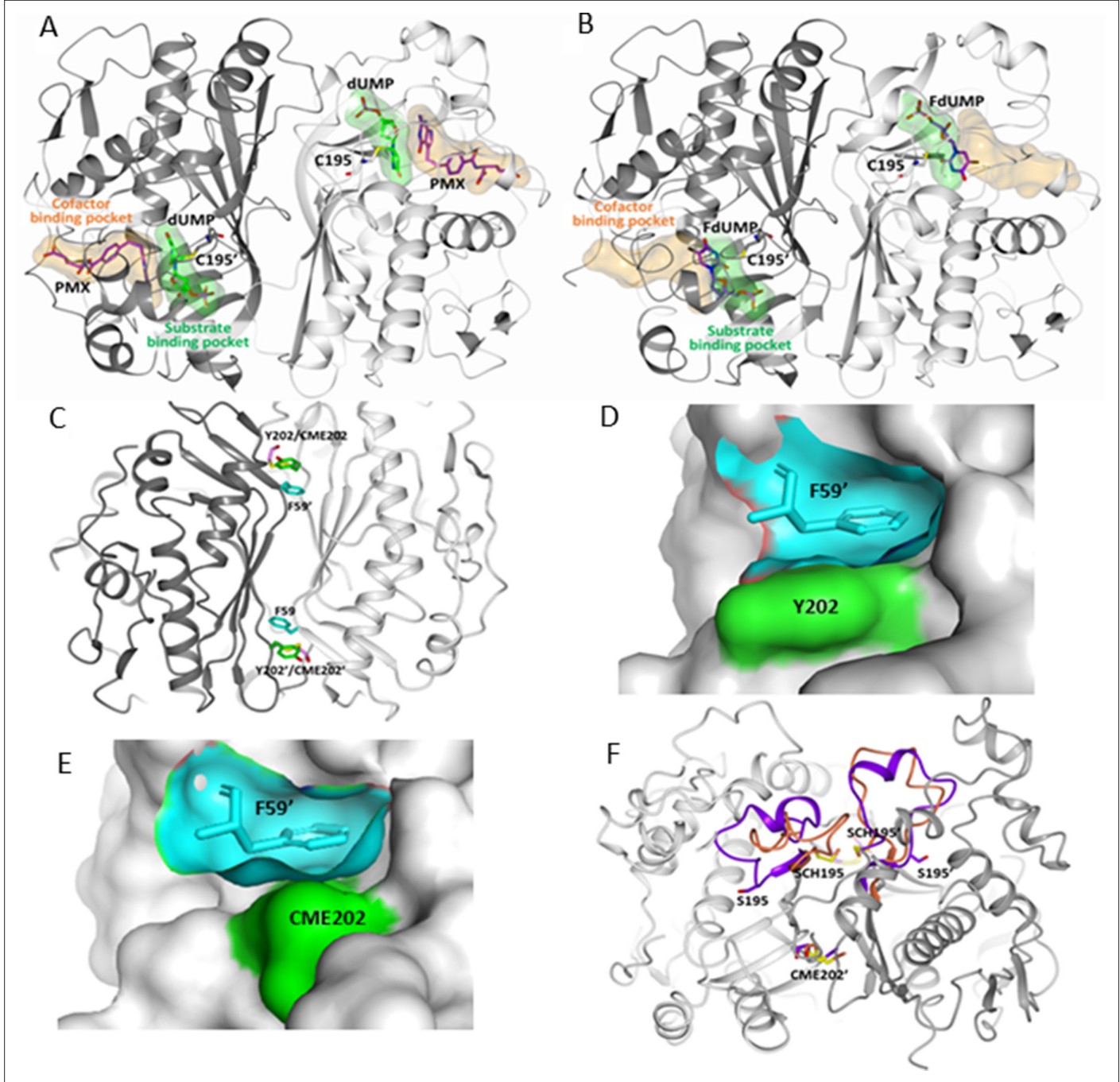

**Figure 1.** Drug target sites on hTS: the active-site (**A,B**) and the Y202 pocket (**C–F**). Drugs that are directed to the active site of hTS can mimic either the cofactor MTHF or the substrate dUMP and bind in their binding pockets. (**A**) hTS (in cartoon with the A and B subunits colored white and grey, respectively) inhibited by the cofactor analogue pemetrexed (PMX, in sticks, purple carbons), which occupies the cofactor binding pocket by establishing a π-π interaction with the dUMP substrate (in sticks, green carbons) (PDB ID: 1JU6). The substrate and cofactor binding sites are represented as green and orange surfaces, respectively. The catalytic cysteine, C195, is highlighted in sticks. (**B**) The substrate analogue 5-fluoro-2′-deoxyuridine monophosphate (**FdUMP**) (in sticks, purple carbons) binds inside the catalytic cavity, mainly filling the substrate binding pocket (PDB ID: 3H9K, hTS variant R163K). (**C**) Location of F59, Y202 and CME202 residues at the hTS dimer interface by superimposition of native hTS and hTS-C195S-Y202C double mutant structures (CME: S,S(2-hydroxyethyl)cysteine; PDB-ID: 3N5G, 4O1X). (**D**) Surface of the Y202 pocket (cyan) filled by F59′ from the opposite monomer (cyan sticks). The surface of Y202 is highlighted in green (PDB-ID: 3N5G). (**E**) Surface of the C202 pocket (cyan) filled by F59′ from the opposite monomer (cyan sticks) in the C195S-Y202C double mutant (PDB-ID: 4O1X). The surface of C202 (oxidized as CME) is highlighted in green. (**F**) Superimposition of the hTS-C195S-Y202C and hTS-Y202C crystal structures with the catalytic loop in hTS-Y202C (inactive conformation, in orange) and in hTS-C195S-Y202C (active conformation, in violet); notice the different orientations of the catalytic residue couples S195/SCH195 and S195′/SCH195′

*Figure 1 continued on next page*

*Figure 1 continued*

(PDB-ID: 4O1X, 4O1U) (SCH: S-methyl-thio-cysteine). The one-letter code is used for the standard amino-acids, while for the chemically modified amino-acid, a conventional three-letter code is used as described above.

The online version of this article includes the following figure supplement(s) for figure 1:

**Figure supplement 1.** Y202 and K47 pockets at the hTS monomer-monomer interface (PDB ID: 1HVY, A chain) are represented.

of exported treated cancer tissues demonstrated decreased hTS protein and mRNA levels. These findings establish **E7** as a new lead with an hTS dimer-disruptive ability and link the dimer-to-monomer equilibrium shift of this protein to its faster intracellular degradation.

## Results

### Choice of the ligand binding-site on the target protein surface

To target the monomer-monomer interface and dissociate the hTS dimer, we applied the 'cysteine tethering' approach to fragment-based drug design (*Erlanson et al., 2000*). We investigated the crystal structures of the active and inactive hTS (PDB-IDs 1HVY and 1YPV, respectively) and carried out molecular dynamics (MD) simulations of the monomeric and dimeric forms of the enzyme to explore the monomer-monomer interface of homodimeric hTS and identify targetable binding pockets and residues suitable to be mutated to cysteine for the tethering experiments (*Supplementary file 1*). We selected the 'Y202 pocket', which accommodates the side chain of F59' from the other monomer, as the most promising region to target (*Figure 1C–F*, *Figure 2A–B*, *Figure 1—figure supplement 1*; *Salo-Ahen et al., 2015*).

This pocket lies in the perimeter space of the intermonomer interface and is accessible in the dimeric form of the protein. It is large enough (22 Å$^3$) to accommodate a phenyl group and has several smaller crevices around it. During the MD simulations of monomeric hTS, the side chain of Y202 moved away leaving a larger pocket and behaving like a gate. We therefore made the Y202C mutant and screened for compounds able to form a covalent disulfide bond with C202 from a library of organic disulfides. We also mutated the active site cysteine to serine to prevent the selection of compounds that bound at the active site (C195S). We made mutants with single point mutations at C195S and Y202C to test their functional and structural properties (*Supplementary file 1*, *Figure 1—figure supplement 1*) as well as the double mutant, hTS C195S-Y202C. This was used for the ligand selection according to the above-described tethering approach. As expected from removal of the catalytic C195 residue, this double mutant was inactive, but X-ray crystallography showed that it takes the active conformation of hTS (*Figure 1F*, *Figure 2—figure supplement 1*, *Supplementary file 2A and B*).

We employed the C195S-Y202C double mutant to capture compounds that, driven by some affinity for the region near residue 202, formed a covalent disulfide bond with C202 (*Supplementary file 3A*). These compounds were selected by their stability towards reduction by β-mercaptoethanol (BME), and the C202-S-S-small-compound disulfides formed were identified by two mass spectrometric (MS) experiments, MALDI-TOF and ESI-QTOF (*Supplementary file 3B and C*).

### Mass-spectrometric ligand selection

To identify molecules able to covalently bind near Y202, we designed a library of commercially available compounds from the ZINC database (*Irwin and Shoichet, 2005*; *Figure 2C*). We first selected 1066 disulfide compounds that were either symmetric (RSSR) or asymmetric (R$_1$SSR$_2$) and had drug-like or fragment-like properties. 1066 molecules were screened for their molecular properties (PCA, MW, logP, size) and 55 were evaluated with the MS techniques. Thirteen out of the 55 compounds were found to bind the protein according to the results of both MS techniques (*Figure 2D and E*, *Appendix 1*, *Supplementary file 3A, B and C*). Among them, seven compounds, **A5, A6, A10, A15, A20, A38,** and **TPC**, were selected based on the feasibility of their chemical modification and were further analyzed by a bottom-up MALDI-TOF and ESI-QTOF MS proteomic analysis to identify the cysteine residues they had bound (*Supplementary file 3C*). The 7 compounds bound either peptides (I177-R185) IIMCAWNPR (sequence A) or (R176-R185) RIIMCAWNPR (sequence B), both containing the C180 residue of the digested protein (*Supplementary file 3C*). Four molecules, **A5, A10, A20,** and **TPC** also bound peptide (D186-R215) DLPLMALPPSHALCQFCVVNSELSCQLYQR (sequence C)

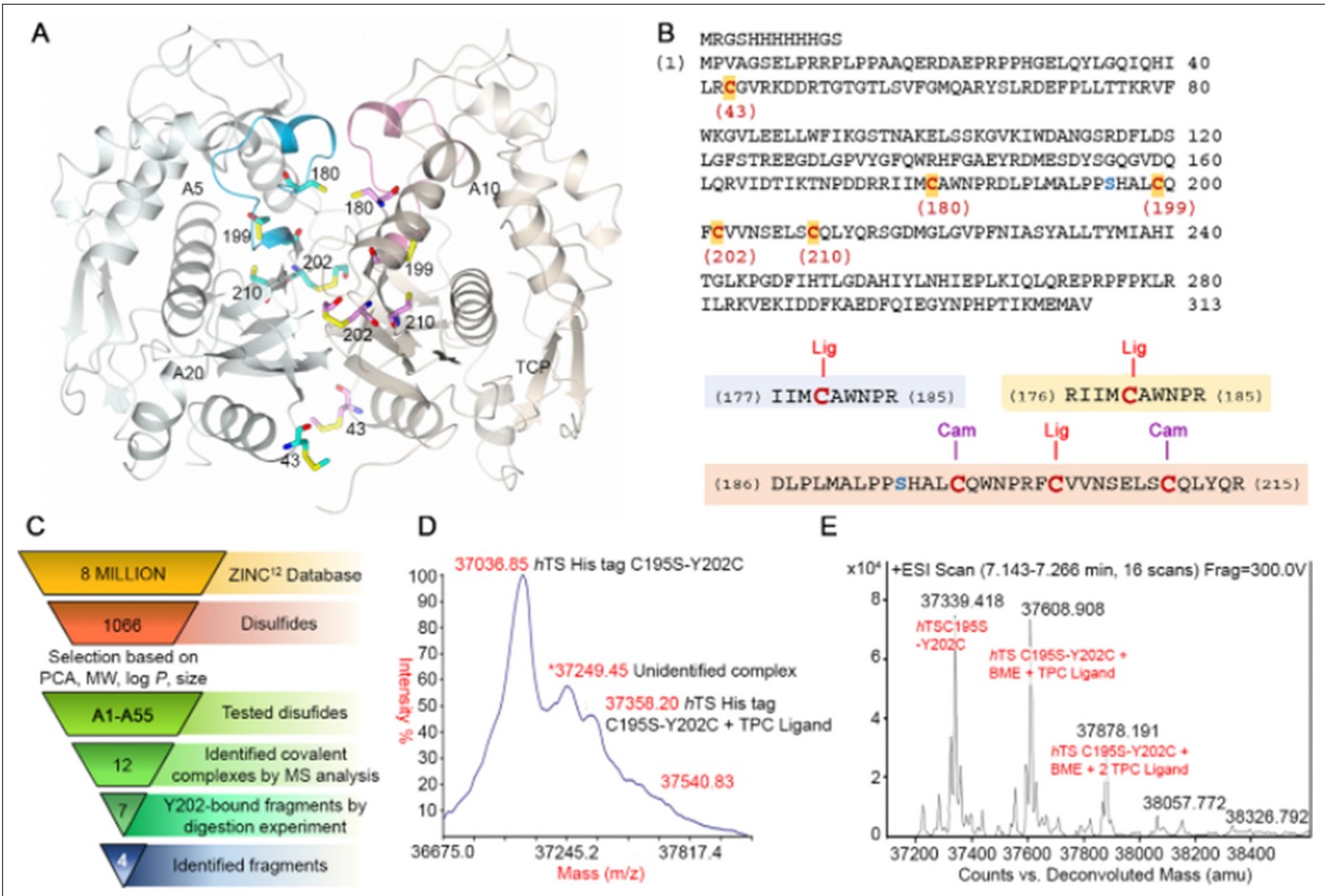

**Figure 2.** Disulfide library design and screening on the hTS C195S-Y202C variant. (**A**) Crystal structure of the hTS C195S-Y202C dimer (in the cartoon, the A and B subunits are colored pale blue and grey, respectively; the catalytic loops, in active conformation, are highlighted in cyan and pink, respectively) showing the positions of the cysteine residues (in sticks, carbon atoms are colored cyan and pink in subunits A and B, respectively). Compounds A5, A10, A20 and the reference compound N-tosyl-D-prolinecysteamine (TPC) (**Appendix 2—figure 1**) were selected through the tethering experiments. They are indicated close to the peptide sequence found in the mass spectrometry identification process (**Supplementary file 3B and C**). Their chemical structures are reported in **Supplementary file 3A**. (**B**) Sequence of the hTS C195S-Y202C variant and cysteine-containing peptides generated by trypsin digestion. Cysteine residues are displayed in red (the residue number is reported below each cysteine) and the mutated S195 (the catalytic residue) in blue. Three cysteine-containing peptides were generated by proteolytic cleavage of the hTS C195S-Y202C variant: peptide I177-R185 and peptide R176-R185 containing C180 and peptide D186-R215, containing C199, C202, and C210. Ligand (Lig)-bound cysteines and those alkylated as carbamidomethyl (Cam)-cysteines are displayed. (**C**) Screening cascade from the 6 million compounds from the ZINC database (**Irwin and Shoichet, 2005**) to the final four fragments expected to bind in the Y202 pocket. (**D, E**) Representative MS spectra in the hTS-binding evaluation step. Maldi (**D**) and ESI (**E**) MS spectra for the reference compound TPC (**Erlanson et al., 2000**). The three labeled peaks are due to hTS C195S-Y202C and to the protein bound to BME and one and two molecules of TPC.

The online version of this article includes the following figure supplement(s) for figure 2:

**Figure supplement 1.** X-ray crystallographic structure of hTS-Y202C and hTS C195S-Y202C.

(**Supplementary file 3C**), containing the engineered Y202C together with the native C199 and C210. In agreement with the described computational and crystallographic data (see above), Y202C is the most exposed of these three cysteines and is thus easily available for disulfide-bond formation with the RS fragments, being therefore suitable for tethering experiments. Other reasons to select the Y202 pocket are that Y202 is not predicted to be part of the mRNA binding region and has the capability to interact with various functional groups. Hence, despite the difficulties in the identification of the precise cysteine to which the fragments could bind, all the above arguments support the choice of the Y202 pocket at the hTS interface as a suitable binding site for further structure-based studies.

Therefore, we considered the fragments **A5, A10, A20,** and **TPC** for further medicinal chemistry modifications (*Figure 3A*).

## From low-affinity compounds to TS homodimer destabilizers

Starting from the four fragments identified by tethering/MS (**TPC**, **A5**, **A10**, **A20**) (*Figure 3A*), we performed an analogue search that yielded a dataset of 331,600 commercially available compounds (Specs dataset, https://www.specs.net) potentially able to bind non-covalently at the Y202 pocket of the monomer/monomer interface. The initial dataset was progressively reduced to 5,774 candidates using MW, structural and chemical criteria (*Figure 3B*). These were then docked into the Y202 pocket with the software FLAP (*Cross et al., 2010*) using a receptor-based pharmacophore model built from the X-ray crystal structure of hTS (PDB: 1HVY). A final set of 26 molecules (**B1-B26**, *Supplementary file 4A*) was selected to be tested for their ability to inhibit recombinant hTS and destabilize its dimeric form using, respectively, a kinetic and a spectroscopic, FRET (Förster resonance energy transfer)-based assays (*Ponterini et al., 2016*). For the FRET-based assay (*Genovese et al., 2010*), we conjugate one protein monomer with red-emitting tetramethylrhodamine (T) and the other with green-emitting fluorescein (F). Only when the monomers are close to each other, that is, combined to form a dimer, F-to-T excitation energy transfer occurs, a FRET signal is detected, and its efficiency determined. We can thus investigate the dimer/monomer equilibrium of hTS, characterize it by the corresponding dissociation equilibrium constant and evaluate the effect of a tested compound thereupon.

Among the 26 compounds selected by VS, **B12** and **B26** (*Figure 3B*, *Supplementary file 4A*), caused low but reproducible decreases in the efficiency of FRET ($\Delta\Phi_{FRET}$ = -0.13 and $\Delta\Phi_{FRET}$ = -0.02, respectively), thus indicating some ability to perturb the enzyme dimeric assembly. In the docked poses, interactions were established by **B12** with Y202 through the *o*-chloro-phenyl ring and with R175 through the *p*-nitro-pyridyl ring, and by **B26** with Y202 through the piperidin-1-yl ring and with R175 through the quinoline ring (*Figure 3C and D*).

Next, we designed and synthesized four sets of compounds (**C-F**) (*Supplementary file 4*) to develop **B26** into biologically active compounds that exhibited the ability to both disrupt the dimer and inhibit recombinant hTS (*Appendix 2—figures 2–7*). The most relevant compounds were **C2**, **C3**, **C4**, **E5**, **E6,** and **E7** (*Supplementary file 4B-E*). The former three, differing only for the position of the carboxylic group on the phenyl ring, featured IC$_{50}$ values 5.25, 83 and 246 µM vs. 300 nM hTS (*Supplementary file 4*) and were selected for a quantitative mechanistic analysis of FRET and inhibition data with recombinant hTS (see the next paragraph). It was however compounds **E5**, **E6,** and **E7** that proved to be the most interesting to continue our study. They featured IC$_{50}$ values against 300 nM recombinant hTS of 40, 10 and 7 µM respectively, and decreased FRET efficiency by –0.24 at 50 µM (**E5**, **E6**) and –0.1 at 10 µM (**E7**, *Supplementary file 4*). Compound **E7**, that was the most active on cells, was determined to be a racemic mixture of two enantiomers that equilibrated with a half-life of 56 min (Appendix 3, *Appendix 3—figure 1*, *Appendix 3—figure 2*).

Additional evidence of the ability of compounds **C3** and **E5** to bind hTS and destabilize its dimers was provided by anion-exchange (AE) chromatographic experiments. Two bands, separated by ca. 2 min, both corresponding to fractions characterized by the absorption spectrum of hTS, were found in the AE chromatogram based on absorbance at 280 nm (*Figure 3—figure supplement 1A*). The relative area of the second band was only about 3.5% that of the first one and decreased further (*Figure 3—figure supplement 1B*) when the protein was loaded together with dUMP and Raltitrexed, both at 250 µM, a concentration more than one order of magnitude larger than their dissociation constants from hTS. Because binding of hTS to its nucleotidic substrate stabilizes the active, dimeric form of the enzyme, and consistently with the expectations based on protein size and the correlated elution times, we attribute the two bands to the hTS dimers the first one, and monomers the second. In keeping with this assignment, when hTS was loaded on the AE column after a 1 hr incubation with compound **C3** at three concentrations, namely 4, 40, and 80 µM, the monomer band was delayed by additional 3–4 min relative to the dimer band in the chromatograms (*Figure 3—figure supplement 1A* and *Figure 3—figure supplement 1C–E*) and its relative area increased markedly, from 0.035 to 0.08, to 0.24 and to 1.09, respectively. Co-elution, hence stable non covalent binding of the **C3** inhibitor with hTS, was established by the observation that the fractions containing the hTS dimers and monomers also exhibited absorption at 350 nm (*Figure 3E*), a wavelength at which no absorption

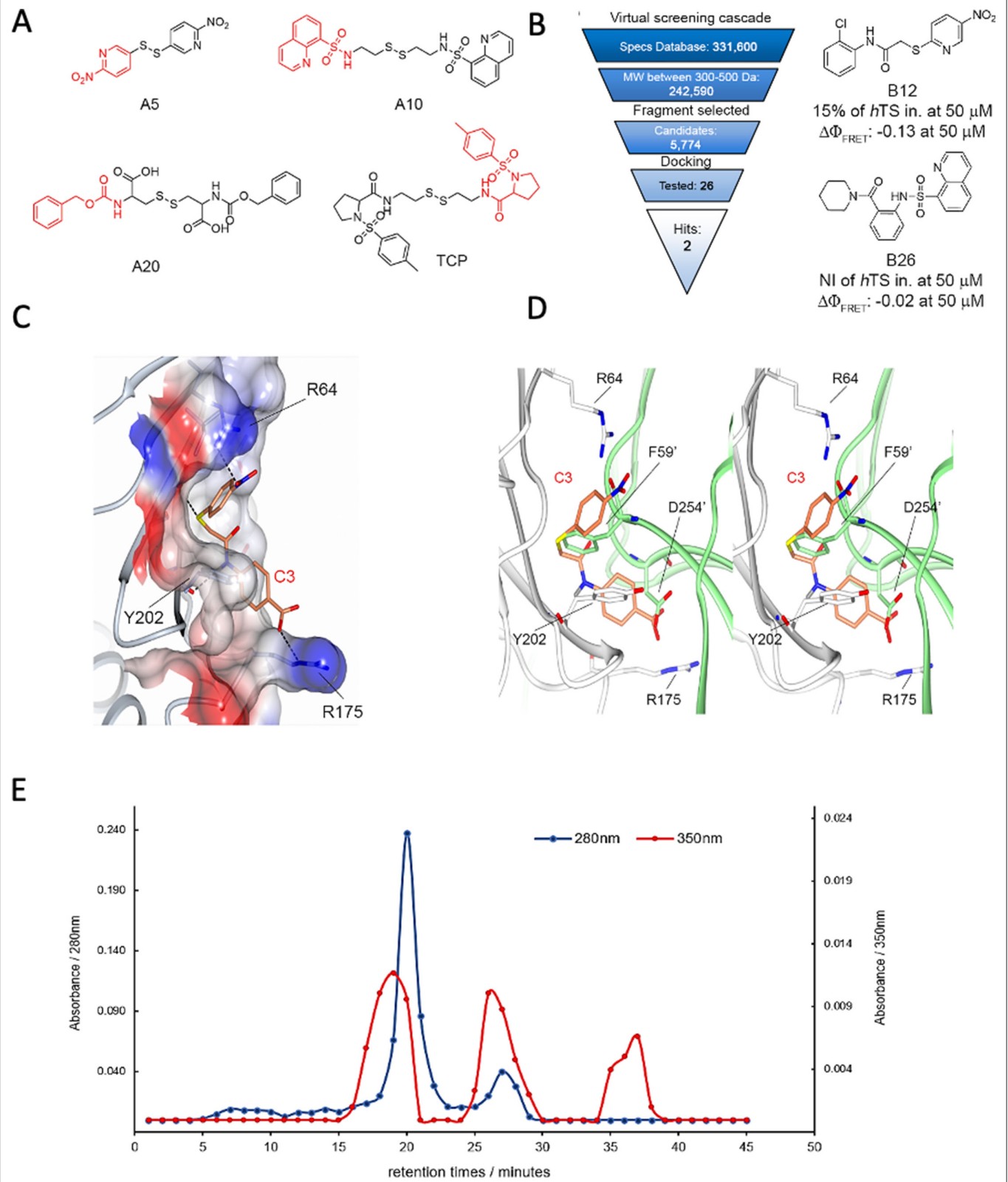

**Figure 3.** From hits to dimer-destabilizing compounds. (**A**) Structures of the compounds identified in MS studies (in red the fragments used for the virtual screening approach to the design of the compound library without the disulfide bonds). (**B**) Virtual screening cascade and chemical structures of the identified compounds, B12 and B26. (**C**) Detail of the binding mode of compound C3 (orange sticks) with the hTS interface residues R64, Y202 and R175 (positive and negative potential regions are colored in blue and red, respectively) obtained with docking methods. (**D**) Stereo view of the

*Figure 3 continued on next page*

Figure 3 continued

superimposition of the calculated binding mode of C3 onto the first monomer (white cartoon) with the opposite monomer overlapping (lime cartoon). Atom color code: nitrogen – blue; oxygen – red; sulphur – yellow. (**E**) Results of anion exchange chromathography showing co-elution experiments of C3 with hTS dimer and monomer through anion exchange chromatography. Absorbances at 280 (blue) and 350 nm (red) of chromatographic fractions obtained by loading a 50 μM hTS and 4 μM **C3** solution on an anion-exchange AEX HiPrep Q HP 16/10 mm column. (***Figure 3—figure supplement 1***, ***Figure 3—figure supplement 1—source data 1***).

The online version of this article includes the following source data and figure supplement(s) for figure 3:

**Figure supplement 1.** Anion-exchange chromatograms obtained with a 280 nm detection (blue traces) by loading on an AEX HiPrep Q HP 16/10 mm, Cytiva, column.

**Figure supplement 1—source data 1.** Anion-exchange chromatograms of hTS obtained with a 280 nm and 350 nm detection.

**Figure supplement 2.** Profile of the variation in concentration of compound C3 in the AE chromatographic fractions determined by high-resolution MS (UltiMate 3000 UHPLC tandem Orbitrap Q-Ex).

**Figure supplement 2—source data 1.** Profile of the variation in concentration of compound C3 in the AE chromatographic fractions determined by high-resolution MS (UltiMate 3000 UHPLC tandem Orbitrap Q-Ex).

comes from the protein. Accumulation of compound **C3** in these fractions was confirmed by identifying and quantifying it in the AE chromatographic fractions by high-resolution mass spectrometry (***Supplementary file 5***, ***Figure 3—figure supplement 2***).

Absorbances at 350 nm in ***Figure 3E*** indicated similar concentrations of the compound in the fractions corresponding to the protein dimers and monomers, in spite of the fact that the 280 nm dimer chromatographic band was much more intense than the monomer band. This finding suggests a somewhat larger affinity of the ligand for the latter. Consistently, parallel MS experiments in which 4 μM **E5** was loaded on the AE column with hTS identified this ligand mainly in the fractions containing the protein monomers (see additional information in the ***Supplementary file 5***). So, overall, the results synthesized here show that these small compounds co-elute with hTS, thus proving beyond doubt that they remain bound to it, possibly to the monomer more tightly than to the dimer, throughout the AE chromatographic runs. Also, they qualitatively show the ability of these ligands to de-stabilize the hTS dimers in favor of the monomers.

## Mechanistic analysis of dissociative inhibition

The nitrothiophenyl derivative, **C3** and its isomers, **C2** and **C4** (***Supplementary file 4B***), are structurally similar to compound **E7** (***Supplementary file 4D***) and are much more soluble in water. So, they were used for quantitatively investigating the molecular mechanistic basis of their ability to both inhibit recombinant hTS and disrupt its dimer. The three isomeric compounds showed different inhibitory potencies against 300 nM hTS, with $IC_{50}$ values of 5.25, 83 and 246 μM, respectively. Consistently, **C3** caused the largest decrease in FRET efficiency ($\Delta \Phi_{FRET}$) at concentrations lower than 50 μM (***Figure 4A***, ***Figure 4B***).

The dependence of $\Phi_{FRET}$, that in our conditions is a direct measure of the mole fraction of hTS dimers, on each inhibitor concentration (***Figure 4B***) was fitted according to the dimer-monomer equilibrium model sketched in ***Figure 4C***. This was analyzed as reported in the *Appendix 4*. We searched for evidence of the disruptive character of compound **C3** by investigating the dependence of its dose-effect curves on total protein concentration, $E_T$ (***Figure 4D***).

As expected for a dissociative inhibition mechanism, the limit values of $\Phi_{FRET}$ at high **C3** concentrations were consistent with a 5-fold increase in the equilibrium dissociation constant for the protein bound to **C3** with respect to the free protein (***Figure 4E***).

From the common limit values of $\Phi_{FRET}$ at high concentrations (***Figure 4B***), we conclude that, when saturating hTS, compounds **C2-C4** have similar effects on the dimer stability. The fivefold increase in the protein-dimer dissociation equilibrium constant and the corresponding decrease in $\Delta G°$ for dimer disruption, from 40.5 to about 36.5 kJ mol$^{-1}$, $\Delta\Delta G° = -4$ kJ mol$^{-1}$, are evidence of the dimer-destabilizing ability of compounds **C2-C4**. On the other hand, their different abilities to decrease the FRET efficiency at lower concentrations can be attributed to different affinities for the protein dimers, with $K_I$"s of $2.8 \times 10^{-5}$, $1 \times 10^{-5}$ and $2 \times 10^{-5}$ M, for **C2**, **C3**, and **C4**, respectively. Thus, the observed overall effect of such an inhibitor results from a combination of affinity for the protein and ability to destabilize its dimeric assembly by impairing crucial attractive inter-monomer interactions.

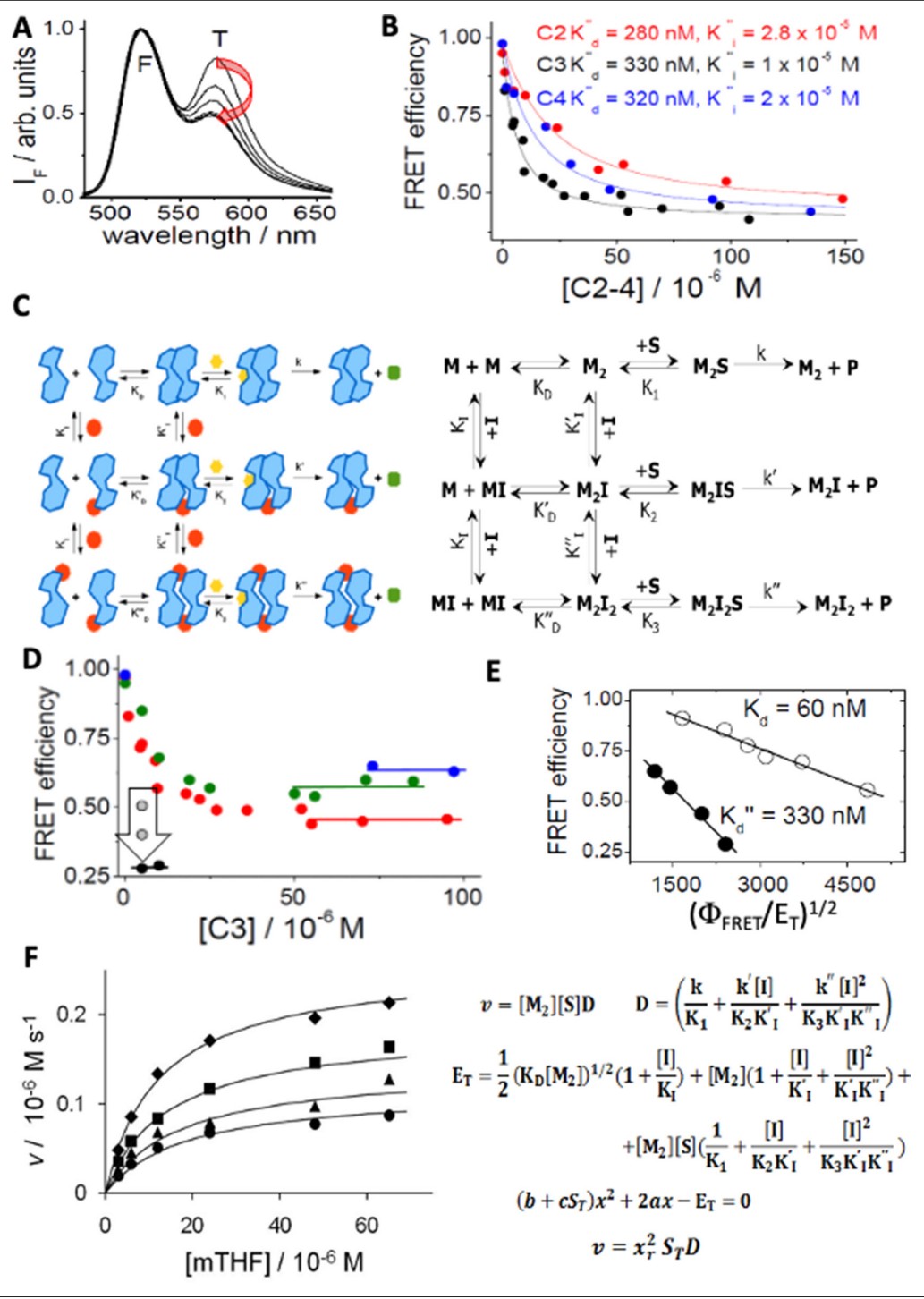

**Figure 4.** Spectroscopic and mechanistic analysis of hTS dissociative inhibition for compounds C2, C3 and C4. (**A**) Emission spectra of fluorescein(F)- and tetramethylrhodamine (T)-labelled hTS ($\lambda_{exc}$ = 450 nm): the decreasing emission contribution of T relative to F with increasing concentrations of C3 (0, 4.5, 9.5, 15, 25, 100 μM) parallels the decrease in the hTS dimer mole fraction. (**B**) Dependence of the observed FRET efficiency on the concentrations of three inhibitors (C3, black; C4, blue; C2, red); the hTS dimer concentration was 100 nM. The lines represent the best fittings of the experimental results to Equation S3 in the *Appendix 4*. The most significant fitting parameters are reported. (**C**) Cartoon, and standard chemical representations of the dissociative inhibition mechanism of hTS. Blue shapes represent enzyme monomers (M) in the active conformation bound to dUMP; red dots indicate a dissociative inhibitor (I), yellow hexagons, the folate substrate (S), green octagons, the product. (**D**)

*Figure 4 continued on next page*

*Figure 4 continued*

Dependence of FRET efficiency on the concentration of C3 with total protein concentrations ($E_T$) of 490 nM (blue circles), 250 nM (green circles), 100 nM (red circles) and 50 nM (black circles). The horizontal segments represent the values of the efficiency in the limit of high inhibitor concentration. At $E_T$ = 50 nM and $I_T$ = 5 µM the vertical arrow indicates the evolution of the measured value of the FRET efficiency 5, 7, and 12 minu after inhibitor addition (grey and black circles). (**E**) Analysis of the hTS monomer-dimer equilibrium according to eq. $\Phi_{FRET} = 1-0.5(\Phi_{FRET}/E_T)^{1/2}K^{1/2}$ (*Genovese et al., 2010*) and corresponding equilibrium constants without C3 (white circles, $K=K_D$) and at saturating C3 concentrations (black circles, $K=K''_D$, $\Phi_{FRET}$ = FRET efficiency). (**F**) Dependence of the initial reaction rate ($v$) on mTHF concentration (free substrate, [S], and total, $S_T$) at four different C3 concentrations (from top to bottom, [I]=0, 12, 24, and 36 µM). The fitting curves represent equation $v = x_r2S_TD$ derived in the *Appendix 4* and computed with $E_T$ (total enzyme dimer concentration)=300 nM, k=k'=0.9 s$^{-1}$, k''=0.1 s$^{-1}$, $K_1$=$K_2$=10$^{-5}$ M, $K_3$=2 × 10$^{-5}$ M, $K_I$ = 10–6 M, $K_I'$ = $K_I'$'=10$^{-5}$ M,x = $[M_2]^{1/2}$ and using for $x_r$ the only acceptable root of the quadratic equation in panel 4 F (*Figure 4—source data 2*).

The online version of this article includes the following source data for figure 4:

**Source data 1.** Spectroscopic and mechanistic analysis of hTS dissociative inhibition for compounds C2, C3 and C4.

**Source data 2.** Inhibition of hTS by compound C3.

---

Binding of compounds **C2-C4** at the Y202 pocket of the hTS monomer/monomer interface is supported by computational docking (*Figure 3C and D*). In the hTS dimer, the aromatic ring of the Y202 residue forms a π-π interaction with the phenyl ring of the F59' residue from the opposite subunit. Such a ring is replaced by the phenyl ring of the benzoic acid portions of **C2-C4** in their modeled complexes with hTS. Thus, these compounds mimic some of the residue side chains from the opposite monomer in the dimer, providing similar interactions in this region. This finding is supported by the higher affinities of the inhibitors for the hTS monomer ($K_I$ = 10$^{-6}$ M) than for the dimer ($K_I'$=1.7 x 10$^{-5}$ M), as obtained by fitting of the FRET data in *Figure 4B*.

The reliability of our docking virtual experiments was strengthened by their ability to explain the observed higher affinity for hTS of compound **C3** relative to its isomers, **C2** and **C4**. While the nitro-groups of all these ligands can make head-to-head hydrogen-bonding interactions with R64 (*Figure 3C–D*), only the *para* carboxylic group of compound **C3** arranges properly to establish an additional H-bond with R175. Instead, the *meta* and *ortho* carboxylic groups of **C2** and **C4** do not have the correct orientations for interacting with this arginine residue.

As for the mechanism of inhibition of the enzymatic activity of hTS by compounds **C2-C4**, we show that the scheme in *Figure 4C* can account for the dependence of the initial reaction rate ($v$) on the substrate concentration at different inhibitor concentrations with very reasonable fitting parameters (*Figure 4F*). Simple observation of the dependence of $v$ on [**C3**] with total enzyme concentration ($E_T$) 300 nM (*Figure 4F*) indicates a regular decrease with increasing [**C3**] to a limiting rate that is about 1/5 the rate in the absence of inhibitor. On the other hand, the $E_T$-dependent limiting value of $\Phi_{FRET}$ was about half the full one (*Figure 4B*). This difference in the limit behaviors of $v$ and $\Phi_{FRET}$ suggests that, when bound to the inhibitor, the dimeric enzyme is less catalytically active than when free. Or, in terms of the dissociative/noncompetitive model in *Figure 4C*, $K_2$ and $K_3$ are larger than $K_1$, or k' and/or k'' are smaller than k. The curves that go through the kinetic data in *Figure 4F* are plots of eq. $v = x_r2S_TD$ that was obtained by fast-equilibrium solution of the scheme in *Figure 4C* (for the derivation of the equation and the meanings of the symbols therein, see the *Appendix 4*). Remarkably, the fitting parameters, provided in the caption, are consistent with the results of the FRET-based analysis of the equilibria in *Figure 4C* and the limit behaviors of the initial rate and of $\Phi_{FRET}$. More in detail: (i) k'' is much smaller than k and k', that is, $M_2I_2$ is almost catalytically inactive, a feature typical of noncompetitive inhibition; (ii) $K_3$ is larger than $K_{1,2}$, i.e., inhibitor binding reduces the enzyme affinity for MTHF; (iii) $K_I'$ and $K_I''$ remain one order of magnitude larger than $K_I$; i.e., when saturated with dUMP, the enzyme dimer shows lower affinity for the inhibitor than the monomer; (iv) using the relationships among the equilibrium constants, we estimate that $K_D'/K_D = K_I'/K_I \approx 10$ and $K_D''/K_D'=K_I''/K_I \approx 10$, that is, the kinetic results confirm a progressive dissociation of the enzyme dimer to monomers upon subsequent additions of **C3**. The latter is therefore confirmed to be both a dimer disrupter and a dissociative inhibitor, the two roles being intimately connected with each other.

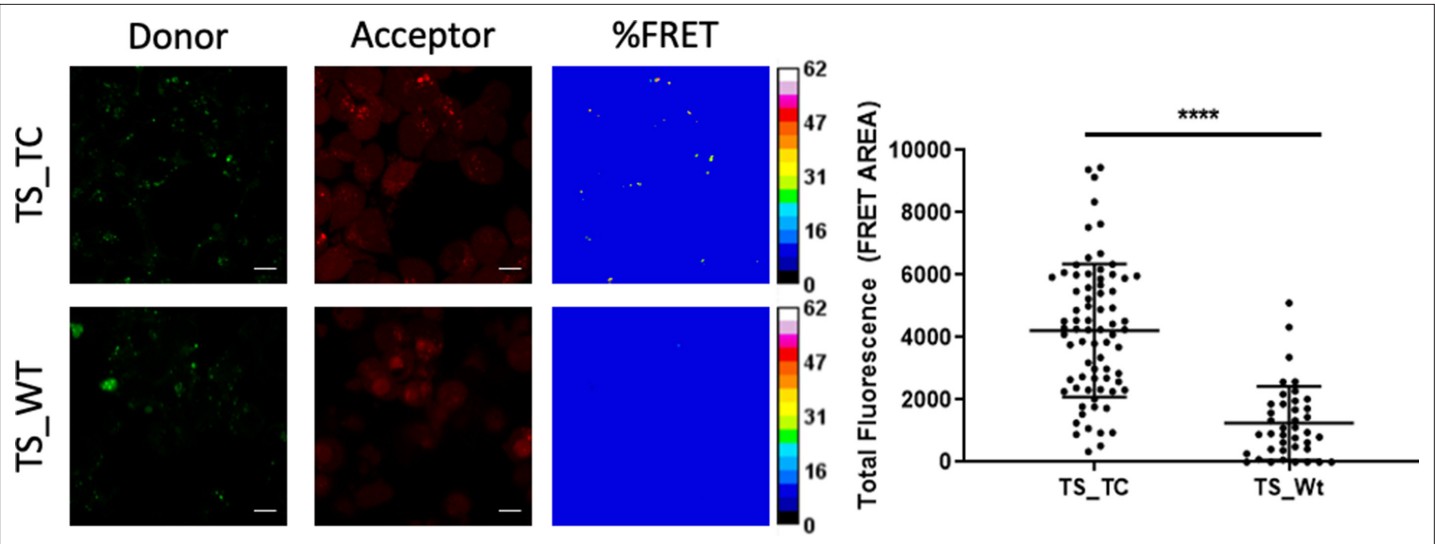

**Figure 5.** Fluorescence microscopy analysis of hTS dissociative inhibition. Representative fluorescence microscopy images of TS-TC- and WT-TS-expressing HT-29 cells stained with red-emitting ReAsH and incubated with green-emitting E5-FITC. From left to right: green-emission and red-emission channels, and FRET efficiencies depicted according to the colour scale (%) in the right bars. Right: comparison of the average FRET areas for cells overexpressing TS with (TS_TC) or without (TS_Wt) ReAsH binding sites, obtained using PixFRET within ImageJ. Each dot represents the total FRET area of an individual field with at least 10 cells. The statistical analysis was performed using unpaired two tailed t-test (p-value >0.0001) using Prism 8 for windows (version 3.1.1).

## The dimer destabilizers engage hTS in cancer cells

To gain evidence that our hTS dimer destabilizers engage this enzyme in cells, we resorted to fluorescence microscopy. For this experiment, we selected compound **E5** to be incorporated in the final conjugated probe because its chemical precursor, **E3**, has a reactive carboxylic group (*Supplementary file 4D*) and is therefore suitable for probe conjugation chemistry. **E6** and **E7** do not have the same suitable precursors. Using a short ether linker, we conjugated it to fluorescein isothiocyanate (FITC), a green-emitting probe that favors cell internalization (*Appendix 2—figures 8–10*). Indeed, cells were permeable to the **E5-(O₂Oc)-CAM-FITC (E5-FITC)** conjugate. It proved slightly less active at inhibiting recombinant hTS than **E5** (IC$_{50}$s=40 and 100 μM, for **E5** and **E5-FITC,** respectively), likely because the probe tail caused some steric hindrance for its binding to the enzyme. For these experiments, we ectopically expressed in HT-29 cells either Wild Type (WT) hTS or a mutant with a tetracysteine motif (CCPGCC, TC) inserted near the N-terminus (TS-TC) (*Ponterini et al., 2016*). The tetracysteine motif tightly binds ReAsH, a rhodamine-based diarsenical probe. The ReAsH/TC-TS complex emits red fluorescence with much higher efficiency than the unbound probe. HT-29 cells (transfected with either WT TS or TC-TS) were treated with ReAsH and **E5-FITC**. In the ReAsH/TC-TS/**E5-FITC** complex, fluorescein transferred excitation energy to rhodamine thus signaling formation of the complex with a FRET signal. Indeed, our image analysis (*Figure 5*) showed that cells expressing TC-TS produced a significantly higher ($P$ <0.0001) average FRET signal (4222±2128) than control cells that expressed WT TS (1250±1178), thus proving the occurrence of intracellular hTS/**E5** engagement.

## Cancer cell growth inhibition by hTS dimer destabilizers

The compounds that both inhibited recombinant hTS and decreased the FRET efficiency, that is, **C2-C5**, **C9**, **C10**, **C13**, **D5-D9, D12**, **E1** and **E3-E7**, were selected to test cancer-cell growth inhibition. They were tested as racemates, employing **5-FU,** metabolized to **FdUMP** inside cells, as a reference compound. Tests were performed on several model cancer cell lines. Because **5-FU** is a key therapeutic in the pancreatic-cancer first-line regimen FOLFIRINOX, we chose three pancreatic adenocarcinoma primary cell cultures, PDAC-2, PDAC-5, and LPc167, which express low, intermediate and high TS levels, respectively (*Funel et al., 2008*, *Firuzi et al., 2019*). These cell lines showed a similar trend in the TSmRNA levels (3.93±0.58, 11.49±1.39, 49.02±2.76, respectively) (*Figure 6—figure supplement 1*). These cultures were obtained from radically resected patients which reflected

the histopathological and genomic features of human PDACs. Noteworthy, **E7** was almost as equally active against both PDAC2 and PDAC5 cells. Both were sensitive to **5-FU** (*Supplementary file 6A*). The LPc167 primary culture, which expressed high levels of TS was resistant to **5-FU** but still sensitive to **E7** (*Figure 6—figure supplement 2*).

Other models used in our investigation were two ovarian cancer cell lines, IGROV1 and TOV112D, and two gynecological cancer cell lines that feature high hTS levels resulting from cross-resistance to cisplatin (C13*, A2780/CP) as well as the corresponding cisplatin-sensitive lines (2008, A2780) (*Cardinale et al., 2011*; *Figure 6A*, *Figure 6—figure supplement 3*, *Supplementary file 6B*).

Since **5-FU** is always included in the treatment of colorectal cancer, we added three epithelial colorectal cancer cell lines, HCT116, HT29, and LoVo, which exhibit different sensitivities to **5-FU** (*Bracht et al., 2010*). We observed similar results in all tested cell lines. While most compounds produced weak effects, we could determine $IC_{50}$ values for compounds **E5-E7** (*Figure 6—figure supplement 4*, *Supplementary file 6*).

A crucial issue remains the intracellular mechanism of action of these dimer destabilizers. Among the classical active-site inhibitors of hTS, **5-FU** forms a stable ternary complex with the MTHF cofactor and the hTS protein and induces high expression of hTS by stabilizing the dimeric assembly, thereby impairing both the protein feed-back regulation and the proteasomal degradation (*Chu et al., 1991*, *Berger et al., 2004*, *Peña et al., 2009*). On the other hand, **PMX**, another TS active-site inhibitor with a folate-analog structure, binds at the MTHF binding site and forms a reversible non-covalent ternary complex with dUMP (*Costi et al., 2005*; *Wilson et al., 2014*). Our dimer destabilizers inhibit recombinant hTS with a different mechanism of action, that is, by destabilizing its dimer. Thus, we expect them to act differently in cells too, and to produce a different modulation of the intracellular hTS levels with respect to **5-FU** and **PMX**. The results of these studies are shown in *Figure 7*.

## The dimer destabilizers decrease the hTS levels and promote cell death

To investigate the intracellular mechanism of action of compounds **E5** and **E7**, we first determined whether they induce apoptotic cell death both in cisplatin-sensitive A2780 cells and in cisplatin-resistant A2780/CP cells. As a control, we included cisplatin (cDDP) that, as expected, caused a higher rate of apoptotic cell death in sensitive than in resistant cells. Remarkably, **E7** induced apoptotic death in the two cell lines after a 48 hr treatment with a slightly higher efficiency than **PMX** (*Figure 6B and C*, , *Figure 7—figure supplement 1* , *Figure 7—figure supplement 2*).

Following treatment of both A2780 and A2780/CP cells with **E5** and **E7**, while we found no change in the levels of the DHFR protein, the level of the hTS protein decreased significantly in a dose-dependent manner (*Figure 7A*). It was low, nearly undetectable after 12–24 hr of exposure to **E7** in HT29 cell (*Figure 7—figure supplement 1* ) and undetectable after 24–36 hr of treatment in A2780/CP cells (*Figure 7—figure supplement 1*). In the same cell lines, we observed increased levels of hTS after treatment with both **PMX** and **5-FU** (*Figure 7B* ).

The ubiquitin-independent proteasomal degradation pathway is an important mechanism to destroy destabilized proteins in cells, and hTS is one of the substrates of this pathway (*Peña et al., 2009*). The observed decrease in hTS protein levels caused by **E5** and **E7** led us to hypothesize that it might be associated with a post-translational regulation mechanism, including proteasomal degradation. To test this hypothesis, we selected **E7**, our best cellular inhibitor, to determine the hTS half-life after blocking the de novo protein synthesis with the cycloheximide (CHX) inhibitor. As shown in *Figure 7C*, we found that the half-life of hTS was 3 hr in untreated cancer cells and that the degradation rate of the enzyme increased by 20% in **E7**–treated A2780 cell (*Figure 7D*).

To check whether the 26 S proteasome was involved in such an accelerated degradation, we added the proteasome inhibitor MG132 (10 µM, for 5 hrs) to **E7**-treated and untreated A2780 cells. Indeed, as shown in *Figure 7E*, MG132 restored the hTS levels in treated cells. These findings are opposite to the known effects on the hTS intracellular life exhibited by the reference inhibitor, **5-FU**. In fact, previous reports on other cell models have shown that **5-FU** causes a remarkable slowdown of the TS degradation rate because most of the protein is engaged in a ternary complex with the folate substrate and the inhibitor, being thus protected from the proteasome (*Berger et al., 2004*; *Peña et al., 2009*; *Wilson et al., 2014*; *Kitchens et al., 1999a*; *Kitchens et al., 1999b*).

To investigate whether engagement of **E5** and **E7** with hTS in cancer cells could affect the protein catalytic activity, their effect on endogenous TS activity was evaluated on cellular extracts from

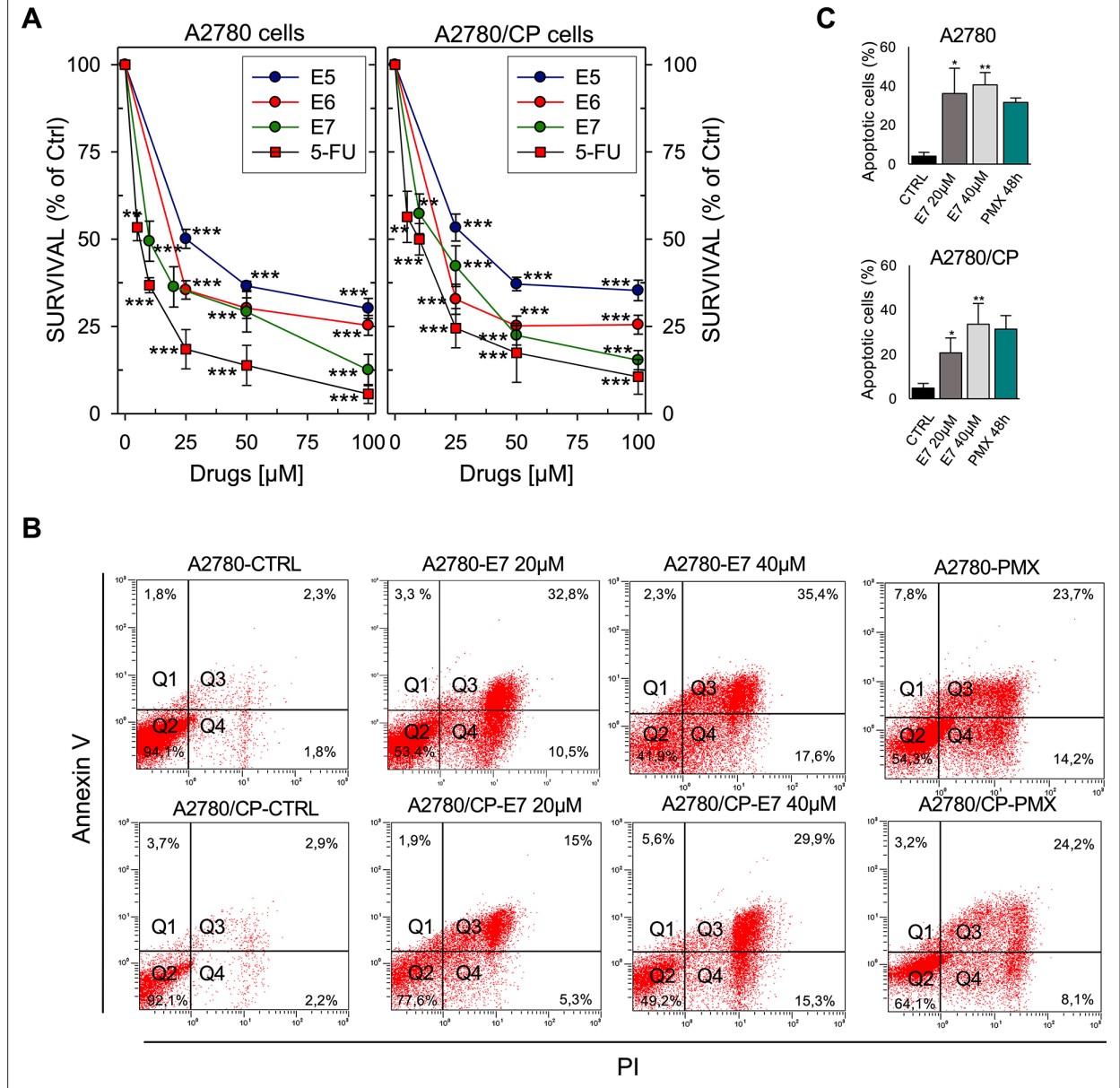

**Figure 6.** Effects of hTS inhibitors in human cancer cell lines. (**A**) Dose-response curves for E5–E7 and **5-FU** against A2780 and A2780/CP cell growth. (**B**) Flow cytometric analysis of apoptosis of A2780 and A2780/CP cells treated with E7 or PMX (Annexin V–/PI–: live cells, Annexin V+/PI–: early apoptotic cells, Annexin V+/PI+: late apoptotic cells, Annexin V–/PI+: necrotic cells). (***Figure 6—source data 1***). (**C**) Quantification of Annexin V-positive cells (Q2 + Q3) performed using the Annexin V/PI kit. Data indicate mean values and standard deviation of at least two biological repeats performed in duplicate. p Values were calculated with two-sided Student's *t*-test and ANOVA followed by the Tukey's multiple comparison. * p<0.05; ** p<0.01; *** p<0.001.

The online version of this article includes the following source data and figure supplement(s) for figure 6:

**Source data 1.** Quantification of Annexin V-positive cells of apoptosis of A2780 and A2780/CP cells treated with E7 or PMX.

**Figure supplement 1.** TS mRNA expression in the primary pancreatic cancer cells.

**Figure supplement 1—source data 1.** TS mRNA expression in the primary pancreatic cancer cells.

**Figure supplement 2.** Dose-response curve of Pancreatic cancer cells LPc167 growth inhibition for compounds E7, E5, and 5FU.

**Figure supplement 2—source data 1.** Pancreatic cancer cells LPc167 growth inhibition for compounds E7, E5, and 5FU.

**Figure supplement 3.** Ovarian cancer cell growth inhibition".

**Figure supplement 3—source data 1.** Ovarian cancer cell growth inhibition all compounds.

**Figure supplement 4.** Colorectal cancer cell survival percentages as the mean ± S.E.M. of three separate experiments performed in duplicate.

*Figure 6 continued on next page*

*Figure 6 continued*

**Figure supplement 4—source data 1.** Colon cancer cell inhibition E7 and E5.

untreated cells of four cell lines, namely, A2780, A2780/CP, 2008 and C13*. These extracts were incubated with the inhibitors at their IC$_{50}$ values for 60 min. In these conditions, the catalytic activity of hTS was systematically reduced by 25–30% in the cellular extracts of all cell lines (*Figure 7—figure supplement 1*).

This reduction of the hTS activity supports the hTS target engagement by the dimer destabilizers and links the observed effect with the intracellularly observed dimer to monomer equilibrium shift (*Figure 7F* in the next section). Based on the above data, we conclude that **E7** reduces the hTS levels in cancer cells by enhancing its proteasomal degradation. Because **E7** behaves as a disrupter of the dimeric recombinant protein, we anticipate that the promotion of its proteasomal degradation in cells is due to an increased fraction of more labile monomers (*Figure 7G*).

## E7 disrupts endogenous hTS dimers yielding monomers that are more rapidly degraded by the proteasome

We then explored how treatment with **E7** perturbed the intracellular hTS dimer/monomer equilibrium using a cross-linking strategy on endogenous hTS from A2780 cells (*Figure 7F*; *Shi et al., 2017*).These cells were treated with **E7** or with the vehicle (DMSO) for 3 hr, after which cross-linking was carried out by adding bis(sulfosuccinimidyl)suberate (BS$^3$) to the cell lysate. Next, the hTS dimers and monomers were detected via immunoblot after SDS-PAGE in reducing conditions. The treatment caused a significant decrease of the dimer/monomer ratio relative to the control (- 40 ± 10%, *Figure 7F*). We thus confirmed that **E7** disrupts the hTS dimers also in cells.

To confirm that the hTS monomer is less stable than the dimer in cells, we used a model system in which HCT116 cells ectopically express either WT hTS or the hTSF59A dimer interface mutant tagged with Myc-DDK (*Figure 7G*). While WT, with its 60–80 nM K$_d$ in buffer, is expected to be essentially dimeric at all physiological concentrations, the F59A variant, with a K$_d$ of $1.3 \times 10^{-4}$ M in buffer is, instead, expected to be essentially monomeric (*Salo-Ahen et al., 2015*). We analyzed the protein turnover of WT hTS and of the monomeric F59A variant, in the presence of the de novo protein synthesis inhibitor CHX. The expressions of the mRNAs of the two proteins showed no statistically significant differences (*Figure 7—figure supplement 3*).

As for the proteins, the F59A variant was already hardly discernible at time 0 (24 hrs after ectopic expression) while the dimeric WT hTS showed a turnover of about 8.5 hr. Thus, the monomeric hTSF59A variant is unstable and undergoes much faster turnover than the dimeric WT protein.

## In vivo anticancer activity

Based on their in vitro profile, we tested **E5** and **E7** in appropriate animal models. We first performed pharmacokinetic (PK) studies on healthy mice to determine whether the compounds would show suitable PK. Initial PK data were obtained using intravenous (i.v.) administration of **E5** and **E7** (*Supplementary file 6C*). **E7** showed the best PK profile (*Figure 8A*). Both after i.v. and oral administration it reached its t$_{max}$ value after about 5 min with plasma levels in the micromolar range, i.e. concentrations that had caused a cell-growth inhibition in vitro. With a half-life of 13.6 hr, a prolonged exposure of the target tissue to the drug is expected. **E7** was then selected to be tested for its antitumor effect in vivo. In general, for low molecular weight compounds, intraperitoneal (i.p.) and i.v. administrations result in comparable PK. Furthermore, because of the lipophilicity of the compound, a rapid transfer from the peritoneal cavity to the blood and back is expected. Since we preferred to analyze the antitumor activity of **E7** in a PDAC orthotopic model that recapitulates several human-tumor phenotypic characteristics, we chose an i.p. drug delivery with an intermittent schedule (*Figure 8B*). Such a relatively frequent schedule would have been too stressful for the mice via the i.v. route because of the damage to veins. The ability of the hTS dimer disrupter **E7** to inhibit tumor growth was evaluated on bioluminescent PDAC models genetically engineered to express Gaussia-luciferase (G-luc), simplifying the monitoring of the tumor volume. Five days after injection, primary pancreatic tumors were detectable in all mice (100% take rate, without surgery-related mortality), and G-luc activity, proportional to the number of cancer cells, increased with time (*Figure 8C*). This mean signal was

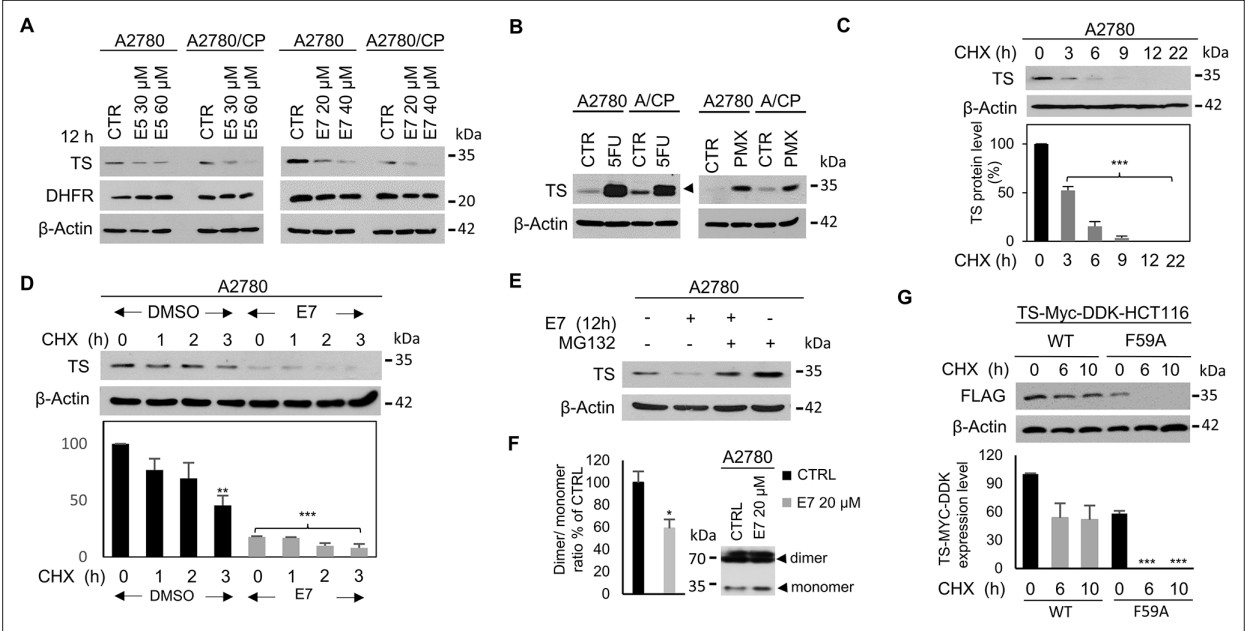

**Figure 7.** Dimer destabilizers effects on wild type and mutant hTS levels in human cancer cell lines. (**A**) Effects of E5 and E7 on hTS and DHFR protein levels in A2780 and A2780/CP cells. (**B**) Increase of hTS protein level in A2780 and A2780/CP cells following treatment with **5-FU** (5 µM, 72 hr) (ternary-complex, arrowhead) or PMX (5 µM, 48 hr). (**C**) hTS half-life determination in A2780 cells after treatment with CHX (90 µg/ml) for 0–22 hr. (**D**) Stability of hTS in A2780 cells treated with E7 for 12 hr, followed by treatment with CHX for the indicated times. (**E**) hTS levels in A2780 cells treated with E7 for 12 hr, then with 10 µM MG132 for 5 hr. (**F**) Effects of a 3 hr exposure to E7 on hTS dimer/monomer equilibrium in A2780 cells. (**G**) Half-life determination of exogenous hTS protein (anti-FLAG stain) in HCT116 cells transfected with TS-Myc-DDK (CHX, 0, 6, and 10 hr) or with the F59A mutant (CHX 0, 6, and 10 hr) tagged vector. Data indicate mean values and standard deviation of biological repeats performed in duplicate. p Values were calculated with two-sided Student's *t*-test and ANOVA followed by the Tukey's multiple comparison. * $p<0.05$; ** $p<0.01$; *** $p<0.001$.

The online version of this article includes the following source data and figure supplement(s) for figure 7:

**Source data 1.** Effects of E5 on hTS and DHFR protein levels in A2780 and A2780/CP cells.

**Source data 2.** Effects of E5 on hTS and DHFR protein levels in A2780 and A2780/CP cells_TS and DHFR.

**Source data 3.** Effects of E7 on hTS and DHFR protein levels in A2780 and A2780/CP cells.

**Source data 4.** Effects of E7 on hTS and DHFR protein levels in A2780 and A2780/CP cells_TS and DHFR.

**Source data 5.** hTS protein level in A2780 and A2780/CP cells treated with 5-FU and PMX (PMX and 5FU).

**Source data 6.** hTS protein level in A2780 and A2780/CP cells treated with PMX.

**Source data 7.** Stability of hTS in A2780 cells treated with E7 for 12 hr, followed by treatment with CHX for the indicated times.

**Source data 8.** Stability of hTS in A2780 cells treated with E7 for 12 hr, followed by treatment with CHX for the indicated times.

**Source data 9.** hTS levels in A2780 cells treated with E7 for 12 hr, with MG132 for 5 h.

**Source data 10.** hTS levels in A2780 cells treated with E7 for 12 hr, with MG132 for 5 hr.

**Source data 11.** Effects of a 3 hr exposure to E7 on hTS dimer/monomer equilibrium in A2780 cells.

**Source data 12.** Effect of Dimer destabilizers and hTS levels and half-life in human cancer cell lines (Figure C, D, G).

**Source data 13.** Half-life determination of exogenous hTS protein (anti-FLAG stain) in HCT116 cells transfected with TS-Myc-DDK (CHX) or with the F59A mutant (CHX).

**Source data 14.** Half-life determination of exogenous hTS protein (anti-FLAG stain) in HCT116 cells transfected with TS-Myc-DDK (CHX) or with the F59A mutant (CHX).

**Source data 15.** Effects of E5 on hTS and DHFR protein levels in A2780 and A2780/CP cells.

**Source data 16.** Effects of E5 on hTS and DHFR protein levels in A2780 and A2780/CP cells_TS and DHFR.

**Source data 17.** Effects of E7 on hTS and DHFR protein levels in A2780 and A2780/CP cells.

**Source data 18.** Effects of E7 on hTS and DHFR protein levels in A2780 and A2780/CP cells_TS and DHFR.

**Source data 19.** hTS protein level in A2780 and A2780/CP cells treated with 5-FU and PMX (PMX and 5FU).

**Source data 20.** hTS protein level in A2780 and A2780/CP cells treated with PMX.

*Figure 7 continued on next page*

*Figure 7 continued*

**Source data 21.** Stability of hTS in A2780 cells treated with E7 for 12 hr, followed by treatment with CHX for the indicated times.

**Source data 22.** Stability of hTS in A2780 cells treated with E7 for 12 hr, followed by treatment with CHX for the indicated times.

**Source data 23.** hTS levels in A2780 cells treated with E7 for 12 hr, with MG132 for 5 h.

**Source data 24.** hTS levels in A2780 cells treated with E7 for 12 hr, with MG132 for 5 hr.

**Source data 25.** Effects of a 3 hr exposure to E7 on hTS dimer/monomer equilibrium in A2780 cells.

**Source data 26.** Half-life determination of exogenous hTS protein (anti-FLAG stain) in HCT116 cells transfected with TS-Myc-DDK (CHX) or with the F59A mutant (CHX)_actin.

**Source data 27.** Half-life determination of exogenous hTS protein (anti-FLAG stain) in HCT116 cells transfected with TS-Myc-DDK (CHX) or with the F59A mutant (CHX).

**Figure supplement 1.** Effect of cisplatin on cell death, effect of dimer destabilizers on hTS activity and hTS protein levels in cancer cells.

**Figure supplement 1—source data 1.** Quantification of Annexin V-positive cells by flow cytometric analysis of apoptosis of A2780 and A2780/CP cells treated with cisplatin, E7 and PMX.

**Figure supplement 1—source data 2.** Effect of E7 on hTS protein levels in HT29 cells _actin.

**Figure supplement 1—source data 3.** Effect of E7 on hTS protein levels in HT29 cells.

**Figure supplement 1—source data 4.** Effect of E7 on hTS protein levels in HT29 cells _actin.

**Figure supplement 1—source data 5.** Effect of E7 on hTS protein levels in HT29 cells.

**Figure supplement 1—source data 6.** Effect of dimer destabilizers on hTS protein levels in A2780/CP cells, E7 and actin.

**Figure supplement 1—source data 7.** Effect of dimer destabilizers on hTS protein levels in A2780/CP cells, E7 and actin.

**Figure supplement 1—source data 8.** TS activity assay for A2780 and A2780/CP cell lines after treatment with E5 and E7.

**Figure supplement 2.** An example of gating strategy used for cell death analysis.

**Figure supplement 3.** Expression levels of total (exogenous plus endogenous) and exogenous hTS mRNAs 24 hr after transfection of HCT116 cells with wild-type hTS-Myc- DDK tagged or mutant hTS-Myc-DDK-F59A vectors.

significantly lower in mice treated with **E7** compared with control mice (46183±2015 vs 67905±6159 RLU/sec at day 25; p=0.001, two-sided t test). Moreover, starting from day 30, the group receiving **E7** had significantly lower G-luc intensity compared with mice receiving **5-FU** (i.e., at day 30,–70% and –48%, relative to control mice, respectively). Mice treated with **E7** had only a slight reduction of body weight (*Figure 8D*), similar or less evident than that shown by mice treated with **5-FU** and showed no other side effects. Tumor growth inhibition was reflected in a significantly longer survival of mice treated with **E7** as compared with control mice (*P*=0.03, *Figure 8E*) and **5-FU** (58 days, 43.3 days, and 50 days, respectively). To demonstrate that the therapeutic effect of **E7** is not cancer-model specific, $2 \times 10^6$ human ovarian cancer cells (A2780) were subcutaneously xenotransplanted into the flanks of 7-week-old Athymic Nude-Foxn1nu female mice (*Figure 8—figure supplement 1*).

Seven days after injection, as for the orthotopic model, the engrafted mice were randomly divided into three experimental groups and then treated with a similar drug schedule and dose. We found that also in this model, **E7** caused a significant reduction of tumor growth compared with control mice (596.1±167.7 vs 2319.9±289.7 mm$^3$ at day 6; 776.5±191.6 vs 3102.0±516.2 mm$^3$ at day 7; p<0.05, two-sided t test), which was confirmed by quantification of the area under curve and resulted in significantly longer survival (p=0.0111; 24 days *vs* 16 days; *Figure 8—figure supplement 1*).

To understand whether the antitumor activity of **E7** was associated with an increased anti-TS effect, we measured hTS mRNA (*Figure 8F*) and protein expression (*Figure 8G*) in tumor tissues from the PDAC orthotopic model. The mice treated with **5-FU** did not show a decrease in TS mRNA expression and showed a more intense staining for TS, in agreement with earlier studies in which treatment with **5-FU** had induced TS activity both in mice and patients (*Van der Wilt et al., 1992*; *Peters et al., 2002*). Conversely, **E7** treatment led to a reduced expression of TS mRNA, down to 40–50% of the reference level (*Figure 8F*) in complete agreement with the results obtained with model cancer cells (*Figure 8—figure supplement 2* ).

Immunohistochemistry staining showed that **E7** caused a decrease in the levels of hTS protein that was even more pronounced than **5-FU** (*Figure 8G*). The hTS protein levels measured with the histological score (H-score) was 48±5 a.u. for **E7** vs. 242±26 a.u. for **5-FU** (p<0.001, two-sided t test). Additional analyses showed that **E7** reduced the proliferation of the cancer cells, as detected by

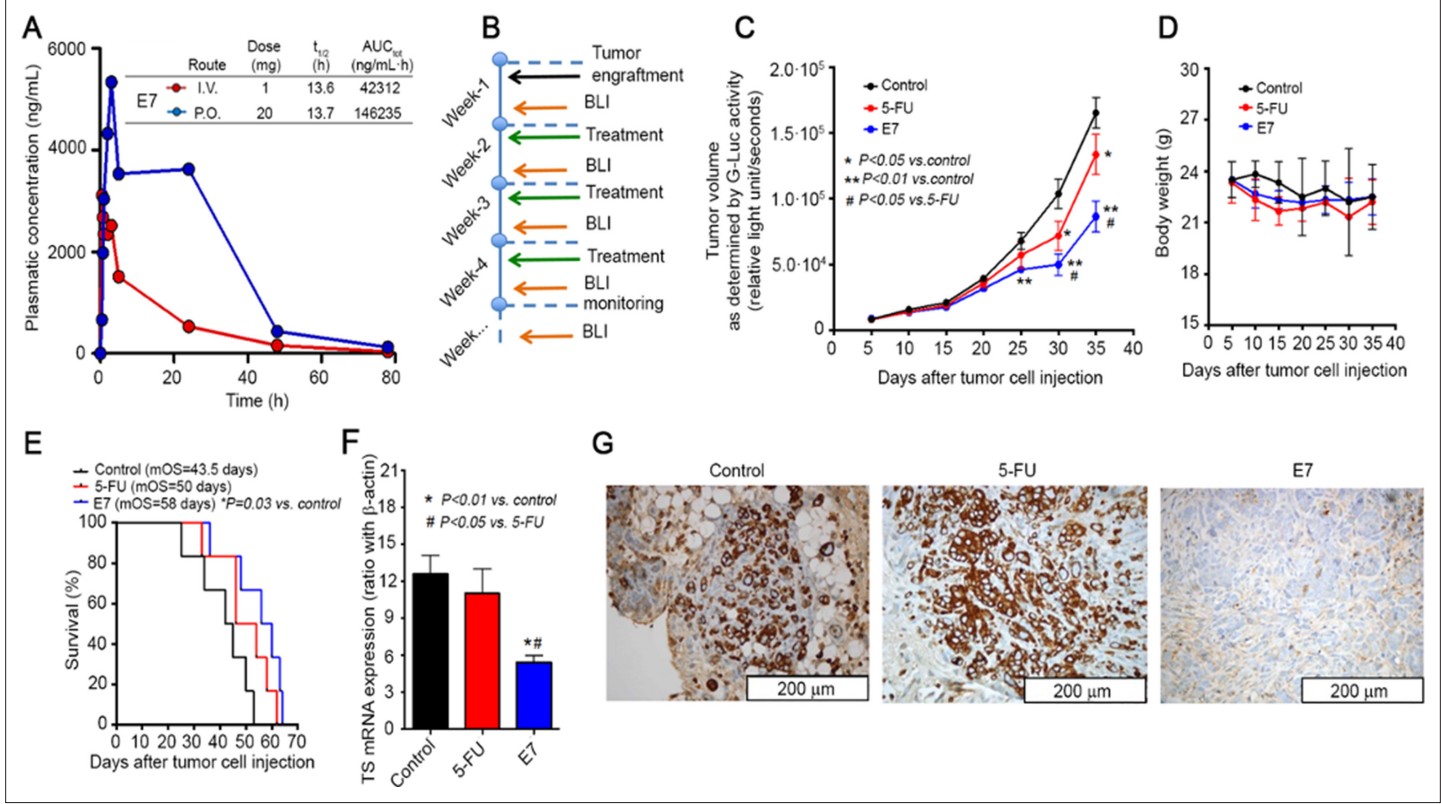

**Figure 8.** Effects of E7 in vivo pancreatic cancer model. (**A**) PK data on healthy mice for E7. (**B**) Set-up of in vivo experiments, using mice injected orthotopically with PDAC-2-primary cells, and monitored every 5 days by bioluminescence imaging (BLI). (**C**) Effects of E7 and **5-FU** on tumor growth, as detected in blood samples with the G-luc (proportional to the number of cancer cells, carrying G-luc). Points indicate mean values obtained from the analysis of the six mice in each group. Bars indicate standard deviation. p Values were calculated with two-sided Student's *t*-test and ANOVA followed by the Tukey's multiple comparison. (**D**) Effects of E7 and **5-FU** on the weight of the mice, demonstrating that tumor shrinkage induced by these treatments was not accompanied by severe toxicity. Points indicate mean values obtained from the analysis of the six mice in each group. (**E**) Survival curves in the groups of mice treated with E7 and **5-FU**. Median overall survival (mos) is reported. Statistically significant differences were determined by two-sided log-rank test. (**F**) Quantitative PCR results (calculated with the standard curve method, using the ratio with the housekeeping gene β-actin) showing the reduced expression of hTS mRNA in lysates from frozen tissues from PDAC-2 mice treated with E7 (24 hr before their sacrifice) compared with untreated control mice. Columns and bars indicate mean values and standard deviation. p Values were calculated with two-sided Student t test and ANOVA followed by the Tukey's multiple comparison. (**G**) Representative immunohistochemical images showing a weak staining for hTS in the tissues from mice treated with E7 compared to the strong staining in **5-FU**-treated and control mice. (*Figure 8—source data 1*).

The online version of this article includes the following source data and figure supplement(s) for figure 8:

**Source data 1.** Orthotopic pancreatic cancer in vivo.

**Figure supplement 1.** In vivo effects of E7 on Athymic Nude-Foxn1nu female mice.

**Figure supplement 1—source data 1.** In vivo effects of E7 on Athymic Nude-Foxn1nu female mice.

**Figure supplement 2.** Real time PCR (qPCR) experiments.

**Figure supplement 2—source data 1.** Real-time PCR (qPCR).

**Figure supplement 3.** Representative immunohistochemical images showing extensive caspase- 3 activation after treatment with E7, which was also able to significantly reduced cell proliferation (Ki67 staining).

the reduced number of Ki67-positive cells, while increasing apoptotic cell death, as assessed by the staining of cleaved caspase-3 (*Figure 8—figure supplement 3*).

Notably, these effects were markedly stronger in tumors treated with **E7** compared to tumors treated with **5-FU**. These results provide clear evidence of the higher anticancer efficacy of **E7** compared to **5-FU** in the cancer animal models employed; moreover, they support our hypotheses of the different mechanisms of action of these compounds in cells and in tissues.

## Discussion

Currently, several first-line chemotherapies commonly comprise cytotoxic agents such as 5-FU and its pro-drug derivatives to uproot or control various solid tumors including colorectal cancer.

5-FU induces cytotoxicity either by interfering with essential biosynthetic activities, namely, by either inhibiting the action of thymidylate synthase (TS) or misincorporating its metabolites into RNA and DNA (Sethy C and Kundu CN, 2021). hTS is an obligate, stable homodimer that plays a crucial role in cell proliferation. As an enzyme, it provides the sole de-novo pathway to deoxythymidylate (dTMP) synthesis in human cells. Additionally, by interacting with its own and other mRNAs, this protein regulates its own levels and those of several other proteins involved in apoptotic and proliferation processes. Its inhibition has usually been achieved with compounds that bind at the protein active-site, competing with either the dUMP substrate, such as 5-fluorodeoxyuridine 5'-monophosphate (FdUMP), or with the folate cofactor, such as raltitrexed (RTX) and pemetrexed (PMX).

Despite several advantages, the clinical application of 5-FU is limited by the development of drug resistance after chemotherapy. The major mechanism responsible for 5-FU resistance is controlled by key enzymes such as thymidylate synthase (TS), dihydropyrimidine dehydrogenase (DPD) and thymidine phosphorylase (TP). Numerous reports have shown that increased TS expression is a result of the common mechanisms of 5-FU resistance (*Longley et al., 2003*; Iijchi K et al, 2014; Vodenkova S et al, 2020; *Ahn et al., 2015*; Aschele C et al, 2002). Because response to TS-directed chemotherapy critically depends upon the enzyme concentration, such an over-induction has been identified as a barrier to successful therapeutic outcomes. Hence the need to design and develop drugs that, having different mechanisms of action, can prevent or delay the onset of resistance.

In the present work we developed small molecules able to reduce the concentration of hTS dimers by shifting its dimer-monomer equilibrium toward the inactive monomer. As a consequence, they inhibit the activity of the enzyme and produce, as a crucial consequence, its enhanced proteasomal degradation and down-regulation in cells.

Unlike 5-FU, these compounds do not inhibit TS by forming the unproductive enzyme dimer-substrate-inhibitor complex. Formation of this stable complex leads to prolonged engagement of the enzyme by the drug and, eventually, to thymineless cell death. However, it also induces translational derepression of TS-mRNA and results in the enzyme up-regulation, accounting for the occurrence of 5-FU resistance (*Chu and Allegra, 1996*). Thus, the proposed dimer destabilizers are believed to avoid the onset of drug-resistance based on this mechanism. Additionally, enzyme stabilization has been indicated as an important primary mechanism of TS induction by fluoropyrimidines (*Kitchens et al., 1999a*). TS turnover is carried out in an ubiquitin-independent manner (*Forsthoefel et al., 2004*; *Peña et al., 2006*) and FdUMP binding to the TS protein leads to dramatic conformational changes which account for a stabilization of the enzyme and an increase in its half-life and intracellular levels (*Kitchens et al., 1999b*; *Marverti et al., 2009*). Again, use of the dimer destabilizers characterized by a different mechanism of action with respect to 5-FU, results in a decrease of the TS half-life by accelerating its proteasomal degradation, thus also counteracting this mechanism of TS over-expression and drug resistance.

Indeed, reduction of the intracellular levels of the TS enzyme and a marked decrease of both in vitro cancer cell growth and in vivo tumor growth, together with a reduced expression of hTS mRNA in lysates from treated frozen tissues and cells, are the most promising results obtained with these dimer destabilizers. Therefore, they represent new, promising tools to fight drug resistance associated with high TS levels in different tumor types such as colorectal, ovarian, and pancreatic cancers.

In conclusion, we have proposed a genuinely new class of inhibitors of a critical anti-cancer drug target, hTS. They act as destabilizers of the active obligate dimer, resulting in the inactive monomers. With their unprecedented mechanism of inhibition, compounds such as **E7** break the long-established link between inhibition and enhanced expression of this essential enzyme, a link that is typical of classical hTS active-site inhibitors, such as 5-FU and PMX. The superior in vivo behavior of **E7** relative to **5-FU**, despite its similar capacity to inhibit cancer cell growth, confirms the high potential of this dimer disrupter for further development. Our experiments suggest that it may overcome drug resistance in cancer cells that directly or indirectly increase the hTS levels. By doing so, these compounds, once combined with classical anti-TS drugs in suitably designed anticancer therapeutic schemes, may help avoiding or delaying the onset of anti-hTS drug resistance associated with over-expression of the enzyme, thus maintaining or restoring cancer-cell drug sensitivity.

## Materials and methods

### Computational protein-pocket identification for tethering

To detect possible pockets for the tethering experiments at the hTS monomer-monomer interface, we examined the crystal structures of both the active (PDB-ID: 1HVY) (*Phan et al., 2001*) and inactive (PDB-ID: 1YPV) (*Lovelace et al., 2005*) hTS conformation. SYBYL 7.3 (Tripos Inc, St. Louis, MO) and PyMOL (version 1.8) were used for structure preparation and visualization of molecular structures. PASS (*Brady and Stouten, 2000*), SiteID (Tripos Inc, St. Louis, MO) and CASTp (*Dundas et al., 2006*) were used to identify the pockets at the monomer interface as well as the surface crevices of dimeric hTS that extend into the interface. The GRID software (*Case et al., 2005*; *Goodford, 1985*; *Wade and Goodford, 1993*) was used to calculate molecular interaction fields of different probes. The KYG server (*Kim et al., 2006*) was used to predict the interface residues that could bind to RNA to avoid selecting interface pockets that would overlap with the predicted RNA binding site(s). We searched possible transient interface pockets by molecular dynamics (MD) simulations with the AMBER 8 simulation package (*Case et al., 2005*). We used the ff03 force field (*Duan et al., 2003*) 7-ns long simulations were performed for the monomeric (active and inactive) hTS both in implicit (NPSA) (*Wang and Wade, 2003*) and explicit (TIP3P) (*Jorgensen et al., 1983*) solvent. We performed shorter (4 ns) simulations for dimeric hTS, both active, with and without the dUMP substrate (from 1HVY), and inactive, with or without a phosphate ion in the active site (from 1YPV).

### Detailed results description

The MD studies showed that, in general, the hTS structures and the energy of the simulations remained stable throughout the simulation time. The flexible regions (loops) of the inactive hTS were more mobile than in the active hTS structure and especially the modeled small domain was moving relatively much compared with the other loops. The dUMP-bound active dimeric hTS exhibited the smallest residue fluctuations. The identified interface pockets are reported in *Figure 1—figure supplement 1* and *Supplementary file 1*. We examined the interface pockets of the snapshot structures from the MD trajectories at every sharp nanosecond (*Supplementary file 1B*). In all simulations, new pockets were formed while some disappeared, and others reappeared. *Supplementary file 1B* shows the pockets along the active monomer simulation in explicit water (without dUMP). Finally, the Y202 pocket was selected as the most promising target for tethering experiments since this pocket does not appear in the hTS dimer, it is not predicted to be part of the mRNA region and it can interact with various functional groups. It is also of reasonable size and there are several smaller crevices around it that can fuse with it, as observed from the simulations.

### Details on methods used in MD studies

The proteins and ligand preparation were carried out with the SYBYL 7.3 (*Tripos Inc, St. Louis, MO*) Biopolymer module; modified cysteines were reconstructed as normal cysteine residues, hydrogen atoms and missing side chains were added. The dimeric form of the inactive hTS was built with the Protein Quaternary Structure file server (*Henrick and Thornton, 1998*) and the missing small domain (residues 107–128) was modeled according to the 1HVY structure. The hydrogen of the sugar hydroxyl group in dUMP was directed toward the neighboring H256. The Antechamber tool (*Wang et al., 2006*) of AMBER 8 was used to create the ligand parameters for the MD simulations. The ligand partial charges were modified manually: for the phosphate ion, the charge of phosphorus (P) atom was set to 1.216e (NPSA) or 1.4e (TIP3P) and the oxygen (O) atoms to –0.55e (NPSA) or –1.1e (TIP3P). The total charge of the phosphate ion in the NPSA model was –1e, and in the explicit water –3e. For dUMP, the charges were partly taken from the all_nucleic02.lib of AMBER 8: the uracil moiety from the RU unit and the sugar moiety from the DT3 unit. The phosphate group was set to have a total charge of –2e. Firstly, the systems were energy minimized, heated and equilibrated. For the implicit water (NPSA) simulations: 30 ps (10 K -->100 K), 20 ps (100 K -->200 K), 20 ps (200 K -->300 K) and then 400 steps of minimization were carried out. For the explicit water simulations, an octahedral box extending 10 Å from the protein was filled with TIP3P water (including the crystallographic water sites) and neutralizing Na +ions. Energy minimization was performed in six steps, decreasing the constraints on the protein stepwise from 5 to 0 kcal/molÅ². Each step was carried out for a maximum of 200 iterations. Equilibration: (i) restrained solute (5 kcal/mol restraint force constant, 10 K -->100 K, 10 ps, constant volume, Langevin dynamics); (ii) free solute (10 K -->100 K, 10 ps, constant volume, Langevin

dynamics); (iii) free solute (100 K -->200 K, 10 ps, constant volume, Langevin dynamics); (iv) free solute (200 K -->300 K, 10 ps, constant volume, Langevin dynamics); (v) free solute (300 K, 10 ps, constant volume and temperature, Langevin dynamics); (vi) free solute (300 K, 10 ps, constant pressure and temperature). The production simulations were performed at a constant temperature of 300 K and a pressure of 1 bar with Berendsen coupling constants of 5.0 and 2.0 ps, respectively (*Berendsen et al., 1984*). Periodic boundary conditions, particle-mesh Ewald electrostatics (*Essmann et al., 1995*) and a cut-off of 9 Å for non-bonded interactions were used. A time step of 2 fs was applied together with the SHAKE algorithm (*Ryckaert et al., 1977*) to constrain the bonds to hydrogen atoms.

## Site-directed mutagenesis, protein purification

The mutants Y202C and C195S were generated by site directed mutagenesis using the plasmid hTS-pQE80l as template, as already reported (*Salo-Ahen et al., 2015*) (forward primer Y202C: CATG CGCTGTGTCAATTTTGCGTAGTCAACAGTGAACTG; forward primer C195S: *CTCATGGCTCTT CCGCCAAGCCATGCGCTGTGTCAATTT*). The double mutant C195S-Y202C was generated using the plasmid hTS-Y202C pQE80l as template. Wt-hTS and mutants were produced and purified according to established protocols (*Salo-Ahen et al., 2015*).

## X-ray crystallography

Crystals of the mutantshTS-Y202C and hTS-C195S-Y202Cwere obtained under high salt conditions by established protocols (*Cardinale et al., 2011*; *Salo-Ahen et al., 2015*), and grew in 4 months and in about 1 year, respectively. X-ray diffraction data were collected at ESRF (Grenoble, France), processed with MOSFLM 7.0.4 (*Leslie, 2006*) and scaled with SCALA (*Evans, 2006*) from the CCP4 suite (*Winn et al., 2011*). Structures were solved by molecular replacement using MOLREP (*Vagin and Teplyakov, 1997*) (search modeleither inactive or active wt-hTS, PDB-IDs 3N5G (*Cardinale et al., 2011*) and 1HVY (*Phan et al., 2001*), respectively) and refined with REFMAC5 (*Winn et al., 2003*) and PHENIX (*Adams et al., 2011*) with TLS parameterization (*Murshudov et al., 1997*). Water molecules were added using ARP/wARP (*Langer et al., 2008*) and Coot (*Emsley et al., 2010*) was used for manual rebuilding and modeling. The model stereochemical quality was assessed using Coot and PROCHECK (*Laskowski et al., 1993*). Final coordinates and structure factors were deposited in the Protein Data Bank under the accession codes 4O1U (hTS Y202C) and 4O1X (hTS C195S-Y202C). Data collection and refinement statistics are reported in *Supplementary file 2*. Figures were generated using CCP4MG (*McNicholas et al., 2011*).

## X-ray crystal structure determination of hTS-Y202C and hTS C195S-Y202C

The hTS mutants Y202C and C195S-Y202C were crystallized under the same high salt conditions previously reported for the WT enzyme. The hTS-Y202C crystallized in the space group P3$_1$ with an enzyme dimer in the inactive conformation in the asymmetric unit (ASU). The structure of hTS-Y202C closely resembles that of the native enzyme in the inactive conformation (r.m.s.d. upon Cα-atom matching of 0.4 Å), explaining the similar biochemical profiles of the two proteins. The hTS C195S-Y202C crystallized in the space group P2$_1$2$_1$2$_1$with four independent subunits in the ASU, all in the active conformation. The hTS-C195S-Y202C structure is the first example of hTS crystallized in the active conformation under high salt conditions. Since both mutants crystallized under the same conditions, the shift of hTS towards the active conformation seems due to the active site C195S mutation. A possible explanation is that the clustering of the four residues C180 and C195 from the two dimer subunits stabilizes the inactive conformation of hTS, thus the C195S mutation prevents this occurrence. In both structures, the Fourier difference map clearly shows the replacement of the native Y202 by cysteine. Moreover, a positive extra electron density is found extending beyond the cysteine sulfur atom strongly indicating its reaction with β-mercaptoethanol (BME) present in the crystallization solution. Thus, C202 is chemically modified to S,S-(2-hydroxyethyl)thiocysteine (CME) (*Figure 2—figure supplement 1a, b and c*). The other cysteines are modified to either CME or S-methyl-thio-cysteine (SCH), apart for C210 which is in the reduced, unmodified form. The two mutants, Y202C andC195S-Y202C, have essentially the same behavior as regards thiol reactivity. In the A subunit of hTS-Y202C, CME202 entails water-mediated interactions with D173, R175, K47', T55', and D254' (*Figure 2—figure supplement 1d*). In subunit B, CME202 is H-bonded to K47' and forms water-mediated interactions with P172, D174, and

V203 (*Figure 2—figure supplement 1e*). Analogous intra- and inter-subunit interactions of CME202 are also observed in the structure of the double mutant. Thus, the structure of hTS C195S-Y202C validates the choice of Y202 as tethering mutation site, showing that this residue is prone to react with thiol compounds independently of the enzyme conformation.C202 is located in a dimer inter-face pocket considered important for protein-protein interactions. The CME modification, observed in both structures, indicates that C202 is exploitable for tethering with reactive molecules aimed at disrupting the hTS dimer interface (*Figure 2—figure supplement 1f and g*). In both variants, the loop K47-T55 of the facing subunit is displaced with respect to the structure of wt-hTS, increasing the intersubunit distances by 1.0–2.5 Å. As already suggested by the structure of wt-hTS in complex with the LR peptide (*Cardinale et al., 2011*), small-to-medium-sized molecules, such as BME or the LR octapeptide, can access the dimer interface of the enzyme. Thus, the structural evidence indicates that the hTS dimer undergoes either monomer/dimer interconversion or 'breathing' movements that allow the binding at the dimer interface of exogenous molecules, such as disulfide fragments.

Several attempts were made to obtain the co-crystal structure of the monomeric hTS in complex with C compounds. All trials failed, always yielding the uncomplexed dimeric hTS structure.

## Design of a disulfide-compound library (A1-A55)

Version 8 of the ZINC archive (*Irwin and Shoichet, 2005*) was screened to recover compounds containing a disulfide function. A preliminary search for S-S bonds was performed using the SMILES notation, finding 1066 molecules. Compounds with the disulfide bond within a cyclic structure (167/1066), and those with a disulfanyl group (SSH, 16/1066) or an oxidized sulphur atom (211/1066) were excluded. The remaining disulfides were subjected to a fragment-likeness analysis: molecular descriptors were calculated with VolSurf+ (*Cruciani et al., 2000*). LogP and molecular weight (MW) were used to filter out very large and flexible compounds, as well as too hydrophilic or too hydrophobic ones. Finally, multivariate statistical methods such as the Principal Component Analysis (PCA) were used for the final selection of 55 disulfides. The disulfide library is reported in *Supplementary file 3A*.

## Design of a library of commercially available compounds (B1-B26) and docking of C3-C5

A dataset of 331,600 compounds, commercially available from Specs, was downloaded from the ZINC website (https://www.zinc.org; ZINC version 8) (*Irwin and Shoichet, 2005*). We retained 242,590 compounds with a MW between 300 and 500 amu. Then we used SMARTS open babel (*O'Boyle et al., 2011*), (http://openbabel.org/wiki/Main_Page) to search for the presence of specific fragments (*Figure 3*): 97% of compounds were filtered out. Then, a library of 5774 compounds was screened against WT hTS (1HVY) focusing on the subunit-subunit interface close to Y202. We used FLAP (*Cross et al., 2010*) to build two receptor-based pharmacophore models, to screen the library versus these receptor-models, and to dock the most interesting candidates into the binding pockets. Each molecule was subjected to a conformational analysis and the Molecular Interaction Fields (MIF) were generated with the GRID probes DRY, O, N1, and H. Out of compounds with the highest overall scores (>0.60 for both receptor-based pharmacophore models), we selected a subset of 26 compounds, available for purchase at the time of study, and displaying reasonable structural diversity (*Supplementary file 4A*). Compounds **C3**, **C4** and **C5** were docked against WT hTS (1HVY) focusing on the subunit-subunit interface close to Y202, using Autodock, version 4.2, software (*Morris et al., 1998*).

## Mass Spectrometry (MS) studies

Matrix-Assisted Laser Desorption/Ionization (MALDI) - Time of Flight (TOF) (Voyager-Pro MALDI-TOF (Applied Biosystems)) was chosen for a medium-throughput screening of the disulphide-compound library (**A1-A55**). The results obtained were validated using an Electron Spray Ionization Quadrupole - Time of Flight (ESI-QTOF) MS (Quadrupole - Time of Flight (ESI-QTOF) mass spectrometer (Agilent Technologies Inc)). Analyzing our pools with both platforms allowed us to exclude any ionization-related issue in the ligand selection process. The screening approach was an iterative selection of the best ligands in a competitive environment obtained by arranging the ligands in pools properly chosen to avoid ambiguous assignments. Successive screening steps were performed in which the selected ligands from the previous screening were subjected to further screening using a differently

constituted pool. For the disulphide-compound library screening up to 3 ligands were pooled and mixed a hTS-C195S-Y202C solution in a buffer containing 1 mM 2-mercaptoethanol and 20 mM potassium phosphate (pH 7.5). The reaction mixture was incubated at room temperature for 1 hr and then purified on a C4 ZipTip (Millipore) to remove salts and unbound ligands. Details of the methods used for MS disulphide-compound library screening are reported in *Appendix 1*. To identify the binding site of these molecules, the protein-ligand complexes underwent enzymatic digestion in a bottom-up proteomic analysis using both ionization techniques for result evaluation. Details of the methods used are reported in *Appendix 1*.

## Fluorescence experiments

To investigate the dimer/monomer equilibrium modulation by the selected compounds, we employed a previously developed FRET assay in which hTS was tagged with fluorescein (F, excitation energy donor) and tetramethylrhodamine (T, acceptor) probes as described in *Genovese et al., 2010*. Tagging yielded samples with spectrophotometrically determined F:T:protein-dimer concentration ratios near the expected 1:1:1 values. Absorption spectra were measured on a Varian Cary 100 UV-vis spectrophotometer. Fluorescence spectra were measured on a Spex-JobinYvon Fluoromax2 spectrofluorometer. Usually, samples with spectrophotometrically determined protein and probe concentrations in the range 50–490 nM in phosphate buffer at pH 7.5 were excited at 450 nm, a wavelength at which the F/T excitation ratio is very high and T emission is almost negligible. All measurements were performed at 20 ± 3 °C. The value of the observed FRET efficiency, $\Phi_{FRET}$, was determined from the emission intensities at 580 nm, $I_{580}$, the maximum of sensitized T emission, and at 520 nm, the maximum of residual F emission, $I_F$ (*Genovese et al., 2010*; *Figure 5A*). In the unperturbed dimer such an efficiency is near 1 because a distance that is roughly half the critical Förster distance for the F/T pair separates the C43 and C43' residues of the two subunits (25 Å). On the other hand, it is zero when the protein is fully dissociated. Developed on a spectrofluorometer, the assay has later been adapted to a medium-throughput format. In short, 1 or 2 µL of a 10 mM DMSO solution of each inhibitor was distributed in a 96-well multiplate to a final concentration of 10 or 20 µM together with 300 nM hTS properly derivatized with the two fluorescent probes (at this concentration, in the absence of inhibitors, the more than 95% of the protein is in the dimer form). Control wells contained only hTS, either alone or with 2 µL of DMSO. Each sample was assayed in duplicate. The Tecan GeniusPRO equipment was employed to read the fluorescence emission from each well at both 535 nM (fluorescein signal) and 590 nm (fluorescein and tetramethylrhodamine signals) following excitation at 470 nm. From the $I_{590}/I_{535}$ ratio, the FRET efficiency in each well and its difference with respect to the control wells were determined.

## hTS inhibition assay and IC$_{50}$ determination

All compounds were tested against the hTS enzyme and IC$_{50}$ was measured spectrophotometrically using Spectramax 190 UV/Vis spectrophotometer, 96 wells multiplate reader. The inhibition reactions were conducted as reported (*Salo-Ahen et al., 2015*). The compounds were all tested in two different modes: without incubation and after 60' incubation time selected optimization studies. The reaction was followed at $\lambda$ =340 nm for 180 s in triplicates. Each inhibitor was firstly dissolved in DMSO in a 10 mM stock solution. At least eight concentrations were evaluated considering the compounds solubility (1–100 uM) at 37 °C. The reaction mixture included the following reagents solutions (a-e) added in the proper order: a, TES buffer 50% v/v (TES/water) (where TES = 2-[[1,3-dihydroxy-2-(hydroxymethyl)propan-2-yl]amino]ethane sulfonic acid), 100mM; MgCl$_2$ 50 mM; Ethylenediaminetetraacetic acid (EDTA) 2 mM; β-mercapto ethanol-(BME) 150 mM; pH 7.4; b, hTS enzyme (0.37 uM); c, the inhibitor at the appropriate concentration; d. MTHF cofactor (55 uM). e. dUMP is added as last reagent to start the reaction. For incubation studies, solutions a, b and c were added to each well. After a 1 hour incubation, solution d and e were added, and the reaction monitoring started. For both experimental conditions, the final DMSO concentration into the experimental cuvette (for each assayed concentration) was less than 1% v/v to remove a potential DMSO interfering inhibition effect on the catalytic activity of the target enzyme protein. Results were analyzed according to the standard double reciprocal or Dixon plots. Additionally, IC$_{50}$ values were obtained from nonlinear regression analysis of the initial rate data. All results showed an experimental standard error within the 20%. All the IC$_{50}$ values are collected in ***Supplementary file 4***.

## Enantioseparation and racemization of E7

The enantiomeric separation and biological characterization of E7 are reported in Appendix 3.

## Co-elution of compound C3 with the hTS dimer and monomer with anionic exchange Chromatography

hTS monomer and dimer were separated according to their isoelectric point with preparative Anion Exchange Chromatography (AEX) on a Q HP 16/10 column, Cytiva. 300 µL of a 50 µM hTS solution were injected onto the column and eluted at 2 mL/min with a 70 mL gradient of 0–1 M NaCl in 30 mM Tris-HCl, pH = 8.2. Two main peaks emerged from the 280 nm UV chromatograms of hTS only, that were attributed to protein's monomer and dimer fraction. Indeed, when hTS was incubated with 250 µM dUMP and RTX prior to injection to induce dimer formation, the second UV peak was suppressed as expected. hTS was incubated with different **C3** concentrations (4–80 µM) prior to FPLC injection to measure its ability to disrupt hTS dimer. The UV-Vis spectrum of each fraction was collected with a Multiskan GO multiplate reader (Thermo Fisher) to confirm hTS-**C3** co-elution by measuring its absorbance at 350 nm. Also, all FPLC collected fractions from 4 µM experiment were treated to precipitate proteins, and the amount of **C3** was measured by UHPLC tandem HRMS (Orbitrap Q Exactive, Thermo Fisher) (*Supplementary file 5*). **C3** was quantified in SIM mode after area interpolation into a calibration curve, revealing an excess of compound co-eluting with hTS monomer with respect to dimer. Also, hTS identity and activity of the two peaks was validated with a spectrophotometric kinetic specific assay.

## Analysis of combined equilibria and fitting to FRET data and Analysis of a dissociative inhibition mechanism and fitting to data are described in *Appendix 4*

The availability of kinetic inhibition data and FRET study results on compounds **C2-C4** allowed a detailed mechanistic analysis to test the suitability of the proposed mechanism to fit the experimental inhibition results.

## Cellular in vitro experiments

### Fluorescence microscopy

HT-29 cells were plated at $4 \times 10^4/cm^2$ in IBIDI 15 µ-Slide 8 Well plates for 24 hr and maintained at 37 °C, 5% $CO_2$. Then they were transfected with 500 ng/well of either TS-TC or WT TS using the Lipofectamine 3000 transfection Kit (Invitrogen) and following the manufacturer's recommendations. Twenty-four hr later, the medium was removed and 50 µM **E5-FITC** was added to the cells for 6 hours. Subsequently, the compound was removed, cells were washed with Opti-MEM reduced Serum Medium (Gibco), stained with the TC-ReAsh II in-cell Tetracysteine Tag Detection Kit (Life Technologies Corporation) according to the manufacturer's recommendations for 60 minutes. Then, the cells were washed three times with Opti-MEM, twice with BAL and placed in Opti-MEM until visualization. The image acquisition was done on a Leica SP8 confocal microscope. Subsequent FRET analysis was performed using PixFRET (Version 1.5.0) within ImageJ (Version 1.52e). Each dot represents the total FRET area of an individual field with at least 10 cells. The statistical analysis was performed using unpaired two tailed t-test (p-value >0.0001) using Prism 8 for windows (version 3.1.1).

## Experiments on cells

### Cell lines

The human ovarian cancer cell lines IGROV-1, TOV112D, 2008, C13*, A2780 and A2780/CP were grown as monolayers in RPMI 1640 medium. The C13* and A2780-CP human ovarian carcinoma cell lines are 9- to 15-fold resistant to cisplatin (cDDP) and derived from the parent 2008 and A2780 cell lines (*Marverti et al., 2013*; *Beaufort et al., 2014*). The HT29, HCT116, and LoVo cell lines were cultured in DMEM medium (Euroclone, Devon, UK). Cell culture media were supplemented with 10% heat-inactivated fetal bovine serum (Euroclone) and 1% Pen/Strep (Euroclone). The PDAC primary cell cultures were established from patients undergoing pancreatic duodenectomy, as described previously (*Giovannetti et al., 2014*). All experiments in the present study utilized cells collected during passages 5–8. Cultures were equilibrated with humidified 5% CO2 in air at 37 °C. All studies were

performed in Mycoplasma negative cells, as routinely determined with the Myco Alert Mycoplasma detection kit (Lonza, Walkersville, MD, USA).

## Cell density

Cell growth was determined using a modified crystal violet assay (*Marverti et al., 2009*). After 72 hr treatment with **E5**, **E7,** and **5FU**, the cell monolayer was fixed and stained with 0.2% crystal violet solution in 20% methanol for at least 30 min. After washing to remove excess dye, the incorporated dye, proportional to the cell number, was solubilized in acidified isopropanol (1 N HCl: 2-propanol, 1:10) and determined spectrophotometrically at 540 nm. The percentage of cytotoxicity was calculated by comparing the absorbance of cultures exposed to the drug with un-exposed (control) cultures. The PDAC cell growth was evaluated using the sulforhodamine B assay after exposure for 72 hr (*Giovannetti et al., 2014*).

## Half-life determination using cycloheximide

A2780 cells were analysed for hTS protein half-life using cycloheximide (90 µg/mL; for 0, 3, 6, 9, 12, and 22 hrs).

## Treatment of cancer cells with cycloheximide

Cells were treated with **E5** or **E7** at concentrations corresponding to their $IC_{50}$ values or $2xIC_{50}$ and harvested at different times (0, 1, 2, and 3 hrs) in the presence of medium containing 90 µg/ml of cycloheximide. The A2780 cells were treated with **E7** and then with a proteasome inhibitor (MG132 10 µM) for 5 hr.

## HCT116 cell transfection and cycloheximide treatment

HCT116 cells were transfected by the reverse transfection, using MirusIT-X2 reagent. The reverse transfection increases the % of transfection and makes it homogeneous. The complexes were made with a ratio of 1:3 between DNA and reagent (2.5 µg of TS-MYC-DDK-WT or TS-MYC-DDK-F59A and 7.5 µl of the reagent). The complexes were incubated 20 min at RT and in the meantime the cells were counted to be seeded (500.000 cells/well of a 6-well). After 24 hr from transfection the cells were treated with cycloheximide (CHX, 90 µg/ml) for 0, 6, and 10 hr to evaluate the hTS degradation.

## Cellular TS catalytic assay

Extracts from exponentially growing cells were used for the TS catalytic assay, by measuring the amounts of $^3$H released during the TS-catalyzed conversion of [5-$^3$H]dUMP to dTMP, as previously reported (*Marverti et al., 2009*). Briefly, the enzyme and 650 µM MTHF were contained in a final volume of 50 µl of the assay buffer. The reaction was started by adding [5-$^3$H]dUMP (1 µM final concentration, specific activity 5 mCi/mol), followed by incubation at 37 °C for 60 min, and was stopped by adding 50 µl of ice-cold 35% trichloroacetic acid. Residual [5-$^3$H]dUMP was removed by adding 250 µl of 10% neutral activated charcoal. The charcoal was removed by centrifugation at 14,000 g for 15 min at 4 °C, and a 150 µl sample of the supernatant was assayed for tritium radioactivity in the liquid scintillation analyzer Tri-Carb 2100 (Packard).

## Western blotting

Cells were harvested and washed in ice-cold PBS and suspended in RIPA buffer supplemented with protease and phosphatase inhibitors (Sigma Aldrich). The insoluble debris was removed by centrifugation at 14,000 g for 30 min. Cellular extracts were resolved using 12% SDS-PAGE and transferred to PVDF membranes (Hybond-P, Amersham). Immunoblot analysis was performed using anti-hTS (clone TS106, Millipore, 1:500 dilution), anti-hDHFR (clone 872442, Millipore, 1:1000) and anti-FLAG M2 (clone M2, F1804, Sigma-Aldrich, 1:1000 dilution), anti-β-Actin (clone AC-15, Santa Cruz Biotechnology, 1:1500 dilution). Horseradish peroxidase-conjugated secondary antibody (GE Healthcare UK Limited) was used to detect the bound primary antibody. Immune complexes were visualized by enhanced chemiluminescence (Amersham ECL Prime Western blotting reagent) following the manufacturer's instructions. Band density was calculated using Image J software or an image analyzer (GS-690 BIORAD).

## Treatment of A2780 with E7 and cross-linking experiments

The human ovarian cancer cell line A2780 were exposed to **E7** (stock solution 20 mM) at the final concentrations of 20 µM or vehicle (DMSO 0.1%) for 3 hr. The **E7** dose was decided based on the $IC_{50}$ of this compound in these cells. Proteic extracts from human A2780 cells exposed to **E7** or vehicle (DMSO) were cross-liked for 1 hr at room temperature with Bis(sulfosuccinimidyl)suberate ($BS^3$) (3 mM) (Sigma) on a rocker and then processed with the standard procedure (see western-blotting section) (*Shi et al., 2017*).

## RNA extraction and qPCR

Total RNA was isolated from the cells using TRIzol Reagent (Invitrogen). Reverse transcription was performed with 0.5 µg or 0.25 µg of total RNA using SuperScript first-strand synthesis for RT-PCR (Invitrogen). RT-PCR was performed with 50 ng of cDNA using SsoAdvanced Universal SYBR Green Supermix (Bio-Rad Laboratories). Samples were amplified by an initial denaturation at 95 °C for 30 s, followed by 40 cycles of denaturation at 95 °C for 10 s, primer annealing at 60 °C for 30 s, in CFX Connect Real-Time PCR machine (Bio-Rad Laboratories). The amplified were analyzed by the CFX maestro software (Bio-Rad Laboratories). The amount of target expressed was normalized with GAPDH. Twenty-four hr after transfection, total RNA was isolated from HCT116 tranfected with hTS-Myc-DDK-F59A mutant and hTS-Myc-DDK tagged wild type vectors (N=6 for each vectors from two indipendent experiment) and analyzed as previously described. Two primer pairs were designed for hTS target, one pair to amplify the total hTS mRNA and one pair for the exogenous hTS mRNA: FW_hTS1, 5'-CAGATTATTCAGGACAGGGAGTT-3'; REV_hTS1, 5'- CATCAGAGGAAGATCTCTTGGAT T-3'; FW_hTS2, 5'-*AGCGAGAACCCAGACCTTTC-3*'; REV-FLAG, 5'-TCATTTGCTGCCAGATCCTCTT-3'. For the reference target: GAPDH primer forward (hGAPDH1 fw) (Sigma Genosys 1062, 3174–083): 5'-CAAGGTCATCCATGACAACTTTG-*3*', reverse: *5'-GGGCCATCCACAGTCTTCTG-3*' (hGAPDH1 rw) (Sigma Genosys 1062, 3174–084). The expressions levels of total (endogenous plus exogenous) and exogenous hTS mRNAs were not significantly different (t=0.248, p=0.809 and t=1.583, p=0.145), respectively (see *Supplementary file 3*).

TS mRNA expression in the primary pancreatic cancer cells were measured by standard curves obtained with dilutions of cDNA from Quantitative-PCR Human-Reference Total-RNA (Stratagene), as described previously (*Zucali et al., 2011*).

## hTS-F59A mutant generation

Human thymidylate synthase (TYMS)-Myc-DDK-tagged vector (Origene RC204814) was used to generate a hTS-F59A mutant. hTS mutations were introduced by PCR, using the GENEART Site-Directed Mutagenesis System (Invitrogen). The mutagenic oligonucleotides used were: FW_hTS-F59A: 5'- GGCACCCTGTCGGTAGCCGGCATGCAGGCGCG - 3'; REV_hTS-F59A: 5'- CGCGCCTG CATGCCGGCTACCGACAGGGTGCC - 3'. PCRs for single amino acid mutations were run for 18 cycles of 20 s at 94 °C and 30 s at 57 °C, followed by 3 min and 40 s at 68 °C. Then, the products were analyzed on a 0.8% agarose gel, the recombination reaction was performed and DH5α-T1 competent cells were transformed. The resulting mutated plasmid were verified by DNA sequencing. After sequencing, 0.5 µg of hTS-MYC-DDK and hTS-MYC-DDK-F59A vectors were loaded on the agarose gel to determine the purity of DNA isolation.

## Apoptosis

After treatment, cells were stained with Annexin V and PI according to the manufacturer's protocol (Immunological Sciences, IK-11130). Then, apoptotic cells percentage was evaluated by flow cytometry. $1 \times 10^6$ cells were washed with PBS and stained with 2 µl of Annexin V and 2 µl of PI in 1×binding buffer for 15 min at room temperature in the dark. Early apoptotic cells (Annexin V-positive, PI-negative), late apoptotic cells (Annexin V-positive and PI positive) and necrotic cells (Annexin V-negative, PI-positive) were included in cell death measurements (*Figure 7—figure supplement 2*). Staurosporine treatment was used as a positive control (FACS Coulter Epics XL flow cytometer). An example of the gating strategy used for cell analysis is in *Figure 7—figure supplement 2*.

## Pharmacokinetic studies on BALB/c mice of E5 and E7

We tested both compounds in a preliminary pharmacokinetic study (Snapshot-PK) (*Li et al., 2013*).

Female BALB/c mice were purchased from Charles River Laboratories and maintained at IBMC Animal Facilities, in sterile IVC cabinets, with food and water available ad libitum. All animals used in experiments were aged from six to eight weeks. The mice were randomized into four groups of six and then treated with **E7** or **E5** evaluating two independent administration routes, oral or intra-venous (IV). The doses administered were 1 mg/kg for I.V. (administered in 100 µl in the tail vein) and 20 mg/kg for per os (oral gavage with 100 µl). Blood samples were taken from the tail vein at specific time points to heparinized tubes (two animals for each time point were used with around 20 µl of total blood recovered), plasma was recovered by centrifugation and stored at −20 °C until quantification by UHPLC-MS/MS ESI+. Plasma samples from 6 BALB/c mice treated with (1 m/kg/IV or 20 mg/kg/oral) compound **E5** or **E7** recovered at 0, 5, 15, 30, 45, 60, 120, 300 min and 24, 48 or 72 hr post administration for IV or 0, 30,45,60,120,180, 300 min and 24, 48, and 72 hr post administration. The detection of the compounds in the plasma was done by UHPLC-MS/MS ESI+ Acquity Quattro Premier WATERS, column Acquity BEH HILIC 1,7 µm (2,1x100 mm). All data are reported in *Figure 8—source data 1*. All animal experiments were carried out in accordance with the IBMC.INEB Animal Ethics Committees and the Portuguese National Authorities for Animal Health guidelines, according to the statements on the Directive 2010/63/EU of the European Parliament and of the Council. NS and ACS have an accreditation for animal research given by the Portuguese Veterinary Direction (Ministerial Directive 1005/92).

## In vivo cancer mouse model studies

### Orthotopic model

The orthotopic pancreatic ductal adenocarcinoma (PDAC) cancer mouse model was generated via direct injection of the primary PDAC-2 cells, transduced with a lentiviral vector containing Gaussia luciferase (G-luc), in the pancreas of female nude mice (age 6–8 weeks) anesthetized with isoflurane, as described previously (*Giovannetti et al., 2014*; *Wurdinger et al., 2008*). Pre-/post-operative pain was counteracted by administering Temgesic (0.05–0.1 mg/kg SC). Frozen tumors from PDAC-2 mice treated with E7 or 5FU (24 hr before their sacrifice) were used for RNA and protein extraction, as described above, to evaluate the mRNA expression of hTS, compared with untreated control mice using already validated primers for Real-time PCR (FW: *5'-CAGATTATTCAGGACAGGGAGTT-3'*; RW: *5'- CATCAGAGGAAGATCTCTTGGAT T-3'*). Immunohistochemical (IHC) staining were performed according to standard procedures. Sections of 3 µm pancreatic slices were cut from paraffin-embedded specimens. IHC was performed with the anti-hTS antibody described above (dilution 1:100), as well as with the anti-Cleaved Caspase-3 (Asp175) Antibody #9661 (dilution 1:100, Cell Signaling, Danvers, MA, USA) and the anti-Ki67 monoclonal antibody (dilution 1:100, clone MIB1; DBA, Milan, Italy), as described previously (*Bianco et al., 2006*, *Su et al., 2018*). Visualization was obtained with Bench Mark Special Stain Automation system (Ventana Medical Systems, inc, USA).

### Antitumor activity

Orthotropic PDAC models were generated via injection of $10^6$ primary cells into the pancreas of six six/eight weeks old female athymic nude mice (Charles River). Tumor growth was monitored using bioluminescence imaging (BLI) of Gaussian luciferase (G-luc) reporter. Five days after surgery, mice were stratified based on BLI intensities into three groups with comparable G-luc activity. Six mice per group were treated with 10 mg/kg **E7** 3 times/week (every 2 days) every 3 weeks (administered intraperitoneally, i.p.), or 100 mg/kg **5FU** i.p. once per week for 3 weeks (or q7dx3) (*Giovannetti et al., 2014*), whereas six control mice were treated only with sterile saline. Mice were sacrificed upon discomfort or >10% weight loss, and the log-rank test was used to evaluate significant differences in survival. Pancreas and internal organs were removed and frozen or fixed in 4% paraformaldehyde. To evaluate the modulation of hTS in vivo, two mice of the control group were exposed to a single dose of **E7** or **5FU** 24 hours before their sacrifice. qPCR and immunohistochemical studies were performed as described above. Animal experiments were approved by the Committees on Animal Experiments of the VU University Amsterdam, The Netherlands and of the University of Pisa, Pisa, Italy, and were performed in accordance with institutional guidelines and international law and policies.

## Xenograft model

Athymic Nude-Foxn1nu female mice aged 7 weeks (Envigo RMS Srl, Udine, Italy) were housed and handled under aseptic conditions, in accordance with the University of Ferrara Institutional Animal Care and Use Committee guidelines.

## Antitumor activity

Tumor xenografts in mice were obtained by subcutaneously implanting (s.c.) with $2 \times 10^6$ of A2780 cells suspended in 100 µL of PBS and 100 µL of Matrigel (BD Biosciences). When tumor mass became palpable in successfully engrafted mice (around 7 days after the injection), animals were weighed and randomly divided into three groups to be subjected to various treatments as indicated in *Figure 8— figure supplement 1*. Treatments administrated via intraperitoneal injection were **E7**, 5FU and vehicle solutions. **E7** was administrated 10 mg/kg three times a week while 5FU (100 mg/Kg) one time a week. During the experiment there was no difference in body weight between control and treatment groups. Tumor growth was monitored daily, and tumor diameters were measured with calipers every two days. The tumor volume was calculated using the following equation: volume = $\pi/6 \times (a \times b2)$, where **a** is the major diameter and **b** is the minor diameter. All mice that reached the endpoint of the experiment were euthanized and, subsequently, tumors were excised.

## Acknowledgements

We thank Dr. Hannu Myllykallio for cloning hTS and preparing bacterial cells expressing mutant proteins, and Dr. Amir Avan and Dr. Niccola Funel for their work on PDAC animal models and staining. The authors also acknowledge the 'Fondazione Cassa di Risparmio di Modena' for funding the UHPLC-ESI-QTOF system at the Centro Interdipartimentale Grandi Strumenti (CIGS). PP is grateful to Camilla degli Scrovegni for continuous inspirational support.

This work was funded by AIRC2015 IG16977 (MPC), European Union (LIGHTS-A Framework 6 STREP: LSH- 2005–2.2.0-8) Grant agreement n°037852 to MPC, RCW, GC, RMS; CCA foundation grants-2012/2015 CCA2015-1-19 to EG and GJP, AIRC 14422 Start-Up grant to EG, the Klaus Tschira Foundation (RCW, SH, OMHS-A) and the Alexander von Humboldt Foundation, the Finnish Cultural Foundation, the Academy of Finland (137918), and the University of Eastern Finland (OMHS-A). RMS is supported by NIH GM24485. PP is supported by Italian Association for Cancer Research (AIRC, IG23670). AR is supported by the Italian Ministry of Health (GR-2016–02364602) and by the Italian Ministry of Education, University and Research (PRIN, Grant 2017XA5J5N).

## Additional information

### Funding

| Funder | Grant reference number | Author |
| --- | --- | --- |
| Associazione Italiana per la Ricerca sul Cancro | IG25785 | Maria Paola Costi |
| Associazione Italiana per la Ricerca sul Cancro | IG16977 | Maria Paola Costi |
| Associazione Italiana per la Ricerca sul Cancro | 14422 Start-Up | Elisa Giovannetti |
| CCA foundation grant | CCA2015-1-19 | Rebecca C Wade |
| National Institute of General Medical Sciences | GM24485 | Robert M Stroud |
| Associazione Italiana per la Ricerca sul Cancro | IG23670 | Paolo Pinton |
| Ministero dell'Istruzione, dell'Università e della Ricerca | 2017XA5J5N | Alessandro Rimessi |

| Funder | Grant reference number | Author |
| --- | --- | --- |
| Academy of Finland | 137918 | Outi MH Salo-Ahen |
| European Commission | GA037852 | Maria Paola Costi |

The funders had no role in study design, data collection and interpretation, or the decision to submit the work for publication.

## Author contributions

Luca Costantino, Conceptualization, Supervision, Validation, Investigation, Writing – original draft, Writing – review and editing; Stefania Ferrari, Investigation, Methodology, Writing – original draft, Writing – review and editing; Matteo Santucci, Validation, Investigation, Methodology, Writing – original draft; Outi MH Salo-Ahen, Emanuele Carosati, Silvia Franchini, Angela Lauriola, Cecilia Pozzi, Puneet Saxena, Giuseppe Cannazza, Lorena Losi, Antonio Quotadamo, Pasquale Linciano, Rosaria Luciani, Filippo Genovese, Stefan Henrich, Silvia Alboni, Nuno Santarem, Anabela da Silva Cordeiro, Elisa Giovannetti, Godefridus J Peters, Alessandro Rimessi, Investigation, Methodology, Writing – original draft; Matteo Trande, Gaia Gozzi, Daniela Cardinale, Alberto Venturelli, Salvatore Pacifico, Paolo Pinton, Investigation, Writing – original draft; Lorenzo Tagliazucchi, Maria Gaetana Moschella, Investigation, Methodology, Writing – review and editing; Remo Guerrini, Supervision, Investigation, Methodology, Writing – original draft; Gabriele Cruciani, Software, Investigation, Methodology, Writing – original draft; Robert M Stroud, Supervision, Writing – original draft, Writing – review and editing; Rebecca C Wade, Supervision, Methodology, Writing – original draft, Writing – review and editing; Stefano Mangani, Domenico D'Arca, Supervision, Investigation, Methodology, Writing – original draft, Writing – review and editing; Gaetano Marverti, Data curation, Supervision, Investigation, Methodology, Writing – original draft, Writing – review and editing; Glauco Ponterini, Conceptualization, Data curation, Formal analysis, Investigation, Methodology, Writing – original draft, Writing – review and editing; Maria Paola Costi, Conceptualization, Data curation, Supervision, Funding acquisition, Investigation, Methodology, Writing – original draft, Writing – review and editing

## Author ORCIDs

Matteo Santucci http://orcid.org/0000-0003-4331-5566
Outi MH Salo-Ahen http://orcid.org/0000-0003-0725-126X
Angela Lauriola http://orcid.org/0000-0001-5286-8627
Cecilia Pozzi http://orcid.org/0000-0003-2574-3911
Rebecca C Wade http://orcid.org/0000-0001-5951-8670
Gaetano Marverti http://orcid.org/0000-0002-9074-0795
Glauco Ponterini http://orcid.org/0000-0002-2115-0775
Maria Paola Costi http://orcid.org/0000-0002-0443-5402

## Ethics

Animal experiments on orthotopic mice were approved by the Committees on Animal Experiments of the VU University Amsterdam, The Netherlands and of the University of Pisa, Pisa, Italy, and were performed in accordance with institutional guidelines and international law and policies. -------- Pharmacokinetic studies on BALB/c mice of E5 and E7. ': All animal experiments were carried out in accordance with the IBMC.INEB Animal Ethics Committees and the Portuguese National Authorities for Animal Health guidelines, according to the statements on the Directive 2010/63/EU of the European Parliament and of the Council. NS and ACS have an accreditation for animal research given by the Portuguese Veterinary Direction (Ministerial Directive 1005/92). ---------- Experiments performed in Ferrara University (IT). In vivo experiments were carried out by competent staff members, with a scientific education relevant to the experimental work proposals, including the ability to manipulate and taking care of laboratory animals. All the procedures were performed in accordance with Institutional Animal Care and Use Committee (IACUC), in strict accordance with the recommendations in the Guide for the Care and Use of Laboratory Animals of the Italian Ministry of Health. The experimental protocol was approved by the Animal Ethics Committee at the University of Ferrara and by the Italian Ministry of Health. Authorization number n° 1098/2020-PR, released according to art. 31 of D.lgs. 26/2014.

Decision letter and Author response
Decision letter https://doi.org/10.7554/eLife.73862.sa1
Author response https://doi.org/10.7554/eLife.73862.sa2

## Additional files

### Supplementary files
- Supplementary file 1. Computational protein-pocket identification for tethering.
- Supplementary file 2. X-ray crystallography.
- Supplementary file 3. Mass spectrometry data.
- Supplementary file 4. Compounds FRET and inhibition data.
- Supplementary file 5. Mass Spectrometry analysis of the fractions eluted from AEX.
- Supplementary file 6. Cancer cell growth inhibition and pharmacokinetic.
- Transparent reporting form

### Data availability

Diffraction data have been deposited in PDB under the accession codes PDB-ID 4O1U humanTS mutant Y202C and PDB-ID 4O1X humanTS double mutant C195SY202C. All other data generated or analyzed during this study are included in the manuscript and as Supplementary files and Source data.

The following previously published datasets were used:

| Author(s) | Year | Dataset title | Dataset URL | Database and Identifier |
| --- | --- | --- | --- | --- |
| Pozzi C, Mangani S | 2015 | Crystal structure of human thymidylate synthase double mutant C195S-Y202C | https://www.rcsb.org/structure/4O1X | RCSB Protein Data Bank, 10.2210/pdb4O1X/pdb |
| Pozzi C, Mangani S | 2015 | Crystal structure of human thymidylate synthase mutant Y202C | https://www.rcsb.org/structure/4O1U | RCSB Protein Data Bank, 10.2210/pdb4O1U/pdb |

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

## Appendix 1

## Details of Mass Spectrometry (MS) methods

### MS-based disulphide library screening

A solution of each individual library member was prepared in dimethyl sulfoxide (DMSO) to a final concentration of 80 mM per ligand. 1 µL of each ligand solution was then pooled to form groups of a maximum of 3 discrete compounds, having molecular weights that differed by at least 150 Da from one another; each pool was brought to a final volume of 10 µL. 1 µL of the desired ligand pool (8 mM) was added to 20 µL of 19 µM C195S/Y202C hTS double mutant solution in a buffer containing 1 mM 2-mercaptoethanol and 20 mM potassium phosphate (pH 7.5) to a final volume of 40 µL. The protein was present at a concentration of 10 µM while each of the disulphide library members was 0.2 mM. The reaction mixture was incubated at room temperature for 1 hour and then purified on a C4 ZipTip (Millipore) to remove salts and unbound ligands. A solution of 10 mg/ml of sinapinic acid in acetonitrile/0.1% aqueous TFA in a 1:1 ratio was used to assist the ionization process. 1 µl of the sample was deposited onto a MALDI sample plate and allowed to air-dry at room temperature; 0.5 µL of matrix solution were then deposited on the same spot, resulting in a uniform layer of fine granular protein including matrix crystals. Mass spectra were recorded in positive-ion linear mode at an accelerating voltage of 20 kV and a delayed extraction time of 1200 ns. Each spectrum obtained was the mean of 100 laser shots. The ions were generated using the 337 nm laser beam from a nitrogen laser, having a pulse width of 3 ns. Data were obtained using the following parameters: 95% grid voltage, 0.08% guide wire voltage and a low mass gate of 15,000 Da. An external calibration was performed based on the molecular weight of the untreated enzyme mutant previously reduced with DTT, alkylated with iodoacetamide and diluted 1:10 in TFA 0.1%. The software Data Explorer (Applied Biosystems) was used for mass spectral data processing; raw spectra were smoothed, and the baseline was corrected according to the normally observed peak broadness. Peak m/z values were annotated according to a peak detection algorithm taking into account an average resolution of 3500. The identity of any library member bonded to the protein through a disulphide bond was determined by subtracting the mass of the free protein from the observed higher mass.

For ESI-QTOF measurement 0.5 µl of ligand (80 mM in DMSO) was added to 12 µl of 17 µM C195S/Y202C hTS mutant solution in a buffer containing 1 mM 2-mercaptoethanol and 20 mM potassium phosphate (pH 7.5) to a final volume of 20 µL. The protein was present at a concentration of 10 µM and the disulphide at 2 mM. After equilibration at room temperature for 1 hour, 1 µl of the reaction mixture was diluted 1:10 with a solution of 5% formic acid in water/acetonitrile 95:5 and injected onto an HPLC-chip ESI-QTOF. The mobile phase composition was as follows: (A) 0.1%formic acid in a water/acetonitrile 98:2 solution and (B) 0.1%formic acid in a acetonitrile/water 98:2 solution. Chromatographic separations were performed at a flow rate of 400 nL/min with the following gradient: 15% B for 5 min, 15–85% B in 5 min, 85% B for 5 min, 85–15% B in 3 min. Under these conditions the protein-ligand complex was eluted between 6.5 and 8.0 min. The capillary voltage was set to 1600 V and the desolvating temperature was 350 °C. Nitrogen was used as a drying gas (flow rate = 6 L/min). The mass spectrometer operated in positive mode in the scan range 200–3200 Da. An external calibration was performed based on the molecular weight of a multi-standard solution (ESI-L Low concentration tuning mixture - SUPELCO). During acquisition, some of these masses were constantly monitored and used for a fine calibration of the instrument. The multiply charged ions deriving from the free protein and the protein-ligand complex were deconvoluted with available software (MassHunter – Agilent Technologies Inc).

### Analysis of the ligand-protein binding site by MS

To investigate the cysteine residue involved in the binding with the selected ligands, the protein-ligand complexes underwent enzymatic digestion in a classical bottom-up proteomic analysis using both ionization techniques for result evaluation.

In a typical experiment, 1 µL of the ligand (80 mM in DMSO) was added to 14 µL of the 29 µM hTS C195S-Y202C hTS mutant solution in a buffer containing 2 mM 2-mercaptoethanol and 20 mM phosphate buffer (pH 7.5) to a final volume of 20 µL. The reaction mixture was incubated at room temperature for 1 hour. 6 µL of iodoacetamide (100 mM) was added and the reaction mixture was incubated in the dark, at room temperature, for 20 min in order to alkylate the unmodified cysteine residues. 3 µL of 0.1 µg/µL sequencing grade modified trypsin (SIGMA) was added and the sample was incubated at 37 °C. After 1 hour, a second aliquot of 3 µL of trypsin was added. The digestion

was carried out at 37 °C overnight. 1 μL of trifluoroacetic acid (TFA) 0,1% was added and the reaction mixture was purified by a tip chromatography system (ZipTip C18, Millipore). The bound peptides were eluted from the tip stationary phase with 10 μL of acetonitrile 75%/TFA 0,1%. and 1 μL of the mixture was spotted onto a MALDI target for MS analysis. A saturated solution of a-cyano-4-hydroxycinnamic acid in 50% acetonitrile/0.1% trifluoroacetic acid (0.5 μL) was used to assist the ionization process. Mass spectra were recorded on a 4800plus MALDI TOF/TOF (Applied Biosystems) mass spectrometer, operating in positive ion reflector mode with an accelerating voltage of 20 KV and delayed extraction time of 300 ns. The MALDI ions were generated using a 355 nm Nd:YAG laser pulsed at 200 Hz with an intensity of 4600 (arbitrary units). Each spectrum was obtained by the accumulation of 1200–2500 laser shots. For MS/MS analysis the mass spectrometer was externally calibrated, and the sample mixture was analyzed in the 750–3500 Da mass range. Fragmentation experiments were performed with and without a collision gas. Signals from 150 laser shots were averaged to increase the S/N ratio of each mass spectrum. Data Explorer (Applied Biosystems) was used for data processing (baseline correction, deisotoping, calibration and peak detection). Peptide mass lists were searched with the MASCOT server in a swiss-prot-based protein database including the mutant version of hTS.

In parallel, 1 μL of the reaction mixture was diluted 1:30 with a solution of 0.5% formic acid in water/acetonitrile 98:2 and 3 μL were injected onto an HPLC-chip ESI-QTOF. The mobile phase composition was as follows: (A) 0.1% formic acid in a water/acetonitrile 98:2 solution and (B) 0.1% formic acid in acetonitrile/water 98:2 solution. A chromatographic separation was performed at a flow rate of 400 nL/min with the following gradient: 3% B for 5 min, 3–30% B in 15 min, 30–40% B in 5 min, 40–85% B in 3 min, 85% B for 2 min, 85–3% B in 5 min. The capillary voltage was set at1600 V and the desolvating temperature was 350 °C. Nitrogen was used as a drying gas (flow rate = 6 L/min). The mass spectrometer operated in positive mode in the 200–3200 Da scan range. For MS/MS experiment up to three precursors were selected for each acquisition cycle. The precursor ion was automatically excluded for fragmentation if it was present in two consecutive spectra, or if it persisted for more than 0.12 min. An external calibration was performed based on the molecular weight of a multi-standard solution (ESI-L Low concentration tuning mixture - SUPELCO). During the acquisition, some of these masses were constantly monitored and used for a fine calibration of the instrument. The multiply charged ions deriving from the free protein and the protein-ligand complex were deconvoluted with available software (MassHunter – Agilent Technologies Inc). The results of the trypsin digestion are reported in *Supplementary file 3* Table S8. To investigate which of the three cysteines of the D186-R215 peptide (*Figure 2* in the main text) were involved in the binding with the ligand, MS/MS experiments were performed. Unfortunately, the MS/MS spectra did not allow the identification of the corresponding fragment due to (i) the low abundance of the ligand bound-D186-R215 peptide and (ii) ion suppression effects. However, molecular dynamics and x-ray crystallography highlighted that among the three cysteine residues, cysteine 202 was the one exposed to the solvent and thus available for tethering.

## Mass spectrometric identification of tethered ligands

Matrix-Assisted Laser Desorption/Ionization (MALDI) - Time of Flight (TOF) Voyager-Pro MALDI-TOF (Applied Biosystems) was chosen for a medium-throughput screening of the disufide-compound library (**A1-A55**). The results obtained were validated using an Electron Spray Ionization Quadrupole - Time of Flight (ESI-QTOF) mass spectrometer (Agilent Technologies Inc). Both platforms allowed us to exclude any ionization-related issue in the ligand selection process. The screening approach was an iterative selection of the best ligands in a competitive environment obtained by arranging the ligands in pools properly chosen in order to avoid ambiguous mass peak assignments. Successive screening steps were performed in which the selected ligands from the previous screening were subjected to further screening using differently constituted pool. For the disulfide-compound library screening up to 3 ligands were pooled and mixed into a hTS-C195S-Y202C solution in a buffer containing 1 mM 2-mercaptoethanol and 20 mM $K^+$ phosphate (pH 7.5). The reaction mixture was incubated at RT for 1 hour and then purified on a C4 ZipTip (Millipore) to remove salts and unbound ligands. To identify the binding site of these molecules, the protein-ligand complexes underwent enzymatic digestion in a classical bottom-up proteomic analysis using both ionization techniques for result evaluation.

# Appendix 2

## Synthetic chemistry for all compounds

### General

All commercial chemicals and solvents were reagent grade and were used without further purification. The compound purchased from NCI and ChemBridge were used without purification. Reaction progress was monitored by TLC on pre-coated silica gel 60 F254 plates (Merck) and visualization was accomplished with UV light (254 nm). [1]H and [13]C NMR spectra were recorded on a Bruker DPX200 or a Bruker FT-NMR AVANCE 400 spectrometers. Chemical shifts are reported as δ values (ppm) referenced to residual solvent (CHCl$_3$ at δ 7.26 ppm, DMSO at δ 2.50 ppm, MeOD at δ 3.31 ppm, CD$_3$OCD$_3$ at δ 2.05 ppm); $J$ values were given in Hz. When peak multiplicities are given the following abbreviations are used: s, singlet; d, doublet; t, triplet; q, quartet; m, multiplet; br, broadened signal. Two-dimensional NMR techniques (heteronuclear single quantum coherence and heteronuclear multiple bond correlation) were used to aid the assignment of signals in [1]H and [13]C spectra. Silica gel Merck (60–230 mesh) was used for column chromatography. Purity of the compounds was assayed by means of TLC (Merck F-254 silica gel). Analysis on compound purity was determined through liquid chromatography UV/Vis using Jasco LC system equipped with a Jasco PU-2080 Plus pump, coupled with a Jasco PU-2075 Plus UV/Vis detector. LC separation was performed on an Agilent Poroshell 120 50 mm × 3.0 mm analytical column, packed with EC-C18 2.7 μM as stationary phase (Agilent Technologies, Milan, Italy). 10 μL of a 100 μg/mL solution of compound in methanol was injected. A gradient was delivered at 0.2 mL/min using (A) 0.1% FA in water and (B) ACN. Samples were eluted with 5% B (0.00–5.00 min); 1–95% B (5.00–20.00 min); 95% B (20.00–300.00 min) and 5% B (30.00–40.00 min). The eluted was detected with UV detection at $\lambda$ =220 nm. All the compounds showed a level of purity above 95%. MS spectra were recorded by means of a Q-TOF Accurate-Mass G6520AA (Agilent Technologies). Microanalyses were carried out in the Microanalysis Laboratory of the Department of Life Sciences, Modena University.

Synthesis of TPC

## Synthesis of TPC

**Appendix 2—figure 1.** Synthesis of TPC. (i) Diethyl ether, NaOH 2 M, RT, 4 h; HCl conc.; (ii) HOBT, EDAC, TEA, DCC.

N-(4-toluenesulfonyl)-D-proline (3). A solution of *p*-toluene sulfonyl chloride 1 (1.64 g, 8.6 moll) in diethyl ether (16 mL) was added to a cold (0 °C) solution of D-proline (2) (0.99 g, 8.6 moll) in NaOH 2 M (8.6 mL, 17.4 moll). The reaction was stirred 4 h at room temperature (RT). The mixture was treated with concentrated HCl until pH 2. The aqueous solution was then extracted with ethyl acetate (6x10 mL), the organic layer was washed with brine (4x10 mL), dried ($Na_2SO_4$) and the solvent was removed under reduced pressure to yield a colorless glue. Yield 1.79 g, 77%; [1]H NMR (CDCl$_3$, 200 MHz) δ 2.12–1.72 (H-7 and H-8, m, 4 H), 2.44 (H-1, s, 3 H), 3.29 (H-9B, m, 1 H), 3.50 (H-9A, m, 1 H), 4.29 (H-6, dd, *J* 3.9, *J* 4.0, 1 H), 7.33 (H-3, d, *J* 7.9, 2 H), 7.77 (H-4, *J* 8.2, 2 H).

N-N'-[D-prolyl-N-(4-toluenesulfonyl)]-cistamine (TPC). A solution of compound **3** (0.16 g, 0.6 mmol) in dichloromethane (5 mL), N,N'-dicyclohexylcarbodiimide (DCC) (0.16 g, 0.8 mmol) were added in succession and the solution was stirred at RT for 30 min; after cooling at 0 °C, the suspended cystamine·2HCl (**4**) (0.068 g, 0.3 mmol) in dichloromethane (8 mL) and TEA (0.083 mL, 0.6 mmol) was added dropwise over 5 min. The reaction mixture was stirred at RT overnight, then filtered, and the solution was washed with NaHCO$_3$ (2x5 mL), NH$_4$Cl (2x5 mL) and H$_2$O (2x5 mL). The organic layer was dried over Na$_2$SO$_4$, the solvent was evaporated at reduced pressure to produce the product that was purified by column chromatography (EtOAc/CHCl$_3$ 2:8). Yield g. 0.087, 44% (oil); [1]H NMR (DMSO-d$_6$, 200 MHz) δ 1.58–1.97 (m, 8 H), 2.25 (s, 6 H), 2.62 (t, 4 H, *j* 5.90), 3.10 (m, 2 H), 3.35 (m, 5 H), 3.95 (m, 3 H), 7.36 (m, 4 H), 7.68 (m, 4 H), 8.05 (t, 2 H, *J* 5.90); [13]C NMR (DMSO-d$_6$, 50 MHz) δ 20.9, 23.2, 30.0, 37.9, 39.2, 42.0, 62.9, 129.5, 129.9, 133.5, 144.7, 174.3. ESI-HRMS calcd for C$_{28}$H$_{39}$N$_4$O$_6$S$_4$: 655.1752 (M+H$^+$), found 655.1723. Anal. Calc. for C$_{28}$H$_{38}$N$_4$O$_6$S$_4$: C 51.35, H 5.85, N 8.56, found, C, 51.66, H, 5.90, N, 8.40.

# Synthesis of compounds C1-C5, C10, D5-D8, D11-D14.

C1 (from **H1b** and **J1**)
C2 (from **H2b** and **J1**)
C3 (from **H3b** and **J1**)
C4 (from **H4b** and **J1**)
C5 (from **H5b** and **J1**)
C10 (from **H6b** and **J1**)
D5 (from **H4b** and **J2**)
D6 (from **H3b** and **J2**)
D7 (from **H3b** and **J3**)
D8 (from **H3b** and **J4**)
D11 (from **H3b** and **J5**)
D12 (from **H3b** and **J6**)
D13 (from **H3b** and **J7**)
D14 (from **H3b** and **J8**)

**Appendix 2—figure 2.** Synthesis of compounds C1-C5, C10, D5-D8, D11-D14. (i): TEA/CH₂Cl₂ (for **H1b**, **H2b**) or Na₂CO₃/H₂O (for **H2b**, **H3b**, **H5b**, **H6b**); (ii): K₂CO₃/acetone (for **C2-C5**, **D5-D8**, **D11-D14**) or TEA/THF (for **C1**, **C10**).

Synthesis of 2-Bromo-N-(substituted-phenyl)acetamide derivatives (H1b, H6b). General procedure. To a solution of o-cloroaniline (H1a) or 8-aminoquinoline (H6a) (10.0 mmol) and TEA (11.0 mmol) in anhydrous $CH_2Cl_2$ (30 mL), under stirring at 0 °C under $N_2$ atmosphere, a solution of bromoacetylbromide (10.0 mmol) in anhydrous $CH_2Cl_2$ (20 mL) was added dropwise. The suspension thus obtained was stirred at RT for 1 hour, then it was diluted with $CH_2Cl_2$, washed with a saturated solution of $NH_4Cl$; afterwards, the organic phase was dried ($Na_2SO_4$) and the solvent removed under reduced pressure. The residue thus obtained was purified by means of column chromatography as described.

2-Bromo-N-(2-chlorophenyl)acetamide (H1b): Yield 2.49 g (96%), m.p. 83–85°C (column chromatography cyclohexane/EtOAc 17.5:2.5), $^1$H-NMR (DMSO-d$_6$, 200 MHz) (δ) (ppm): 9.90 (1H, broad s), 7.70 (1H, dd, $J$ 7.9, $J$ 1.7), 7.50 (1H, dd, $J$ 7.9, $J$ 1.7), 7.35 (1H, dt, $J$ 7.6, $J$ 1.6), 7.25 (1H, dt, $J$ 7.7, $J$ 1.7), 4.15 (2H, s); $^{13}$C-NMR (DMSO-d$_6$, 50 MHz) (δ) (ppm): 165.77, 135.00, 130.05, 127.99, 127.25, 126.42, 30.13.

2-Bromo-N-(quinolin-8-yl)acetamide (H6b): Yield 2.40 g (91%), m.p. 93–96 °C C (column chromatography cyclohexane/EtOAc 8:2), $^1$H-NMR (DMSO-d$_6$, 200 MHz) (δ) (ppm): 10.70 (1H, broad s), 8.90–8.60 (2H, m), 8.15 (1H, dd, $J$ 8.3, $J$ 1.7), 7.65–7.40 (3H, m), 4.10 (1H, s); $^{13}$C-NMR (DMSO-d$_6$, 50 MHz) (δ) (ppm): 148.54, 138.61, 136.31, 127.92, 127.15, 122.43, 121.74, 116.59, 29.68.

Synthesis of Bromo-N-(carboxy- or 4-carboxymethyl- phenyl)-acetamide derivatives (H2b, H3b, H5b). General procedure. These compounds were synthesized according to the published procedure (64-65). To a solution of the appropriate aminobenzoic acid (**H2a** or **H3a** for the synthesis of **H2b** and **H3b,** respectively) or p-aminophenylacetic acid (**H5a**) for the synthesis of (**H5b**) (19.87 mmol) and $Na_2CO_3$ (56.60 mmol) in water (60 mL) at 0 °C under stirring, a solution of bromoacetylbromide (27.81 mmol) in anhydrous $CH_3CN$ was added dropwise during 10 min. The suspension thus obtained was stirred at 0 °C for another 10 min, then at RT for 10 min. Afterwards, the suspension was acidified (HCl 1 N, pH 1), and the precipitate thus obtained collected and washed with water. The products were then purified by trituration with anhydrous diethyl ether.

Bromo-N-(3-carboxyphenyl)acetamide (H2b). Yield 1.20 g (43%), m.p. 213–216°C (dec.), $^1$H-NMR (DMSO-d$_6$, 200 MHz) (δ) (ppm): 10.51 (1H, broad s), 8.22 (1H, s), 7.79 (1H, m), 7.66 (1H, d, $J$ 8.0), 7.45 (1H, t, $J$ 7.6), 4.05 (2H, s); $^{13}$C-NMR (DMSO-d$_6$, 50 MHz) (δ) (ppm): 167.44, 165.50, 139.27, 131.90, 129.61, 125.06, 123.79, 120.43, 30.71.

Bromo-N-(4-carboxyphenyl)acetamide (H3b). Yield 82%, m.p. 223–226°C (dec.), $^1$H-NMR (DMSO-d$_6$, 200 MHz) (δ) (ppm): 10.65 (1H, broad s), 7.90 (2H, m), 7.70 (2H, m), 4.05 (2H, s); $^{13}$C-NMR (DMSO-d$_6$, 50 MHz) (δ) (ppm): 167.26, 165.73, 143.02, 130.91, 126.23, 119.04, 30.72.

Bromo-N-(4-carboxymethylphenyl)acetamide (H5b). Yield 86%, m.p. 130–132°C (dec.), $^1$H-NMR (DMSO-d$_6$, 400 MHz) (δ) (ppm): 12.50 (1H, broad s), 10.50 (1H, s), 7.68 (2H, m), 7.37 (2H, m), 4.19 (2H, s), 3.68 (2H, s); $^{13}$C-NMR (DMSO-d$_6$, 100 MHz) (δ) (ppm): 173.18, 165.15, 137.63, 130.93, 130.22, 119.63, 40.60, 30.89.

Bromo-N-(2-carboxyphenyl)acetamide (H4b). The compound was synthesized according to the procedure reported (66). To a solution of anthranilic acid (H4a) (1.50 g, 10.9 mmol) in anhydrous DMF (5 mL) and dioxane (5 mL), at –5 °C under stirring under $N_2$ athmosphere, bromoacetylbromide (2.76 g, 13.67 mmol) was added dropwise. The suspension thus obtained was stirred at RTR for 12 hours, then ice (50 mL) was added to the reaction mixture; the solid thus obtained was collected and washed with water. Yield 2.52 g (89%), m.p. 163–166°C, $^1$H-NMR (DMSO-d$_6$, 200 MHz) (δ) (ppm): 11.55 (1H, broad s), 8.45 (1H, d, $J$ 8.6), 8.00 (1H, d, $J$ 7.8, $J$ 1.2), 7.60 (1H, m), 7.20 (1H, m), 4.25 (2H, s); $^{13}$C-NMR (DMSO-d$_6$, 50 MHz) (δ) (ppm): 169.65, 165.47, 140.42, 134.48, 131.57, 123.89, 120.51, 117.62, 31.10.

2-(Substituted aryl- or heteroarylthio)-N-(substituted phenyl)acetamide derivatives C2-C5, D5-D8, D11-D14. General procedure. To a suspension of $K_2CO_3$ (3.0 eq.) in anhydrous acetone (20 mL) at RT under stirring in a $N_2$ athmosphere, the appropriate mercaptoderivative (1.0 eq.) was added, followed by the appropriate (carboxyphenyl)-amidomethyl bromides (1 eq.).The suspension was stirred at RT for 12 hours, then the mixture was cooled, acified (HCl 1 N, pH 1.0) and the solid thus obtained was collected by filtration and washed with water. The crude product was purified by crystallization or column chromatography as reported below.

2-(4-Nitrophenylthio)-N-(3-carboxyphenyl)acetamide (C2): Yield 0.20 g (26%), m.p. 250–252°C (column chromatography $CH_2Cl_2$: $CH_3OH$ 9:1)., $^1$H-NMR (DMSO-d$_6$, 400 MHz) (δ) (ppm): 13.20 (1H,

broad s), 10.70 (1H, broad s), 8.37 (1H, s), 8.32 (2H, m), 7.93 (1H, m), 7.80 (1H, d, $J$ 7.9), 7.75 (2H, m), 7.60 (1H, m), 4.25 (2H, s); $^{13}$C-NMR (DMSO-d$_6$, 100 MHz) (δ) (ppm): 166.50, 166.71, 147.27, 145.19, 139.40, 131.86, 129.61, 126.91, 124.90, 124.38, 123.75, 120.38, 36.45; ESI-HRMS calcd for $C_{15}H_{13}N_2O_5S$ (M+H$^+$) 333.0545, found 333.0544. HPLC-UV/Vis $k$ (retention time): 11.76. Anal. Calc. for $C_{15}H_{12}N_2O_5S$, C, 54.21, H, 3.64, N, 8.43; found, C, 54.43, H, 3.69, N, 8.19.

2-(4-Nitrophenylthio)-N-(4-carboxyphenyl)acetamide (C3):Yield 0.28 g (36%), m.p. 234–236°C (column chromatography CH$_2$Cl$_2$: CH$_3$OH 9:1), $^1$H-NMR (DMSO-d$_6$, 400 MHz) (δ) (ppm): 12.89 (1H, broad s), 10.80 (1H, broad s), 8.31 (2H, m), 8.06 (2H, m), 7.83 (2H, m), 7.74 (2H, m), 4.28 (2H, s); $^{13}$C-NMR (DMSO-d$_6$, 100 MHz) (δ) (ppm): 167.30, 166.97, 147.19, 145.21, 143.16, 130.94, 126.92, 126.03, 124.38, 118.99, 36.56; ESI-HRMS calcd for $C_{15}H_{13}N_2O_5S$ (M+H$^+$) 333.0545, found 333.0526 HPLC-UV/Vis $k$: 11.77. Anal. Calc. for $C_{15}H_{12}N_2O_5S$, C, 54.21, H, 3.64, N, 8.43; found, C, 54.13, H, 3.66, N, 8.55.

2-(4-Nitrophenylthio)-N-(2-carboxyphenyl)acetamide (C4):Yield 0.60 (77%), m.p. 203–206°C (column chromatography CH$_2$Cl$_2$: CH$_3$OH 9:1), $^1$H-NMR (DMSO-d$_6$, 400 MHz) (δ) (ppm): 13.79 (1H, broad s), 11.83 (1H, broad s), 8.61 (1H, d, $J$ 8.2), 8.30 (2H, m), 8.10 (1H, dd, $J$ 7.9, $J$ 1.2), 7.76–7.70 (3H, m), 7.32 (1H, m), 4.40 (2H, s); $^{13}$C-NMR (DMSO-d$_6$, 100 MHz) (δ) (ppm): 169.61, 166.88, 146.47, 145.35, 140.52, 134.50, 131.57, 126.97, 124.47, 123.73, 120.63, 117.49, 36.85; ESI-HRMS calcd for $C_{15}H_{13}N_2O_5S$ (M+H$^+$) 333.0545, found 333.0548. HPLC-UV/Vis $k$: 12.71. Anal. Calc. for $C_{15}H_{12}N_2O_5S$, C, 54.21, H, 3.64, N, 8.43; found, C, 54.33, H, 3.45, N, 8.55.

2-(4-Nitrophenylthio)-N-(4-carboxymethylphenyl)acetamide (C5). Yield 64%, m.p. 142–144°C (column chromatography CH$_2$Cl$_2$/CH$_3$OH 9:1), $^1$H-NMR (DMSO-d$_6$, 400 MHz) (δ) (ppm): 12.20 (1H, broad s), 10.30 (1H, s), 8.18 (2H, m), 7.60 (2H, m), 7.55 (2H, m), 7.20 (2H, m), 4.09 (2H, s), 3.50 (2H, s); $^{13}$C-NMR (DMSO-d$_6$, 100 MHz) (δ) (ppm): 173.22, 166.29, 147.42, 145.16, 137.70, 130.82, 130.21, 126.88, 124.36, 119.62, 49.06, 36.45. ESI-HRMS calc. for $C_{16}H_{15}N_2O_5S$ (M+H$^+$) 347.0702, found 347.0709. HPLC-UV/Vis $k$: 11.59. Anal. Calc. for $C_{16}H_{14}N_2O_5S$, C, 55.48, H, 4.07, N, 8.09; found, C, 55.33, H, 3.95, N, 8.44.

2-(4-Carboxyphenylthio)-N-(2-carboxyphenyl)acetamide (D5). Yield 53%, m.p. 244–245°C (trituration with Et$_2$O), $^1$H-NMR (DMSO-d$_6$, 200 MHz) (δ) (ppm): 13.26 (1 H, broad s), 11.72 (1 H, s), 8.48 (1 H, d, J 8.2), 7.96 (2 H, m), 7.85 (2 H, m), 7.58 (2H, m), 7.43 (2 H, m), 7.15 (1 H, t, J 8.2); $^{13}$C-NMR (DMSO-d$_6$, 50 MHz) (δ) (ppm): 169.56, 167.31, 167.28, 142.20, 140.60, 134.45, 131.54, 130.32, 128.40, 126.85, 123.58, 120.48, 117.35, 37.23. ESI-HRMS calc. for $C_{16}H_{14}NO_5S$ (M+H$^+$) 332.0593, found 332.0590. HPLC-UV/Vis $k$: 11.01. Anal. Calc. for $C_{16}H_{13}NO_5S$, C, 58.00, H, 3.95, N, 4.23; found, C, 58.33, H, 3.85, N, 4.55.

2-(4-Carboxyphenylthio)-N-(4-carboxyphenyl)acetamide (D6). Yield 50%, m.p. 294–296°C (DMF/H$_2$O), $^1$H-NMR (DMSO-d$_6$, 400 MHz) (δ) (ppm): 12.80 (2H, broad s), 10.60 (1H, s), 7.88 (4H, m), 7.68 (2H, m), 7.48 (2H, m), 4.05 (2H, s); $^{13}$C-NMR (DMSO-d$_6$, 100 MHz) (δ) (ppm): 167.37, 167.31, 143.21, 142.96, 130.94, 130.24, 128.08, 127.75, 125.91, 118.93, 36.77. ESI-HRMS calc. for $C_{16}H_{14}NO_5S$ (M+H$^+$) 332.0593, found 332.0578. HPLC-UV/Vis $k$: 10.06. Anal. Calc. for $C_{16}H_{13}NO_5S$: C 58.00, H 3.95, N 4.23, found, C, 57.96, H, 3.68, N, 4.71.

2-(Benzoxazol–2-ylthio)-N-(4-carboxyphenyl)acetamide (D7). Yield 77%, m.p. 229–230°C (CH$_3$OH), $^1$H-NMR (DMSO-d$_6$, 400 MHz) (δ) (ppm): 12.70 (1H, broad s), 10.70 (1H, s), 7.91 (2H, m), 7.75–7.55 (4H, m), 7.32 (2H, m), 4.40 (2H, s); $^{13}$C-NMR (DMSO-d$_6$, 100 MHz) (δ) (ppm): 167.28, 165.99, 164.22, 151.81, 143.13, 141.67, 130.94, 126.04, 125.12, 124.81, 118.97, 118.71, 110.67, 37.28. ESI-HRMS calc. for $C_{16}H_{13}N_2O_4S$ (M+H$^+$) 329.0596, found 329.0592; HPLC-UV/Vis $k$: 11.83. Anal. Calc. for $C_{16}H_{12}N_2O_4S$: C 58.53, H 3.68, N 8.53, found, C, 58.87, H, 3.44, N, 8.47.

2-(Benzothiazol-2ylthio)-N-(4-carboxyphenyl)acetamide (D8). Yield 75%, m.p. 220–222°C (CH$_3$OH), $^1$H-NMR (DMSO-d$_6$, 400 MHz) (δ) (ppm):12.78 (1H, broad s), 10.81 (1H, s), 8.03 (1H, d, J 8.0, J 0.6), 7.93 (2H, m), 7.83 (1H, d, J 8.0), 7.73 (2H, m), 7.47 (1H, dt, J 7.4, J 1.2), 7.37 (1H, dt, J 8.2, J 1.1), 4.44 (2H,s); $^{13}$C-NMR (DMSO-d$_6$, 100 MHz) (δ) (ppm): 167.31, 166.41, 166.27, 152.97, 143.21, 135.26, 130.97, 126.89, 126.00, 125.03, 122.37, 121.56, 118.95, 38.27. ESI-HRMS calc. for $C_{16}H_{13}N_2O_3S_2$ (M+H$^+$) 345.0368, found 345.0366. HPLC-UV/Vis $k$: 12.31. Anal. Calc. for $C_{16}H_{12}N_2O_3S_2$, C, 55.80, H, 3.51, N, 8.13; found, C, 55.53, H, 3.45, N, 8.38.

2-(2-Carboxyphenylthio)-N-(4-carboxyphenyl)acetamide (D11). Yield 59%, m.p. 265–266°C (DMF/H$_2$O), $^1$H-NMR (DMSO-d$_6$, 400 MHz) (δ) (ppm): 11.05 (2H, broad s), 9.10 (1H, s), 6.75 (3H, m), 6.56 (2H, m), 6.42 (2H, m), 6.15 (1H, m)4.01 (2H, s); $^{13}$C-NMR (DMSO-d$_6$, 100 MHz) (δ) (ppm): 167.89,

167.87, 167.30, 143.31, 140.83, 132.90, 131.42, 130.89, 128.41, 126.21, 125.90, 124.78, 118.93, 37.18. ESI-HRMS calc. for $C_{16}H_{14}NO_5S$ (M+H$^+$) 332.0593, found 332.0581. HPLC-UV/Vis *k:* 10.09. Anal. Calc. for $C_{16}H_{13}NO_5S$: C 58.00, H 3.95, N 4.23, found, C, 58.19, H, 3.85, N, 4.58.

2-(Pyrimidin-2ylthio)-N-(4-carboxyphenyl)acetamide (D12). Yield 72%, m.p. 208–210°C (CH$_3$OH/Et$_2$O), $^1$H-NMR (DMSO-d$_6$, 200 MHz) (δ) (ppm): 12.60 (1H, broad s), 10.51 (1H, s), 8.64 (2H, d, *J* 9.2), 7.88 (2H, m), 7.68 (2H, m), 7.23 (1H, t, *J* 9.2), 4.10 (2H, s); $^{13}$C-NMR (DMSO-d$_6$, 50 MHz) (δ) (ppm): 167.31, 167.15, 158.25, 143.43, 130.87, 125.75, 118.86, 117.91, 36.00. ESI-HRMS calc. for $C_{13}H_{12}N_3O_3S$ (M+H$^+$) 290.0599, found 290.0594. HPLC-UV/Vis *k:* 9.04. Anal. Calc. for $C_{13}H_{11}N_3O_3S$, C, 53.97, H, 3.83, N, 14.52; found, C, 54.13, H, 3.85, N, 14.55.

2-(4-Acetylaminophenylthio)-N-(4-carboxyphenyl)acetamide (D13). Yield 81%, m.p. 262–264°C (CH$_3$OH), $^1$H-NMR (DMSO-d$_6$, 400 MHz) (δ) (ppm): 12.71 (1H, broad s), 10.40 (1H, s), 9.90 (1H, s), 7.88 (2H, m), 7.66 (2H, m), 7.53 (2H, m), 7.35 (2H, m), 3.80 (2H, s), 2.00 (3H, s); $^{13}$C-NMR (DMSO-d$_6$, 100 MHz) (δ) (ppm): 168.75, 167.97, 167.29, 143.31, 138.75, 130.94, 130.86, 129.00, 125.85, 120.02, 118.92, 39.38, 36.00. ESI-HRMS calc. for $C_{17}H_{17}N_2O_4S$ (M+H$^+$) 345.0909, found 345.0907. HPLC-UV/Vis *k:* 9.68. Anal. Calc. for $C_{17}H_{16}N_2O_4S$, C, 59.29, H, 4.68, N, 8.13; found, C, 59.55, H, 4.45, N, 8.35.

2-(5-Sulfamoylpyridin-2-ylthio)-N-(2-carboxyphenyl)acetamide (D14). Yield 72%, m.p. 268–270°C (CH$_3$OH/Et$_2$O), $^1$H-NMR (DMSO-d$_6$, 200 MHz) (δ) (ppm): 12.65 (1H, s), 10.60 (1H, s), 8.78 (1 H, d, J 1.4), 7.95 (1 H, dd, J 1.4, J 8.6), 7.88 (2 H, m), 7.69 (2 H, m), 7.60 (1 H, d, J 8.6), 7.49 (2 H, s), 4.23 (2 H, s); $^{13}$C-NMR (DMSO-d$_6$, 50 MHz) (δ) (ppm): 167.30, 167.08, 162.49, 146.57, 143.37, 136.89, 134.29, 130.89, 125.83, 121.92, 118.89, 40.05. ESI-HRMS calc. for $C_{14}H_{14}N_3O_5S_2$ (M+H$^+$) 368.0375, found 368.0368. HPLC-UV/Vis *k:* 9.29. Anal. Calc. for $C_{14}H_{13}N_3O_5S_2$, C, 45.77, H, 3.57, N, 11.44; found, C, 45.50, H, 3.45, N, 11.57.

2-(4-Nitrophenylthio)-N-(substituted)acetamide derivatives (C1) and (C10). General procedure. To a solution of (**H1b**) or (**H6b**) respectively (2.0 mmol) and TEA (3.0 mmol) in anhydrous THF (30 mL), 4-nitrothiophenol (**J1**) (2.0 mmol) in anhydrous THF (20 mL) was added under stirring at RT under a N$_2$ atmosphere, and the solution thus obtained was stirred for 2 hours; at the end of the reaction, the solution was diluted with CH$_2$Cl$_2$, washed with 1 N HCl (20 mL); the organic phase was dried (Na$_2$SO$_4$) and the solvent removed under reduced pressure. The residue thus obtained was purified by means of column chromatography (cyclohexane:EtOAc 8:2 then 100% EtOAc).

2-(4-Nitrophenylthio)-N-(2-chlorophenyl)acetamide (C1): Yield 0.44 g (67%), m.p. 144–146°C, $^1$H-NMR (DMSO-d$_6$, 400 MHz) (δ) (ppm): 10.10 (1H, broad s), 8.32 (2H, m), 7.87 (1H, d, *J* 7.7), 7.76 (2 H, m), 7.65 (1H, d, *J* 8.1), 7.47 (1H, m), 7.35 (1H, m), 4.40 (2H, s); $^{13}$C-NMR (DMSO-d$_6$, 100 MHz) (δ) (ppm): 166.9, 147.15, 145.25, 134.89, 130.02, 128.00, 127.08, 126.99, 126.77, 126.29, 124.37, 35.86; ESI-HRMS calcd for $C_{14}H_{12}ClN_2O_3S$ (M+H$^+$) 323.0257, found 323.0261; HPLC-UV/Vis *k:* 13.87. Anal. Calc. for $C_{14}H_{11}ClN_2O_3S$, C, 52.10, H, 3.44, N, 8.68; found C, 52.33, H, 3.22, N, 8.80.

2-(4-Nitrophenylthio)-N-(quinolin-8-yl)acetamide (C10): Yield 0.30 g (39%), m.p. 160–161°C, $^1$H-NMR (DMSO-d$_6$, 400 MHz) (δ) (ppm): 10.90 (1H, broad s), 9.08 (1H, dd, *J* 4.1, *J* 1.6) 8.75 (1H, dd, *J* 7.7, *J* 0.9), 8.56 (1H, dd, *J* 8.4, *J* 1.6), 8.30 (2H, m), 7.87–7.77 (4H, m), 7.73 (1H, m), 4.55 (2H, s); $^{13}$C-NMR (DMSO-d$_6$, 100 MHz) (δ) (ppm): 166.99, 149.50, 146.85, 145.33, 138.59, 137.11, 134.56, 128.33, 127.40, 127.13, 124.43, 122.87, 122.73, 117.18, 35.59; ESI-HRMS calcd for $C_{17}H_{14}N_3O_3S$ (M+H$^+$) 340.0756, found 340.0754. HPLC-UV/Vis *k:* 14.37. Anal. Calc. for $C_{17}H_{13}N_3O_3S$, C, 60.17, H, 3.86, N, 12.38; found, C 60.57, H, 3.60, N, 12.20.

## Synthesis of compounds D9 and D10.

**Appendix 2—figure 3.** Synthesis of D9 and D10. (**i**): $H_2$/Pd:C, $CH_3OH$.

General procedure. The compounds were obtained by means of catalytic reduction of the corresponding nitroderivatives (C3) and (C4), respectively. A solution of C3 and C4 (1.77 mmol) in $CH_3OH$ (100 mL) was hydrogenated on Pd/C 5% (0.26 g) at 2 atm for 6 hours. At the end of the reaction, the catalyst was filtered and the solvent removed under reduced pressure.

2-(4-Aminophenylthio)-N-(4-carboxyphenyl)acetamide (D9). Yield 68%, m.p. 238–240°C (dec.) ($CH_3OH$/EtOAc), $^1$H-NMR (DMSO-$d_6$, 400 MHz) (δ) (ppm): 10.93 (1H, s), 7.88 (2H, m), 7.68 (2H, m), 7.46 (2H, m), 7.23 (2H, m), 3.90 (2H, s); $^{13}$C-NMR (DMSO-$d_6$, 100 MHz) (δ) (ppm): 167.78, 167.32, 143.40, 134.38, 133.01, 132.39, 130.84, 130.16, 125.82, 123.29, 118.93, 112.81, 38.17. ESI-HRMS calc. for $C_{15}H_{15}N_2O_3S$ (M+H$^+$) 303.0803, found 303.0808. HPLC-UV/Vis $k$: 8.36. Anal. Calc. for $C_{15}H_{14}N_2O_3S$, C, 59.59, H, 4.67, N, 9.27; found, C, 59.77, H, 4.60, N, 9.08.

2-(4-Aminophenylthio)-N-(2-carboxyphenyl)acetamide (D10). Yield 0587 g (90%), m.p. 154–155°C, $^1$H-NMR (DMSO-$d_6$, 200 MHz) (δ) (ppm): 11.88 (1H, s), 8.50 (1H, d, J 8.4), 7.93 (1H, d, J 7.7), 7.55 (1H, t, J 7.9), 7.15 (3H, m), 6.50 (2H, m), 3.80 (2H, s); $^{13}$C-NMR (DMSO-$d_6$, 50 MHz) (δ) (ppm): 169.66, 168.39, 148.73, 140.95, 134.46, 133.83, 131.57, 123.25, 120.11, 118.68, 116.94, 115.26, 42.25. ESI-HRMS calc. for $C_{15}H_{15}N_2O_3S$ (M+H$^+$) 303.0803, found 303.0801. HPLC-UV/Vis $k$: 9.46. Anal. Calc. for $C_{15}H_{14}N_2O_3S$, C, 59.59, H, 4.67, N, 9.27; found, C, 59.80, H, 4.66, N, 9.55.

Synthesis of compounds C6-C9, C11, E1-E7

## Synthesis of compounds C6-C9, C11, E1-E7.

**C6** (from **K1d** and **L6**)
**C7** (from **K1d** and **L7**)
**C8** (from **K1d** and **L4**)
**C9** (from **K1d** and **L8**)
**C11** (from **K1d** and **L5**)
**E1** (from **K1a** and **J2**)
**E2** (from **K1b** and **J2**)
**E3** (from **K1c** and **J2**)
**E4** (from **K1c** and **L1**)
**E5** (from **K1c** and **L2**)
**E6** (from **K1c** and **L3**)
**E7** (from **K1c** and **L4**)

$R_1, R_2$

(**K1a**) $R_1$ = $CH_3$; $R_2$ = H
(**K1b**) $R_1$ = $R_2$ = $CH_3$

(**K1c**) $R_1$ = H; $R_2$ =

(**K1d**) $R_1$ = $R_2$ = H

**Appendix 2—figure 4.** Synthesis of compounds C6-C9 and C11, E1-E7. (i): $K_2CO_3$/acetone. In the case of K1d, methylbromoacetate was used instead of ethylbromoacetate; (ii): NaOH/EtOH; (iii): (COCl)$_2$/THF then TEA (for **E1-E6, C6, C7, C9, C11**) or WSC/DMF (for **C8, E7**).

Synthesis of methyl- or ethyl-2-[(4-nitrophenylthio)]acetate derivatives (K1a-K1d esters). General procedure. To a suspension of 4-nitrothiophenol (3.87 mmol) and $K_2CO_3$ (5.79 mmol) in anhydrous acetone (20 mL) under $N_2$ atmosphere under stirring at RT, the appropriate bromoacetate (K1a-K1d) (4.25 mmol) was added dropwise, and the suspension was stirred for 1 hour; in the case of (K1b), the reaction time was 12 hours. At the end of the reaction, water was added (40 mL) and the mixture was extracted with EtOAc (3x10 mL); the organic phase was dried ($Na_2SO_4$) and the solvent removed under reduced pressure.

Ethyl-[2-(4-nitrophenylthio)]propionate (K1a ester). Yield 90%, oil (column chromatography cyclohexane/EtOAc 8:2), $^1$H-NMR (CDCl$_3$, 400 MHz) (δ) (ppm): 8.16 (2H, m), 7.52 (2H, m), 4.20 (2H,

q, J 7.2), 4.05 (1H, q, J 7.2), 1.60 (3H, d, J 7.2), 1. 25 (3H, t, J 7.2); $^{13}$C-NMR (CDCl$_3$, 100 MHz) (δ) (ppm): 171.82, 146.08, 144.61, 128.94, 126.43, 124.45, 123.93, 61.99, 43.73, 17.26, 14.21.

Ethyl-[2-methyl(4-nitrophenylthio)]propionate (K1b ester). Yield 92%, oil (column chromatography cyclohexane/EtOAc 7.5:2.5), $^1$H-NMR (CDCl$_3$, 400 MHz) (δ) (ppm): 8.15 (2H, m), 7.60 (2H, m), 4.15 (2H, q, J 7.2), 1.60 (6H, s), 1.28 (3H, t, J 7.2); $^{13}$C-NMR (CDCl$_3$, 100 MHz) (δ) (ppm): 173.46, 147.71, 141.41, 135.16, 123.53, 61.63, 51.54, 26.08, 14.08.

Ethyl-[2-phenyl(4-nitrophenylthio)]acetate (K1c ester). Yield 77%, oil, $^1$H-NMR (CDCl$_3$, 400 MHz) (δ) (ppm): 8.11 (2H, m), 7.54 (2H, dd, J 8.0, J 2.0), 7.40 (5H, m), 4.20 (3H, m), 1.33 (3H, t, J 7.2); $^{13}$C-NMR (CDCl$_3$, 100 MHz) (δ) (ppm): 169.50, 146.18, 144.57, 134.33, 129.07, 128.95, 128.91, 128.44, 123.98, 62.37, 54.61, 14.04.

Methyl-2-[(4-nitrophenylthio)]acetate (K1d ester). Yield 82%, m.p. 47–49°C, $^1$H-NMR (DMSO-d$_6$, 200 MHz) (δ) (ppm): 8.15 (2H, m), 7.50 (2H, m), 4.15 (2H, s), 3.65 (3H, s); $^{13}$C-NMR (DMSO-d$_6$, 50 MHz) (δ) (ppm): 169.53, 146.46, 145.30, 135.23, 126.90, 124.39, 124.34, 52.97, 33.58.

2-[(4-Nitrophenylthio)] acetic acid derivatives (K1a-K1d acids). General procedure. To a solution of the appropriate ester (K1a-K1d esters) (6.87 mmol) in ethanol (43 ml) aqueous NaOH 2 M (10.31 mmol) was added and the solution thus obtained was stirred elm at RT under N$_2$ for 1 h, then the solvent was removed under reduced pressure. The residue thus obtained was dissolved in EtOAc, and the organic phase was washed with HCl 0.5 N; then the organic phase was dried (Na$_2$SO$_4$) and the solvent removed under reduced pressure; the product was then purified as reported below.

[2-(4-Nitrophenylthio)]propionic acid (K1a acid). Yield 70%, m.p. 110–111°C (trituration with isopropyl ether), $^1$H-NMR (DMSO-d$_6$, 400 MHz) (δ) (ppm): 8.16 (2H, m), 7.52 (2H, m), 4.05 (1H, q, J 7.2), 1.60 (3H, d, J 7.2); $^{13}$C-NMR (DMSO-d$_6$, 100 MHz) (δ) (ppm): 171.82, 146.08, 144.61, 128.94, 126.43, 124.45, 123.93, 43.73, 17.26.

[2-Methyl-(4-nitrophenylthio)]propionic acid (K1b acid). Yield 59%, m.p. 105–107°C (trituration with diethyl ether),$^1$H-NMR (CDCl$_3$, 400 MHz) (δ) (ppm): 8.15 (2H, m), 7.60 (2H, m), 1.60 (6H, s); $^{13}$C-NMR (CDCl$_3$, 100 MHz) (δ) (ppm): 173.46, 147.71, 141.41, 135.16, 123.53, 26.08.

[2-Phenyl(4-nitrophenylthio)]acetic acid (K1c acid). Yield 66%, m.p. 126–128°C (trituration with isopropyl ether), $^1$H-NMR (DMSO-d$_6$, 400 MHz) (δ) (ppm): 8.27 (2H, m), 7.74–7.66 (4H, m), 7.55–7.45 (3 H, m), 5.79 (1 H, s); $^{13}$C-NMR (DMSO-d$_6$, 100 MHz) (δ) (ppm): 171.08, 145.73, 145.59, 135.91, 129.28, 128.87, 128.24, 124.35, 53.22.

[(4-Nitrophenyl)thio]acetic acid (K1d acid). Yield 68%, m.p. 140–142°C (acetone/petrol ether b.p. 40–60°C), $^1$H-NMR (DMSO-d$_6$, 200 MHz) (δ) (ppm): 12.90 (1H, broad s), 8.15 (2H, m), 7.95 (2H, m), 4.0 (2H, s); $^{13}$C-NMR (DMSO-d$_6$, 50 MHz) (δ) (ppm): 170.29, 147.07, 145.12, 126.71, 124.30, 34.10.

2-(4-Nitrophenylthio)-N-(aryl- or heteroaryl-)acetamide derivatives C6, C7, C9, C11, E1-E6: General procedure. To a solution of the appropriate carboxylic acid (K1a-K1d acids) (4.69 mmol) in anhydrous THF (17 mL) and anhydrous DMF (0.033 mL) at 0 °C under elm stirring in a N$_2$ atmosphere, oxalylchloride (7.18 mmol) was added dropwise, then the mixture was stirred for 1 hour at RT. At the end of the reaction, the solvent was removed under reduced pressure and coevaporated with anhydrous CH$_2$Cl$_2$ three times. The oily material was used in the next step without purification. The solution of the acyl chloride thus obtained in CH$_2$Cl$_2$ (10 mL) was added dropwise to the solution of the appropriate amine derivative (1.2 eq) in anhydrous CH$_2$Cl$_2$ (20 mL), TEA (1.2 eq. in the case of (E5) and (E6), in the other cases 2.2 eq.) and anhydrous DMF (2–4 mL) at 0 °C under stirring in a N$_2$ atmosphere. After 12 hours, solvents were evaporated under reduced pressure, water was added to the residue, pH was adjusted to 1.00 (HCl 1 N) and the precipitate thus obtained was collected and washed with water. The crude product thus obtained was purified as reported below.

2-(4-Nitrophenylthio)-N-(indol-6-yl)acetamide (C6): Yield 0.167 g (40%), m.p. 192–194°C [column chromatography (cycloexane:EtOAc 4:6)], $^1$H-NMR (DMSO-d$_6$, 400 MHz) (δ) (ppm): 11.00 (1H, broad s), 10.25 (1H, broad s), 8.18 (2H, m), 7.94 (1H, s), 7.63 (2H, m), 7.46 (1H, d, J 8.3), 7.27 (1H, m), 7.03 (1H, dd, J 8.6, J 1.7), 6.36 (1H, s), 4.10 (2H, s); $^{13}$C-NMR (DMSO-d$_6$, 100 MHz) (δ) (ppm): 165.82, 147.63, 145.13, 136.28, 133.27, 126.90, 125.64, 124.65, 124.35, 120.38, 112.64, 102.76, 101.42, 36.60; ESI-HRMS calcd for C$_{16}$H$_{14}$N$_3$O$_3$S (M+H$^+$) 328.0756, found 328.0740. HPLC-UV/Vis $k$: 12.82. Anal. Calc. for C$_{16}$H$_{13}$N$_3$O$_3$S, C, 58.70, H, 4.00, N, 12.84; found, C, 58.77, H, 4.11, N, 12.89.

2-(4-Nitrophenylthio)-N-(acetylaminosulfonylphen-4-yl)acetamide (C7): The reaction was carried out as described in the general procedure, using a suspension of sodium sulfacetamide (0.54 g, 2.30 mmol) in anhydrous DMF (6 mL) and TEA (0.26 g, 2.53 mmol). Yield 0.37 g (39%), m.p. 227–

228°C (methanol), $^1$H-NMR (DMSO-d$_6$, 400 MHz) (δ) (ppm): 12.10 (1H, broad s), 10.90 (1H, broad s), 8.31 (2H, m), 8.15 (2H, m), 7.93 (2H, m), 7.74 (2H, m), 4.30 (2H, s), 2.05 (3H, s); $^{13}$C-NMR (DMSO-d$_6$, 100 MHz) (δ) (ppm): 169.13, 167.24, 147.06, 145.25, 143.64, 133.91, 129.45, 126.96, 124.39, 119.24, 36.53, 23.65; ESI-HRMS calcd for $C_{16}H_{16}N_3O_6S_2$ (M+H$^+$) 410.0480, found 410.0479. HPLC-UV/Vis $k$: 11.74. Anal. Calc. for $C_{16}H_{15}N_3O_6S_2$, C, 46.94, H, 3.69, N, 10.26; found, C, 46.66, H, 3.80, N, 10.33.

2-(4-Nitrophenylthio)-N-(2-chlorobenzyl)acetamide (C9): Yield 0.40 g (85%), m.p. 122–123°C (column chromatography cyclohexane:EtOAc 8:2)., $^1$H-NMR (DMSO-d$_6$, 400 MHz) (δ) (ppm): 8.90 (1H, t, J 5.5), 8.28 (2H, m), 7.72–7.70 (2H, m), 7.57 (1H, m), 7.43–7.41 (3H, m), 4.50 (2H, d, J 5.5), 4.11 (2H, s); $^{13}$C-NMR (DMSO-d$_6$, 100 MHz) (δ) (ppm): 167.80, 147.41, 145.15, 136.30, 132.62, 129.62, 129.46, 129.26, 127.56, 126.99, 124.28, 40.97, 35.27; ESI-HRMS calcd for $C_{15}H_{14}ClN_2O_3S$ (M+H$^+$) 337.0413, found 337.0403. HPLC-UV/Vis $k$: 13.33. Anal. Calc. for $C_{15}H_{13}ClN_2O_3S$, C, 53.49, H, 3.89, N, 8.32; found, C, 53.66, H, 3.96, N, 8.20.

2-(4-Nitrophenylthio)-N-(3-carboxypyridin-2-yl)acetamide (C11). Yield 21%, m.p. 194–196°C (H$_2$O), $^1$H-NMR (DMSO-d$_6$, 400 MHz) (δ) (ppm): 11.11 (1H, s), 8.53 (1H, dd, J 7.1, J 1.9), 8.22 (1H, dd, J 8.2, J 2.3), 8.15 (2H, m), 7.56 (2H, m), 7.28 (1H, dd, J 8.2, J 7.1), 4.30 (2H, s); $^{13}$C-NMR (DMSO-d$_6$, 100 MHz) (δ) (ppm): 167.92, 167.31, 151.56, 150.12, 147.26, 147.00, 140.26, 126.97, 126.73, 124.31, 120.29, 36.86. ESI-HRMS calc. for $C_{14}H_{12}N_3O_5S$ (M+H$^+$) 334.0498, found 334.0490. HPLC-UV/Vis $k$: 9.05. Anal. Calc. for $C_{14}H_{11}N_3O_5S$, C, 50.45, H, 3.33, N, 12.61; found, C, 50.33, H, 3.45, N, 12.85.

2-(4-Nitrophenylthio)-N-(4-carboxyphenyl)propionamide (E1). Yield 74%, m.p. 232–234°C (trituration with diethyl ether), $^1$H-NMR (DMSO-d$_6$, 400 MHz) (δ) (ppm): 12.80 (1H, broad s), 10.66 (1H, s), 8.18 (2H, m), 7.92 (2H, m), 7.70 (2H, m), 7.61 (2H, m), 4.43 (1 H, q, J 6.9), 1.58 (3H, d, J 6.9); $^{13}$C-NMR (DMSO-d$_6$, 100 MHz) (δ) (ppm): 170.07, 167.29, 145.78, 145.32, 143.06, 130.90, 128.82, 126.15, 124.52, 119.22, 45.58, 18.33. ESI-HRMS calc. for $C_{16}H_{15}N_2O_5S$ (M+H$^+$) 347.0702, found 347.0701. HPLC-UV/Vis $k$: 12.31. Anal. Calc. for $C_{16}H_{14}N_2O_5S$, C, 55.48, H, 4.07, N, 8.09; found, C, 55.23, H, 3.95, N, 8.55.

2-Methyl-2-(4-nitrophenylthio)-N-(4-carboxyphenyl)propionamide (E2). Yield 20%, m.p. 215–217°C (column chromatography CH$_2$Cl$_2$/CH$_3$OH 9.5:0.5), $^1$H-NMR (DMSO-d$_6$, 400 MHz) (δ) (ppm): 12.80 (1H, broad s), 10.11 (1H, s), 18.18 (2H, m), 7.90 (2H, m), 7.75 (2H, m), 7.5r8 (2 H, m), 1.70 (6H, s); $^{13}$C-NMR (DMSO-d$_6$, 100 MHz) (δ) (ppm): 170.28, 167.00, 148.20, 144.10, 143.21, 130.42, 130.15, 126.52, 120.44, 120.00, 50.42, 26.51. ESI-HRMS calc. for $C_{17}H_{17}N_2O_5S$ (M+H$^+$) 361.0858, found 361.0844. HPLC-UV/Vis $k$: 13.02. Anal. Calc. for $C_{17}H_{16}N_2O_5S$, C, 56.66, H, 4.47, N, 7.77; found, C, 56.39, H, 4.45, N, 8.00.

2-Phenyl-2-(4-nitrophenylthio)-N-(4-carboxyphenyl)carboxyilic acid (E3). Yield 40%, m.p. 219–221°C (column chromatography CH$_2$Cl$_2$/CH$_3$OH 9.5:0.5), $^1$H-NMR (DMSO-d$_6$, 400 MHz) (δ) (ppm): 12.80 (1H, broad s), 8.18 (2H, m), 7.90 (2H, m), 7.69 (2H, m) 7.64 (2H, m), 7.55 (2H, m), 7.40 (2H, m), 7.35 (1H, m), 5.75 (1H, s); $^{13}$C-NMR (DMSO-d$_6$, 100 MHz) (δ) (ppm): 167.63, 167.23, 145.72, 145.66, 142.80, 136.15, 130.93, 129.34, 128.99, 128.74, 128.13, 126.41, 124.60, 119.30, 54.98. ESI-HRMS calc. for $C_{21}H_{17}N_2O_5S$ (M+H$^+$) 409.0858, found 409.0867. HPLC-UV/Vis $k$: 13.63. Anal. Calc. for $C_{21}H_{16}N_2O_5S$, C, 61.76, H, 3.95, N, 6.86; found, C, 61.70, H, 3.70, N, 6.95.

2-Phenyl-2-(4-nitrophenylthio)-N-(3-hydroxy-4-carboxyphenyl)acetamide (E4). Yield 23%, m.p. 218–220°C (column chromatography CH$_2$Cl$_2$/CH$_3$OH 8.5:1.5), $^1$H-NMR (DMSO-d$_6$, 400 MHz) (δ) (ppm)15.80 (1H, broad s), 10.58 (1H, s), 8.18 (2H, s), 7.63 (3H, m), 7.52 (2H, m), 7.44–7.30 (3H, m), 6.99 (1H, d, J 1.9), 6.82 (1H, dd J 8.4, J 1.9), 5.71 (1H, s); $^{13}$C-NMR (DMSO-d$_6$, 100 MHz) (δ) (ppm): 167.07, 163.59, 145.2, 145.62, 141.97, 136.47, 131.02, 129.28, 128.88, 128.69, 127.99, 124.57, 116.23, 108.31, 106.70, 54.96. ESI-HRMS calc. for $C_{21}H_{17}N_2O_6S$ (M+H$^+$) 425.0807, found 425.0793. HPLC-UV/Vis $k$: 13.89. Anal. Calc. for $C_{21}H_{16}N_2O_6S$, C, 59.43, H, 3.80, N, 6.60; found, C, 59.59, H, 3.95, N, 6.55.

2-Phenyl-2-(4-nitrophenylthio)-N-(4-carboxamidophenyl)acetamide (E5). Yield 59%, m.p. 228–230°C, $^1$H-NMR (DMSO-d$_6$, 400 MHz) (δ) (ppm): 10.90 (1H, s), 8.31 (2H, m), 8.00 (3H, m), 7.80 (4H, m), 7.70 (2H, m), 7.60–7.45 (2H, m), 7.40 (2H, s), 5.81 (1H, s); $^{13}$C-NMR (DMSO-d$_6$, 100 MHz) (δ) (ppm): 167.64, 167.43, 145.70, 141.34, 136.20, 130.08, 129.34, 128.95, 128.73, 128.11, 124.61, 119.10, 55.04. ESI-HRMS calc. for $C_{21}H_{18}N_3O_4S$ (M+H$^+$) 408.1018, found 408.1038. HPLC-UV/Vis $k$: 12.73. Anal. Calc. for $C_{21}H_{17}N_3O_4S$, C, 61.90, H, 4.21, N, 10.31; found, C, 61.56, H, 3.99, N, 10.50.

2-(4-Nitrophenylthio)-N-(3-carbomethoxypyridin-6-yl)acetamide (E6). Methyl (6-amino)nicotinate (L3), starting material for the synthesis of (E6), was obtained via esterification of 6-amino-nicotinic

acid in MeOH saturated with $HCl_g$., prepared in situ adding $HCl_g$ to a suspension of the nicotinic acid (0.730 g, 5.29 mmol) in MeOH (50 ml) at 0 °C. Then the reaction mixture was refluxed at 100 °C under stirring in a $N_2$ atmosphere for 3 hours. The solvent was removed under reduced pressure and $Na_2CO_3$ was added up to pH = 9. A white powder was collected. Yield 87%. Compound E6 was thus obtained following the general procedure described above. Yield 65%, m.p. 138–140°C (EtOH), $^1$H-NMR (DMSO-$d_6$, 400 MHz) (δ) (ppm): 11.64 (1H, s), 8.90 (1H, s), 8.29 (1H, dd, J 8.7, J 2.3), 8.15 (3H, m), 7.63 (2H, m), 7.55 (2H, m), 7.45–7.35 (3H, m), 5.90 (1H, s)3.94 (3H, s); $^{13}$C-NMR (DMSO-$d_6$, 100 MHz) (δ) (ppm): 168.60, 165.16, 154.97, 150.07, 145.78, 145.38, 140.10, 135.87, 129.42, 129.11, 128.77, 128.12, 124.65, 122.07, 113.26, 54.32, 52.65. ESI-HRMS calc. for $C_{21}H_{18}N_3O_5S$ (M+H$^+$) 424.0967, found 424.0964. HPLC-UV/Vis $k$: 15.00. Anal. Calc. for $C_{21}H_{17}N_3O_5S$, C, 59.57, H, 4.05, N, 9.92; found, C, 59.66, H, 3.98, N, 9.75.

2-(4-Nitrophenylthio)- and phenyl-2-(4-nitrophenylthio)-N-(6-methanesulfonylbenzothiazol-2-yl)acetamide (C8) and (E7). General procedure. To a solution of (K1d acid) and (K1c acid) respectively (0.50 g, 2.35 mmol) in anhydrous DMF (10 mL) under stirring at RT in a $N_2$ atmosphere, 1-ethyl-3-(3-dimethylaminopropyl)carbodiimide (WSC) (0.45 g, 2.35 mmol), 2-amino-6-methanesulphonylbenzothiazole (L4) (0.54 g, 2.35 mmol) and HOBt (0.32 g, 2.35 mmol) were added in sequence. After 12 hours at RT the mixture was treated with water (2x10 mL) and the residue thus obtained was purified by trituration with EtOAc.

2-(4-Nitrophenylthio)-N-(6-methanesulfonylbenzothiazol-2-yl)acetamide (C8): Yield 35%, m.p. 217–218°C (ethyl acetate). $^1$H-NMR (DMSO-$d_6$, 400 MHz) (δ) (ppm): 13.00 (1H, broad s), 8.64 (1H, s), 8.18 (2H, m), 7.97 (2H, s), 7.61 (2H, m), 4.30 (2H, s), 3.22 (3H, s); $^{13}$C-NMR (DMSO-$d_6$, 100 MHz) (δ) (ppm): 168.48, 162.22, 152.43, 146.42, 145.41, 136.09, 132.48, 127.12, 125.39, 124.45, 122.65, 121.49, 44.48, 35.33. ESI-HRMS calc. for $C_{16}H_{14}N_3O_5S_3$ (M+H$^+$) 424.0096, found 424.0096. HPLC-UV/Vis $k$: 12.44. Anal. Calc. for $C_{16}H_{13}N_3O_5S_3$, C, 45.38, H, 3.09, N, 9.92; found, C, 45.65, H, 3.25, N, 9.75.

Phenyl[2-(4-nitrophenylthio)-N-(6-methanesulfonylbenzothiazol-2-yl)]acetamide (E7): Yield 18%, m.p. 95 °C (dec.) (column chromatography $CH_2Cl_2$:$CH_3OH$ 9.5:0.5, then trituration with diisopropyl ether). $^1$H-NMR (DMSO-$d_6$, 200 MHz) (δ) (ppm): 13.30 (1H, broad s), 8.66 (1H, s), 8.12 (2H, m), 7.95 (2H, s), 7.70–7.30 (7H, m), 5.83 (1H, s), 3.20 (3H, s); $^{13}$C-NMR (DMSO-$d_6$, 50 MHz) (δ) (ppm): 174.01, 168.10, 152.09, 146.03, 136.30, 134.99, 129.52, 129.35, 128.93, 128.67, 125.44, 124.67, 122.72, 121.62, 53.98, 44.46. ESI-HRMS calc. for $C_{22}H_{18}N_3O_5S_3$ (M+H$^+$) 500.0409, found 500.0404. HPLC-UV/Vis $k$: 14.16. Anal. Calc. for $C_{22}H_{17}N_3O_5S_3$, C, 52.89, H, 3.43, N, 8.41; found, C, 52.65, H, 3.35, N, 8.65.

# Synthesis of compounds D1-D4.

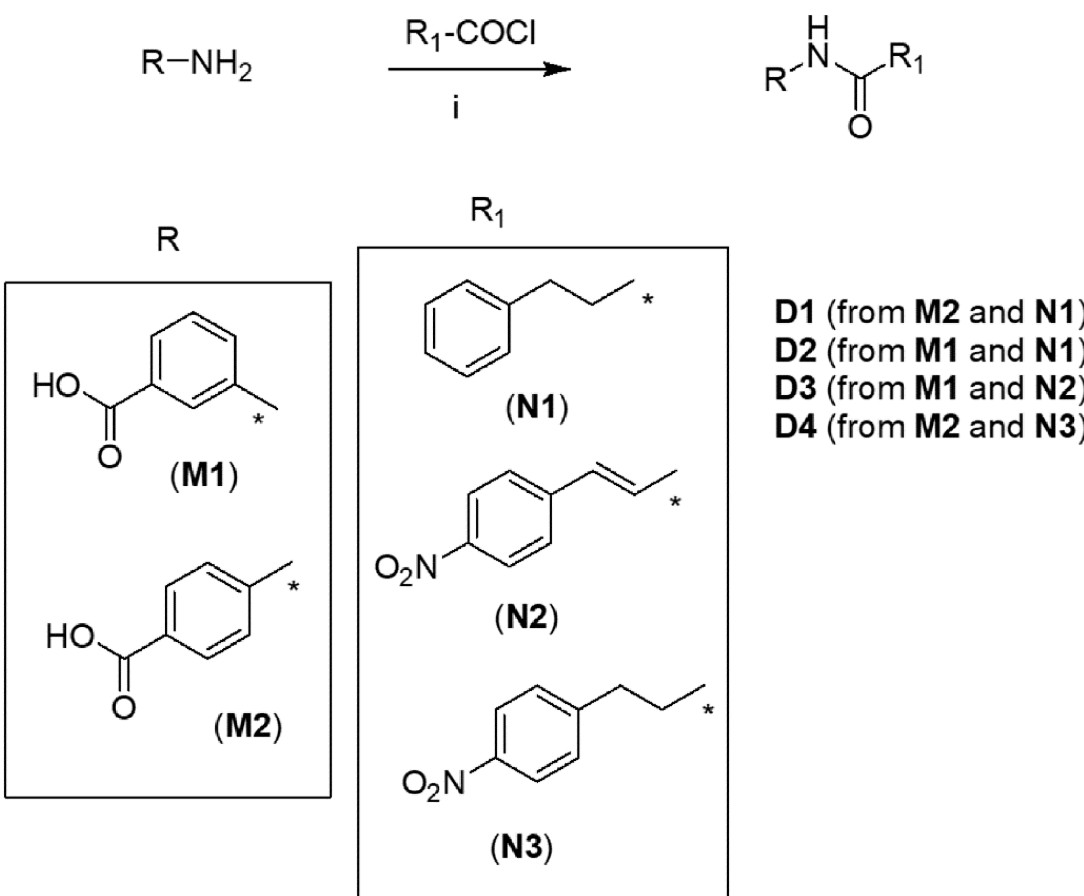

**Appendix 2—figure 5.** Synthesis of compounds D1-D4. (i): TEA/CH₂Cl₂.

General procedure. To a solution of aminobenzoic acid (M1) or (M2) (3.56 mmol) and TEA (6.52 mmol) in anhydrous $CH_2Cl_2$ (20 mL) was added, during 10 min, a solution of the opportune acyl chloride (N1)-(N3) (2.96 mmol) in anhydrous $CH_2Cl_2$ (6 mL) under stirring elm at 0 °C in a $N_2$ atmosphere. The solution was stirred for 12 hours at RT, then the solvent was removed under reduced pressure, the residue was diluted with water/ice, acidified (HCl 0.5 N) and the precipitate was collected and washed with water.

3-phenyl-N-(4-carboxyphenyl)propionamide (D1).Yield 0.156 g (20%) (CH₃OH), m.p. 240–242°C, ¹H-NMR (DMSO-d₆, 400 MHz) (δ) (ppm): 12.80 (1H, broad s), 10.21 (1H, s), 7.89 (2H, m), 7.70 (2H, m), 7.30–7.15 (5H, m), 2.93 (2H, t, J 8.0), 2.68 (2H, t, J 8.0); ¹³C-NMR (DMSO-d₆, 100 MHz) (δ) (ppm): 171.40, 167.39, 143.70, 141.52, 130.85, 128.80, 128.71, 126.44, 125.40, 118.73, 38.50, 31.00. ESI-HRMS calc. for C₁₆H₁₆NO₃ (M+H⁺) 270.1130, found 270.1127. HPLC-UV/Vis *k:* 11.87. Anal. Calc. for C₁₆H₁₅NO₃, C, 71.36, H, 5.61, N, 5.20; found, C, 71.45, H, 5.65, N, 5.35.

3-phenyl-N-(3-carboxyphenyl)propionamide (D2). Yield 63%, m.p. 214–216°C (CH₃OH), ¹H-NMR (DMSO-d₆, 200 MHz) (δ) (ppm): 12.90 (1H, broad s), 10.05 (1H, s), 8.22 (1H, s), 7.82 (1 H, d, J 8.2), 7.60 (1H, d, J 8.2), 7.40 (1H, t, J 8.2), 7.30–7.10 (5H, m), 2.90 (2H, t, 8.0), 2.50 (2H, t, J 8.0); ¹³C-NMR (DMSO-d₆, 50 MHz) (δ) (ppm): 171.09, 167,62, 141.57, 139.88, 131.73, 129.37, 128.78, 128.69, 126.40, 124.30, 123.60, 120.28, 38.39, 31.20. ESI-HRMS calc. for C₁₆H₁₆NO₃ (M+H⁺) 270.1130, found 270.1124. HPLC-UV/Vis *k:* 11.55. Anal. Calc. for C₁₆H₁₅NO₃: C 71.36, H 5.61, N 5.20, found, C, 71.28, H, 5.58, N, 5.24.

3-(4-Nitrophenyl)-N-(3-carboxyphenyl)prop-2-enamide (D3). Yield 43%, m.p. 295–297°C (CH$_3$OH), $^1$H-NMR (DMSO-d$_6$, 200 MHz) (δ) (ppm): 12.90 (1H, broad s), 10.65 (1H, s), 8.28 (3H, m), 7.90 (3H, m), 7.68 (1H, d, J 8.0), 7.65 (1H, d, J 15.5), 7.45 (1H, t, J 8.0), 6.98 (1H, d, J 15.5); $^{13}$C-NMR (DMSO-d$_6$, 50 MHz) (δ) (ppm): 165.53, 163.48, 148.18, 141.65, 139.70, 138.48, 131.89, 129.58, 129.26, 126.79, 124.87, 124.61, 123.85, 120.52. ESI-HRMS calc. for C$_{16}$H$_{13}$N$_2$O$_5$ (M+H$^+$) 313.0824, found 313.0812. HPLC-UV/Vis $k$: 12.04. Anal. Calc. for C$_{16}$H$_{12}$N$_2$O$_5$: C 61.54, H 3.87, N 8.97, found, C, 61.34, H, 4.00, N, 8.77.

3-(4-Nitrophenyl)-N-(4-carboxyphenyl)propionamide (D4). Yield 0.072 (10%), m.p. 292–294°C (trituration with CH$_2$Cl$_2$), $^1$H-NMR (DMSO-d$_6$, 400 MHz) (δ) (ppm): 12.80 (1H, broad s), 10.40 (1H, broad s), 8.31 (2H, m), 8.02 (2H, m), 7.83 (2H, m), 7.70 (2H, m), 3.21 (2H, m), 2.90 (2H, m); $^{13}$C-NMR (DMSO-d$_6$, 100 MHz) (δ) (ppm): 170.91, 167.36, 150.05, 146.46, 143.58, 130.85, 130.10, 125.48, 123.93, 118.76, 37.56, 30.78; ESI-HRMS calcd for C$_{16}$H$_{15}$N$_2$O$_5$ (M+H$^+$) 315.0981, found 315.0980. HPLC-UV/Vis $k$: 11.60. Anal. Calc. for C$_{16}$H$_{14}$N$_2$O$_5$, C, 61.14, H, 4.49, N, 8.91; found, C, 61.50, H, 4.33, N, 8.89.

## Synthesis of compounds F1-F4

**Appendix 2—figure 6.** Design of compounds F (upper panel). Synthesis of compound F1 (bottom panel). (i): CH$_2$Cl$_2$/pyridine/DMAP, RT; (ii): THF/LiOH 2 M; (iii): THF/DMF, (COCl)$_2$; (iv): morpholine, CH$_2$Cl$_2$/TEA 0 °C.

## Synthesis of compounds F1-F4.

**Appendix 2—figure 7.** Synthesis of compounds F2-F4. (i): THF/triphosgene, r.t.; (ii): DMF/DMAP; (iii): CH$_2$Cl$_2$/pyridine/DMAP; (iv): THF/LiOH 2 M.

Methyl 2-[(quinolin-8-ylsulphonyl)amino]benzoate (F1b): To a solution of methyl anthranylate (0.63 g, 4.18 mmol) in anhydrous CH$_2$Cl$_2$ (10 mL) at RT were added 8-quinolinesulfonyl chloride (F1a) (1.00 g, 4.39 mmol), pyridine (0.99 g, 12.5 mmol), and DMAP (0.03 g, 0.21 mmol) under elm stirring at RT in N$_2$ atmosphere. The mixture was stirred at RT for 12 hours, then the solvent was removed under reduced pressure. Water was added to the residue thus obtained and the mixture was extracted with EtOAc (2x20 mL); the organic phase was dried (Na$_2$SO$_4$) and the solvent removed under reduced pressure. This material was used in the next step without further purification. Yield 1.40 g (98%), m.p. 206–209°C, $^1$H-NMR (DMSO-d$_6$, 200 MHz) (δ) (ppm): 11.05 (1H, broad s), 8.98 (1H, m), 8.60–8.20 (3H, m), 7.80–7.55 (4 H, m), 7.42 (1H, m), 6.95 (1H, m), 3.90 (1H, s); $^{13}$C-NMR (DMSO-d$_6$, 50 MHz) (δ) (ppm): 167.57, 151.99, 139.75, 137.54, 135.33, 134.88, 133.08, 131.41, 128.81, 126.10, 123.28, 123.24, 117.73, 116.57, 53.09.

2-[(Quinolin-8-ylsulphonyl)amino]benzoic acid (F1c): A suspension of (F1b) (1.26 g, 3.67 mmol) and aqueous LiOH 2 M (7.1 mL) in THF (8.5 mL) was stirred at RT for 12 hours. The solution thus obtained was cooled with ice and HCl 0.5 N (20 mL) was added until pH 2. The residue thus obtained was filtered and washed with water. Yield 1.10 g (92%), m.p. 253–259°C (dec.), $^1$H-NMR (DMSO-d$_6$, 200 MHz) (δ) (ppm): 11.50 (1H, broad s), 8.95 (1H, m), 8.60–8.40 (2H, m), 8.25 (1H, d, J 8.2), 7.85–7.50 (4H, m), 7.50–7.30 (1 H, m), 7.0–6.80 (1H, m); $^{13}$C-NMR (DMSO-d$_6$, 50 MHz) (δ) (ppm): 169.36, 151.87, 142.84, 140.19, 137.43, 135.20, 134.74, 134.51, 133.04, 131.82, 128.81, 126.06, 123.18, 122.91, 117.19, 117.09.

1-[[(Quinoline-8-yl-sulfonyl)amino]benzoyl]morpholine (F1): To a solution of (F1c) (0.3 g, 0.91 mmol) in anhydrous THF (5 mL) and DMF (0.01 mL) at 0 °C, oxalylchloride (0.13 g, 1.35 mmol) was added dropwise, then the mixture was stirred for 1 hour at RT. At the end of the reaction, the solvent was removed under reduced pressure. The solid thus obtained was coevaporated with anhydrous CH$_2$Cl$_2$ three times, and the compound (F1d) thus obtained was used in the next step without purification. To a solution of (F1d) (0.32 g, 0.91 mmol) and TEA (0.23 g, 2.28 mmol) in anhydrous CH$_2$Cl$_2$ (20 mL) morpholine (0.07 g, 0.82 mmol) was added and the solution thus obtained was stirred at RT for 6 hours. Then the solvent was removed under pressure and the residue thus obtained was purified by column chromatography (EtOAc 100%). Yield 0.099 g (31%), m.p. 191–194°C, $^1$H-NMR (DMSO-d$_6$, 400 MHz) (δ) (ppm): 9.50 (1H, broad s), 9.25 (1H, d, J 2.9), 8.75 (1H, d, J 7.9), 8.45 (2H, m), 7.91 (2H, m), 7.63 (1H, m), 7.48 (1H, m), 7.27 (2H, m), 3.8–3.0 (8H, m); $^{13}$C-NMR (DMSO-d$_6$, 100 MHz) (δ) (ppm): 167.39, 152.16, 142.85, 137.76, 135.59, 134.78, 131.34, 130.89, 128.96, 128.49, 127.47, 126.27, 124.99, 123.94, 123.34, 66.04, 65.38, 47.60, 42.30, 15.63; ESI-HRMS calcd for C$_{20}$H$_{20}$N$_3$O$_4$S (M+H$^+$) 398.1174, found 398.1189. HPLC-UV/Vis k: 10.97.

Substituted (isatoic anhydrides) (1b) and (2b). General procedure. To a solution of the appropriate (substituted)–2-aminobenzoic acid (1 a, 2 a) (5.10 mmol) in anhydrous THF (25 mL), triphosgene (2.20 mmol) was added at RT under stirring elm. The suspension thus obtained was stirred for

24 hours; then, water was added and the solution was extracted with EtOAc (2x50 mL); the organic phase was dried ($Na_2SO_4$) and the solvent removed under reduced pressure.

5,6-Dimethoxy isatoic anhydride (1b): Yield 1.03 g (91%), m.p. 260–261°C (dec.), $^1$H-NMR (DMSO-$d_6$, 200 MHz) (δ) (ppm): 11.5 (1H, broad s), 7.20 (1H, s), 6.60 (1H, s), 3.80 (6H, s); $^{13}$C-NMR (DMSO-$d_6$, 50 MHz) (δ) (ppm): 159.78, 157.01, 147.83, 146.13, 138.06, 109.06, 101.62, 98.14, 56.53, 56.34.

5-Methoxycarbonyl isatoic anhydride (2b): Yield 1.00 g (88%), m.p. 192–194°C, $^1$H-NMR (DMSO-$d_6$, 200 MHz) (δ) (ppm): 11.85 (1H, broad s), 8.0 (1H, d, $J$ 8.1), 7.70 (2H, m), 3.9 (3H, s); $^{13}$C-NMR (DMSO-$d_6$, 50 MHz) (δ) (ppm): 165.31, 147.28, 141.94, 136.86, 130.03, 123.64, 116.39, 114.37, 53.31.

Substituted quinolines (1 c) and (2 c). General procedure. To a suspension of the appropriate (substituted)isatoic anhydride (**1b**) or (**2b**) (4.6 mmol) in anhydrous DMF (30 mL) morpholine (4.6 mmol) and DMAP (0.46 mmol) were added under stirring at RT in a $N_2$ atmosphere. The solution was stirred for 8 hours at RT, then, water was added to the reaction mixture and the product was extracted with EtOAc, the organic phase was dried ($Na_2SO_4$) and the solvent removed under reduced pressure.

1-(2-Amino-4,5-dimethoxybenzoyl)morpholine (1 c): Yield 0.36 g (30%), m.p. 100–102°C, $^1$H-NMR (DMSO-$d_6$, 200 MHz) (δ) (ppm): 6.60 (1H, s), 6.35 (1H, s), 4.95 (2H, s), 3.70 (3H, s), 3.65 (3H, s), 3.58 (4H, m), 3.45 (4H, m);$^{13}$C-NMR (DMSO-$d_6$, 50 MHz) (δ) (ppm): 168.00, 151.62, 142.04, 113.53, 100.76, 66.63, 56.93, 55.66.

1-[(2-Amino-4-methoxycarbonyl)benzoyl]morpholine (2 c): Yield 0.65 g (54%), m.p. 112–113°C, $^1$H-NMR (DMSO-$d_6$, 200 MHz) (δ) (ppm): 7.35 (1H, s), 7.10 (2H, s), 5.45 (2H, s), 3.80 (3H, s), 3.7–3.3 (8H, m); $^{13}$C-NMR (DMSO-$d_6$, 50 MHz) (δ) (ppm): 168.04, 166.66, 146.07, 131.36, 128.47, 123.83, 116.46, 116.42, 66.53, 52.48.

1-[[(Quinoline-8-yl-sulfonyl)amino]–3,4-dimethoxybenzoyl]morpholine (F2): To a solution of (1 c) (0.33 g, 1.25 mmol) in anhydrous $CH_2Cl_2$ (10 mL) under stirring at RT, 8-quinolinesulfonylchloride (0.28 g, 1.26 mmol), pyridine (0.30 g, 3.75 mmol) and DMAP (8 mg, 0.06 mmol) were added. The mixture was stirred at RT for 12 hours, then the solvent was removed under pressure; water and ice were added to the oily residue, and the solid thus formed was filtered, washed with water, dried and purified by trituration with anh. diethyl ether (4 mL). Yield 0.57 g 100%, m.p. 166–168°C, $^1$H-NMR (DMSO-$d_6$, 400 MHz) (δ) (ppm): 9.42 (1H, broad s), 9.25 (1H, m), 8.82–8.68 (1H, m), 8.55–8.33 (2 H, m), 8.02–7.78 (2H, m), 7.15 (1H, m), 6.80 (1H, m), 3.80 (6H, m), 3.70–2.70 (8H, m); $^{13}$C-NMR (DMSO-$d_6$, 100 MHz) (δ) (ppm): 167.26, 152.14, 150.60, 145.99, 142.88, 137.64, 135.46, 134.74, 131.60, 129.10, 128.89, 126.26, 123.34, 119.89, 111.37, 108.68, 81.154, 65.92, 56.194, 55.93; ESI-HRMS calcd for $C_{22}H_{24}N_3O_6S$ (M+H$^+$) 458.1386, found 458.1386. HPLC-UV/Vis $k$: 10.54.

Methyl 3-[(quinoline-8-yl-sulfonyl)amino]–4-[(morpoholin-1-yl)oxo] benzoate (F3): To a solution of (2 c) (0.59 g, 2.22 mmol) in anhydrous $CH_2Cl_2$ (15 mL) at RT under stirring elm, were added 8-quinolinesulfonylchloride (0.56 g, 2.44 mmol), pyridine (0.53 g, 6.66 mmol) and DMAP (0.014 g, 0.11 mmol). The solution was stirred at RT for 12 hours, then the solvent was removed under reduced pressure. A mixture of water and ice (10 mL), then EtOAc (4 mL) was added to the oily residue; the precipitate thus obtained was filtered, washed with water, dried and purified by trituration with diethyl ether. Yield 1.00 g (99%), m.p. 174–176°C, $^1$H-NMR (DMSO-$d_6$, 400 MHz) (δ) (ppm): 9.65 (1H, broad s), 9.26 (1H, dd, $J$ 4.3, $J$ 1.8), 8.77 (1H, dd, $J$ 8.4, $J$ 1.4), 8.50 (1H, m), 8.43 (1H, dd, $J$ 7.3, $J$ 1.2), 8.19 (1H, d, $J$ 1.4), 7.95 (1H, m), 7.88 (1H, m), 7.82 (1H, dd, $J$ 7.9, $J$ 1.6), 7.59 (1H, d, $J$ 8.1), 4.00 (3H, s), 3.70–3.2 (8H, m); $^{13}$C-NMR (DMSO-$d_6$, 100 MHz) (δ) (ppm): 166.24, 165.55, 152.21, 142.79, 137.80, 135.65, 135.58, 134.97, 132.24, 131.59, 131.25, 128.99, 126.31, 125.71, 124.87, 123.44; ESI-HRMS calcd for $C_{22}H_{22}N_3O_6S$ (M+H$^+$) 456.1229, found 456.1227. HPLC-UV/Vis $k$: 11.37.

1-[[(Quinoline-8-yl-sulfonyl)amino]–4-carboxybenzoyl]morpholine (F4): A suspension of (**2 c**) (0.25 g, 0.59 mmol) in THF (3 mL) and aqueous LiOH 2 M (1.15 mL) was stirred at RT for 12 hours. The solution thus obtained was cooled with ice and HCl 0.5 N (20 mL) was added until pH 2.0. The solid thus obtained was filtered and washed with water. Yield 0.22 g (91%), m.p. 255–258°C, $^1$H-NMR (DMSO-$d_6$, 400 MHz) (δ) (ppm): 13.40 (1H, broad, s), 9.60 (1H, broad, s), 9.25 (1H, dd, $J$ 4.3, $J$ 1.8), 8.76 (1H, dd, $J$ 8.4, $J$ 1.6), 8.49 (1H, dd, 8.4, $J$ 1.2), 8.43 (1H, dd, $J$ 7.3, $J$ 1.4), 8.18 (1H, d, $J$ 1.4), 7.95 (1H, m), 7.92–7.85 (1H, m), 7.83–7.77 (1 H, dd, $J$ 7.9, $J$ 1.6), 7.46 (1H, d, $J$ 7.9), 3.70–3.10 (8H, m); $^{13}$C-NMR (DMSO-$d_6$, 100 MHz) (δ) (ppm): 166.56, 166.43, 152.20, 142.79, 137.78, 135.60, 135.57,

134.93, 132.88, 131.74, 131.23, 128.70, 128.83, 126.03, 125.79, 124.90, 123.42, 66.00; ESI-HRMS calcd for $C_{21}H_{20}N_3O_6S$ (M+H$^+$) 442.1073, found 442.1068. HPLC-UV/Vis $k$: 9.85.

## Synthesis of E5-($O_2Oc$)-CAM and E5-($O_2Oc$)-CAM-FITC

### Synthesis of E5-(O 2 Oc)-CAM and E5-(O 2 Oc)-CAM-FITC.

**E5-O₂Oc-CAM**

**E5-O₂Oc-CAM-FITC**

**Appendix 2—figure 8.** Synthesis of compounds E5-$O_2Oc$-CAM and E5-$O_2Oc$-CAM-FITC. (i) Cysteamine 2-chlorotrityl resin (1 eq.), Fmoc-O2Oc-OH (4 eq.), DIC (4 eq.), HOBt (4 eq.), DMF, r.t., 30 min; (ii) 40% piperidine in DMF, r.t., 5 min; (iii) E3 (2 eq.), HATU (3 eq.) DIPEA (2 eq.), DMF, r.t., 3 h; (iv) trifluoroacetic acid (TFA)/$H_2O$/ phenol/triisopropylsilane 88:5:5:2 v/v (10 mL / 0.2 g of resin); r.t., 30 min; (v) E5-($O_2Oc$)-CAM (1 eq.), Fluorescein-5-Maleimide (0.06 eq.), $CH_3CN/H2O$, r.t., 5 min.

## Synthesis of E5-($O_2Oc$)-CAM

The Cysteamine (CAM) 2-chlorotrityl resin (0.079 mmol) was swelled in DMF and suspended in a solution of Fmoc-amino-3,6 dioxaoctanoic acid (Fmoc-$O_2Oc$-OH) (0.316 mmol), N,N'-Diisopropylcarbodiimide (DIC) (0.316 mmol) and hydroxybenzotriazole (HOBT) (0.316 mmol) in DMF (2 mL). This suspension was shaken for 30 minutes. After this time the solid phase was filtered and washed with DMF (3x2 mL). The polymer was successively suspended in a 40% solution of piperidine in DMF (3x2 mL) and shaken for 5 minutes. The resin was filtered and washed with DMF (3x2 mL). In the final step the solid support was suspended in a solution of **E3,** (0.158 mmol), (1-[Bis(dimethylamino) methylene]–1 H-1,2,3-triazolo[4,5-b]pyridinium 3-oxid hexafluorophosphate (HATU)) (0.237 mmol) and diisopropylethylamine (DIPEA) (0.237 mmol) in DMF and shaken for 3 hours. The functionalized resin was washed with DMF (3x2 mL) and the desired molecule was released after a treatment with a mixture of trifluoroacetic acid (TFA)/$H_2O$/phenol/triisopropylsilane 88:5:5:2 v/v (10 mL per 0.2 g of resin) for 30 minutes at room temperature. After filtration of the resin, the solvent was concentrated in vacuo and the residue triturated with ethyl ether. The crude of reaction was purified by preparative HPLC giving, after lyophilization, the E3-($O_2Oc$)-CAM as a pale yellow amorphous solid (49% yield). ESI-HRMS calcd for $C_{29}H_{33}N_4O_7S_2^+$ (M+H$^+$): 613.1785 (*Appendix 2—figure 9*); found: 613.7175.

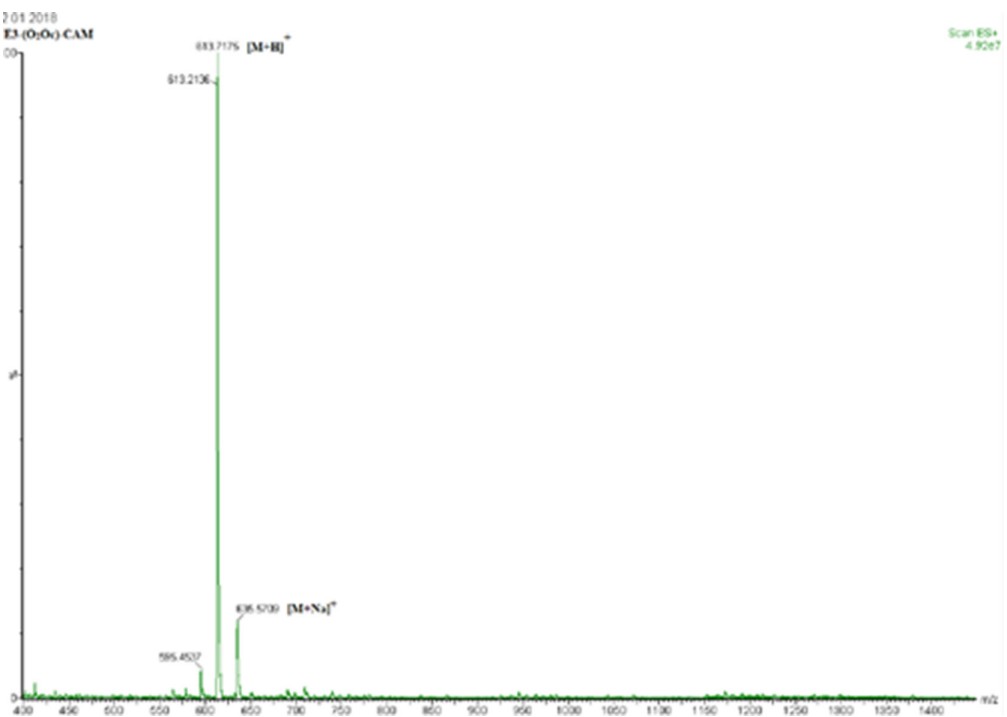

**Appendix 2—figure 9.** Mass spectrum of E5-(O$_2$Oc)-CAM ESI-HRMS calcd for C$_{29}$H$_{33}$N$_4$O$_7$S$_2^+$ (M+H$^+$): 613.1785.

## Synthesis of E5-(O$_2$Oc)-CAM-FITC

To a stirred solution of **E5-(O$_2$Oc)-CAM** (0.0277 mmol) in CH$_3$CN (0,5 mL), Fluorescein-5-Maleimide (FITC) (0.00156 mmol) solubilized in CH$_3$CN/H2O (250 µL/250 µL) was added followed by 20 µL of 5% solution of NaHCO$_3$. The reaction was stirred in the dark and the MS monitoring showed a complete consumption of the fluorescent probe in about 5 minutes. After this time the reaction solution the product was purified directly by preparative HPLC using the same experimental conditions as for the purification of E3-(O$_2$Oc)-CAM. After lyophilization in the dark, a yellow amorphous solid was obtained in quantitative yield and with a purity grade >95%. ESI-HRMS calcd for C$_{53}$H$_{46}$N$_5$O$_{14}$S$_2^+$ (M+H$^+$): 1040.2477 (*Appendix 2—figure 10*); found: 1040.3098.

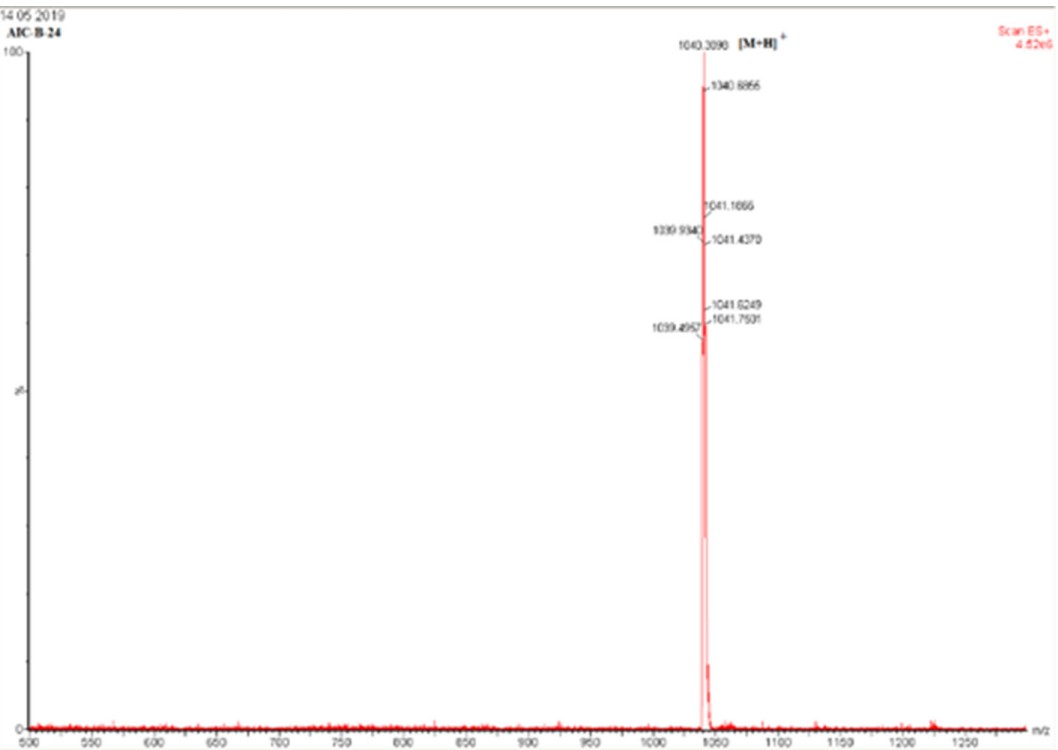

**Appendix 2—figure 10.** Mass spectrum of E5-(O$_2$Oc)-CAM-FITC. ESI-HRMS calcd for C$_{53}$H$_{46}$N$_5$O$_{14}$S$_2^+$ (M+H$^+$):1040.2477.

## Appendix 3

### Enantioseparation and racemization of E7

#### Materials

The columns used were Chiralcel OD-RH [cellulose tris (3,5-dimethylphenylcarbamate); 250×4.6 mm I.D.; 5 µm], Chiralcel OD [cellulose tris (3,5-dimethylphenylcarbamate); 250×10 mm I.D., 10 µm] purchased from Chiral Technologies Europe, Illkirch, France. HPLC analyses were performed on an Agilent Technologies (Waldbronn, Germany) modular model 1200 system, consisting of a vacuum degasser, a binary pump, an autosampler, a thermostatted column compartment and a variable ultraviolet detector (UV). Chromatograms were acquired at 254 nm. HPLC-grade acetonitrile and water were obtained from Baker.

#### Analytical and semipreparative enantioseparation

The analytical enantioseparation of chiral **E7** was performed in reversed phase using a Chiracel OD-RH column with mobile phases of water:acetonitrile mixtures at different percentages. The separation was carried out isocratically at 25 °C. The racemic compound mixture was dissolved in acetonitrile and subsequently diluted 1:10 (v/v) with the mobile phase at a final concentration of 100 µg/mL. The injection volume was 6 µL. The detector was set at 254 nm. The retention factors ($k_1$ and $k_2$) were calculated as $k_1 = (t_1 − t_0)/t_0$ where $t_1$ and $t_2$ are the retention times of the first and second eluted enantiomers and the separation factor ($\alpha$) as $k_2/k_1$. The resolution factor ($R_s$) was calculated as $R_s = 2(t_2−t_1)/(w_1+w_2)$, where $w_1$ and $w_2$ are the peak widths at the base for the two eluted enantiomers. The dead time of the columns ($t_0$) was determined by injection of 1,3,5-tri-*tert*-butylbenzene. Among the mobile phases used, water:acetonitrile 2:98 (v/v) was chosen for the subsequent semipreparative separation because it gave a good baseline enantiomeric resolution, with $k_1$=0.36, $k_2$=0.62 and $R_s$ = 4.76. Semipreparative enantioseparation of (±)-**E7** was carried out on Chiralcel OD employing water:acetonitrile 2:98 (v/v) as the mobile phase at 5 mL/min. By injecting 15 mg of the racemic mixture, the single enantiomers were collected with acceptable enantiomeric excess (99.0% and 80.1% e.e. for the first and second eluted enantiomer, respectively, (*Supplementary file 5*, *Figures 1 and 2*)). 8.8 mg of the first enantiomer and 4.4 mg of the second one was obtained and were sufficient for further stability studies. The specific rotation in acetonitrile of first and second eluted enantiomers were acquired on a Perkin-Elmer 241 Polarimeter resulting in $[\alpha]_D^{20}$ = +7.95° (c 0.00088 g/ml; acetonitrile) and $[\alpha]_D^{20}$ = -9.10° (0.00044 g/mL, acetonitrile), respectively.

#### Racemization

Racemization is the conversion of an enantiomer into a racemic mixture and is associated with a loss in optical activity of the chiral compound over time. The rate constant of racemization $k_{rac}$ refers to this first-order kinetic process and can be calculated from the linear plot of the logarithmic values of the enantiomeric excess (ln (e.e.)) *versus* time, according to Eq. (S1):

$$\text{In (e.e.)} = k_{rac} \cdot t \qquad \text{(S1)}$$

The half-life of racemization ($t_{1/2rac}$) is the time during which the enantiomeric purity of a chiral compound reduces to 50% of its original e.e. For a first-order racemization, the half-life can be calculated by Eq. (S2):

$$t_{1/2rac} = ln2/k_{rac} \qquad \text{(S2)}$$

Single enantiomers of (±)-**E7** were dissolved in acetonitrile [1 mg/ml] and then diluted 1:100 with buffer solution at final concentration of [0.1 µg/mL]. The solution was thermostated at 37.5 °C and racemization was monitored by injecting the solution at fixed times on a Chiracel OD-RH column using water and acetonitrile 2:98 (v/v) as mobile phase. Four repeats of each experiment were carried out.

#### Racemization kinetics

The previously obtained single enantiomers of (±)-**E7** were solubilized in buffer solutions to simulate in vitro test conditions and then incubated for a set of times at 37.5 °C. After the selected times, they were injected in a Chiracel OD-RH column (water:acetonitrile 2:98 (v/v)) to evaluate racemization rates. The data were analyzed. The kinetic parameters were calculated from the corresponding peak areas according to Eqs. S1 and S2.

## Enantiomer and racemate inhibition of hTS

The capability of (±) E7 compound to inhibit the kinetic activity of hTS protein enzyme was spectrophotometrically evaluated using the JASCO V730 UV/Vis spectrophotometer. The reaction was monitored by following the absorption change of MTHF substrate at 340 nm for 180 seconds.

## Inhibition Potency Evaluation of (+)-**E7** compound

The (**+)-E7** compound does not show activity at time zero (% of enzyme inhibition below 10%) at the two 10 and 20 uM; after one hour of incubation with hTS, (**+)-E7** is able to inhibit the enzyme activity at the 34% and 39% at 10 and 20 uM respectively with respect to the non-inhibited enzyme. (**-)-E7** enantiomer shows the same potency of (**+)-E7** compound at time zero (30%–48% at 10–20 uM) and, after 1 hour, it doubles the efficacy to 51 and 67% at the same assayed concentrations. Regarding racemic **E7**, the compound results to be active just at time zero (26% and 40%, 10–20 uM) and it also maintains unaltered the efficacy after 1 hour of incubation (25%–52%, 10–20 uM). We did not analyze the data further at this stage.

## Racemization rate constants of (+)-E7 and (-)-E7 determined in buffer solutions

Time intervals for enantiomerization kinetics analysis: 0', 10', 20', 30', 40' and 50'. T=37.5 °C.

|  | (+)-E7 | (-)-E7 |
|---|---|---|
| $k$ (s$^{-1}$) | $2.03\pm0.18 \times 10^{-4}$ | $2.10\pm0.16 \times 10^{-4}$ |
| $t_{1/2}$ (min) | 56.80 | 54.98 |

## Enantioseparation of E7

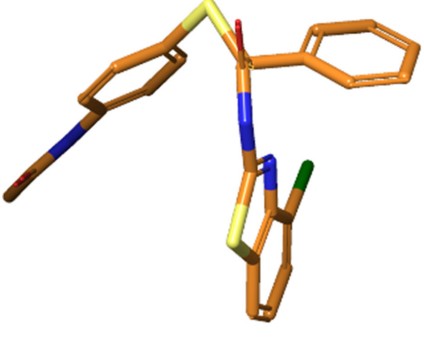
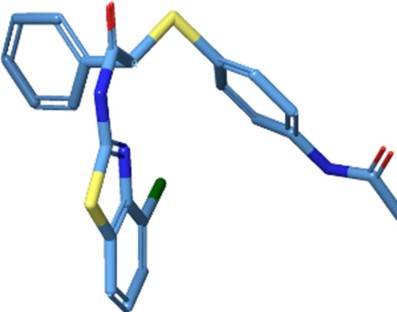

**S isomer**          **R isomer**

**Appendix 3—figure 1.** Structure 2D of the E7 enantiomers. (**A**) Representation of S and R isomers of compound E7 obtained with (**B**) Structural modification of compound B12. The changes in chemical structure and the associated IC50s for recombinant-enzyme inhibition, DFFRET, abilities to dissociate the hTS dimers in cells, cellular growth inhibition and in vivo data of the key compounds C3, E5 and E7 of the sub-series are reported.

## Chromatograms of single (+/-)-E7 enantiomers

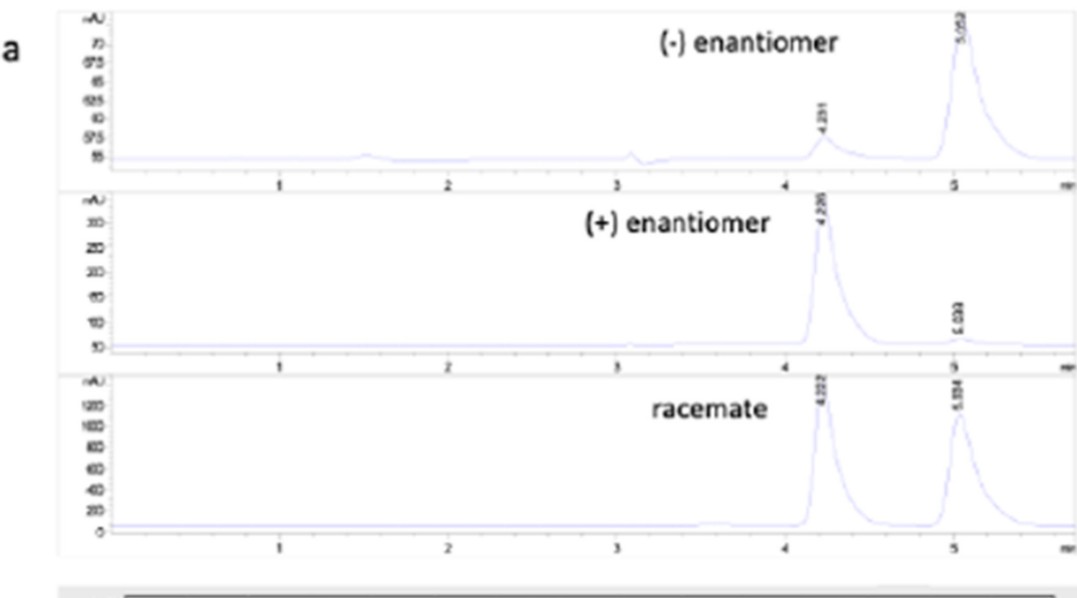

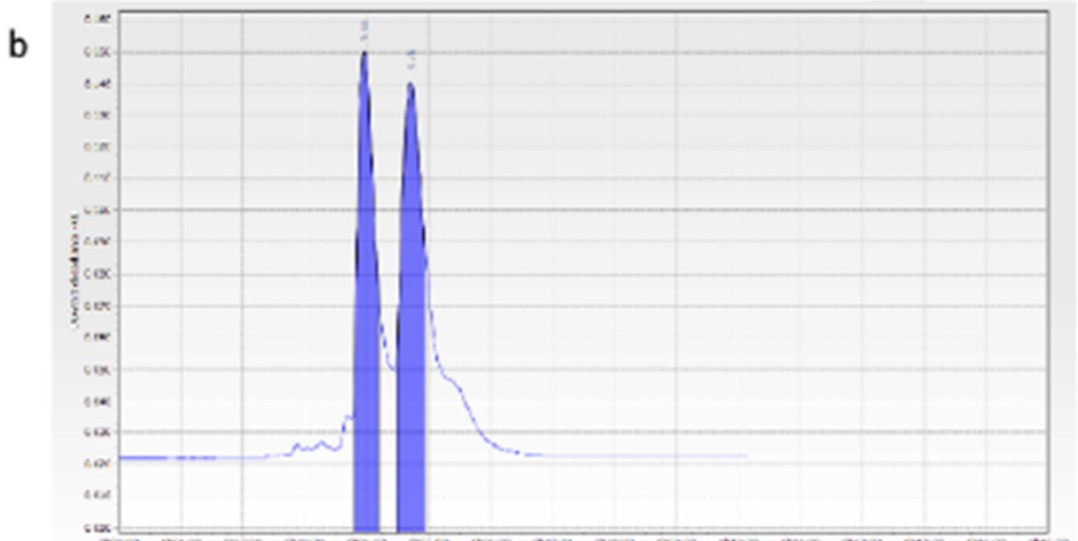

**Appendix 3—figure 2.** Chromatograms of single (+/-)-E7 enantiomers. Column: Chiracel OD(250x4.6 mm I.D., 5 mm), mobile phase: water:acetonitrile 2:98 (v/v), flow rate:1 mL/min; l=254 nm. (**b**) Semipreparative enentiseparation of (+/-)-E7. Column: Chirasphere OD, mobilephase water:acetonitrile 2:98 (v/v), flow rate:5 mL/min, l:254 nm, room temperature.

## Appendix 4

### Analysis of combined equilibria and fitting to FRET data

With the aim of extracting quantitative mechanistic data from the dose dependencies of the $\Phi_{FRET}$ data, we have written and solved the equilibrium scheme in *Figure 4C* of the main text. $E_T$ is the total concentration of enzyme dimers:

$$E_T = \frac{K'_D K''_1}{2K''_D[1]}[MI] + \frac{K''_1}{K''_D[1]}[MI]^2 + \frac{[MI]}{2} + \frac{K'_D 2K''_1 2}{K_D K''_D 2}\frac{[MI]^2}{[I]^2} + \frac{[MI]^2}{K''_D}$$

where M represents an enzyme monomer and I an inhibitor molecule. $\Phi$, the FRET efficiency, is near one for $M_2$; if we assume the same for $M_2I$ and $M_2I_2$, i.e., that binding of the inhibitor does not cause a structural change able to significantly reduce the efficiency of FRET, then the observed efficiency, $\Phi_{FRET}$, will coincide with the total fraction of dimeric protein:

$$\varnothing_{FRET} = \frac{[M_2]+[M_2I]+[M_2I_2]}{E_T} = \frac{[E_T]-\frac{1}{2}([M]+[MI])}{E_T} = 1 - \frac{([M]+[MI])}{2[E_T]}$$

Our goal is to express [M] and [MI] in terms of $E_T$, $I_T$ (the total inhibitor concentration) and the pertinent equilibrium constants.

$$[I_T] = [I] + [MI] + [M_2I] + 2[M_2I_2]$$

$$K_D = \frac{[M]^2}{[M_2]} \quad K'_D = \frac{[M][MI]}{[M_2I]} \quad K''_D = \frac{[MI]^2}{[M_2I_2]} \quad K_1 = \frac{[I][M]}{[MI]} \quad K'_1 = \frac{[I][M_2]}{[M_2I]} \quad K''_1 = \frac{[I][M_2I]}{[M_2I_2]}$$

Because $I_T \gg E_T$, we simplify $[I] \approx I_T$, and rewrite:

$$\varnothing_{FRET} = 1 - \frac{([M]+[MI])}{2E_T} = 1 - \frac{1}{2E_T}\left(\frac{K_i}{[I]}+1\right)[MI] = 1 - \frac{1}{2E_T}\left(\frac{K_i}{I_T}+1\right)[MI]$$

[MI] is calculated from the positive root of the quadratic equation obtained by expressing each term contributing to $E_T$ in terms of [MI], [I] and the Ks, and replacing $[I] \approx I_T$:

$$\left(\frac{1}{K''_D} + \frac{K''_I}{K''_D I_T} + \frac{K'^2_D K''^2_I}{K_D K''^2_D I_T^2}\right)[MI]^2 + \left(\frac{1}{2} + \frac{K'_D K''_I}{2K''_D I_T}\right)[MI] - E_T = \alpha[MI]^2 + 2b[MI] - E_T = 0$$

$$\varnothing_{FRET} = 1 - \frac{1}{2E_T}\left(\frac{K_i}{I_T}+1\right)\left(\frac{-b \pm \sqrt{b^2 + \alpha E_T}}{a}\right) \tag{S3}$$

The dependence of the observed $\phi_{FRET}$ on $I_T$ for the three inhibitors was fitted to eq. S3 (*Figure 4B* in the main text). To narrow the amount of fitting parameters, $K_D$ was kept fixed at the value obtained from FRET measurements without inhibitors ($6\times10^{-8}$ M) and $K'_D$ was estimated by assuming that the effect of each inhibitor molecule on the standard free energy for the enzyme dimer dissociation was constant: $\Delta G°'_D = (\Delta G°_D + \Delta G°''_D)/2$. From the above value of $K_D$ and the $K''_D$ obtained from the limit values of $\phi_{obs}$ at high [C3], $K''_D = 3.3 \times 10^{-7}$ M (paragraph Mechanistic analysis of dissociative inhibition, main text), we estimated $K'_D = 1.5 \times 10^{-7}$ M and used this value in the fitting procedure. $K''_D$ values were obtained from the high $I_T$ behavior of $\phi_{obs}$, while $K_i$ and $K''_I$ were adjusted to fit the low and intermediate $I_T$ data.

### Analysis of a dissociative inhibition mechanism and fitting to data

Here, to test the suitability of the proposed mechanism to fit the experimental inhibition results, rather than searching for linear plots, we aim at expressing the dependence of the observed reaction rate on the total substrate concentration, $S_T$, at different total inhibitor, $I_T$, and total enzyme, $E_T$, concentrations.

$$v = k[M_2][S] + k'[M_2I][S] + k''[M_2I_2][S] = k\frac{[M_2][S]}{K_1} + k'\frac{[M_2I][S]}{K_2} + k''\frac{[M_2I_2][S]}{K_3} =$$

$$= k\frac{[M_2][S]}{K_1} + k'\frac{[M_2][I][S]}{K_2 K'_I} + k''\frac{[M_2][I]^2[S]}{K_3 K'_I K''_I}$$

So, $v = [M_2][S]D$ with

$$D = \left( \frac{K}{K_I} + \frac{k'[I]}{K_2 K'_I} + \frac{K''[I]^2}{K_3 K'_I K''_I} \right)$$

We now express $E_T$ in terms of $v$, $[I]$ and $[S]$:

$$E_T = \frac{[M]}{2} + \frac{[M]}{2} + [M_2] + [M_2I] + [M_2I_2] + [M_2S] + [M_2IS] + [M_2I_2S]$$

$$= \frac{1}{2}(K_D[M_2])^{1/2}(1 + \frac{[I]}{K_I}) + [M_2](1 + \frac{[I]}{K'_I} + \frac{[I]^2}{K'_I K''_I}) + [M_2][S](\frac{1}{K_I} + \frac{[I]}{K_2 K'_I} + \frac{[I]^2}{K_3 K'_I K''_I})$$

Because $S_T$, $I_T \gg E_T$, we approximate $[S]=S_T$ and $[I]=I_T$. Calling 2a, b and c the coefficients that multiply $[M_2]^{1/2}$, $[M_2]$ and $[M_2][S]$ in the three terms and writing $x = [M_2]^{1/2}$, we obtain the second-degree equation:

$$(b + cS_T)x^2 + 2ax - E_T = 0$$

whose positive root, $x_r$, yields the desired value of $v$: $v = x_r^2 S_T D$.

So, given the experimental values of $E_T$, $S_T$ and $I_T$, we can calculate $v$ using the equilibrium and rate constants in the scheme in ***Figure 4C*** of the main text. In the fitting analysis, we kept $K_D$ fixed at $6 \times 10^{-8}$ M, as obtained from dilution experiments in the absence of inhibitors. We also kept the equilibrium constant for the binding of mTHF to hTS saturated with dUMP, $K_1$, fixed to the value obtained in the absence of inhibitors, $10^{-5}$ M. Also, $K_I$ was taken equal to $10^{-6}$ M for the three inhibitors from the results of the fitting of FRET data. The other equilibrium constants were explored subject to the conditions imposed by the combined equilibria in ***Figure 4C*** of the main text, i.e.: $K_D/K_I = K'_D/K'_I$ and $K'_D/K_I = K''_D/K''_I$. So, $K'_I$ and $K''_I$ were kept one order of magnitude larger than $K_I$, i.e., $10^{-5}$ M. Finally, the catalytic rate constant, $k$, was fixed at its value with no inhibitor, 0.9 s$^{-1}$; $k'$ and $k''$ were variable fitting parameters.

