## [Editor Report]

Drugs that therapeutically target human thymidylate synthase are widely used to treat cancer. In this manuscript, the authors have discovered and validated an alternative strategy to target this enzyme using dimer destabilizers. By disrupting homodimerization of these proteins using a series of biophysical assays, this research team identified an analog E7 that inhibits hTS in cancer cells and demonstrates a superior profile to approved drugs targeting this pathway. The work is innovative and novel and could lead to the development of new classes of drugs targeting this key enzyme for cancer therapy.

---

## [Decision Letter]

**Decision letter after peer review:**

Thank you for submitting your article "Disrupters of the thymidylate synthase homodimer accelerate its proteasomal degradation and inhibit cancer growth" for consideration by *eLife*. Your article has been reviewed by 2 peer reviewers, and the evaluation has been overseen by a Reviewing Editor and Erica Golemis as the Senior Editor. The following individual involved in review of your submission has agreed to reveal their identity: Yatrik M Shah (Reviewer #1).

Essential revisions:

The identification of novel chemical inhibitors of thymidylate synthase has the potential to have a significant impact on cancer treatment. The work was viewed as impactful and novel overall. There are a few concerns that need to be addressed prior to this manuscript being suitable for publication including

1. Addressing the issue of the utility of these inhibitors compared to 5-FU.

2. Addressing the degree to which the compounds bind to and physically disrupt TS dimers. This biophysical data is critical to the central conclusions of this manuscript.

See below for specific reviewer comments.

*Reviewer #1 (Recommendations for the authors):*

Thymidylate synthase (TS) is a homodimeric enzyme that is a common anti-cancer target. Currently used TS inhibitors bind to the enzymatic active site which leads to an increase protein expression and subsequent resistance that is observed. The present manuscript through a chemical screening and modeling approach identifies novel inhibitors of TS that bind to the homodimerization interface. These compounds are distinct from the commonly used TS inhibitors. The authors show that their novel compounds do indeed bind, lead to destabilization of protein and decreased cell growth. The work is highly significant and very robustly done.

Only a few concerns revolving around the utility of the inhibitors compared to 5-FU are required to address this reviewer's concern:

– In the panel of cell lines assessing the efficacy of their novel inhibitors in vitro, comparisons to 5-FU should be shown.

– The major concern is that the effects of E7 are not dramatically better than 5-FU, and in most cases are the same or even worse. Please assess if E7 can inhibit viability in cells that are resistant to 5-FU in vitro and in vivo.

– Data on cell death and proliferation are needed for the in vivo tumor studies.

– The discussion needs to focus on the novel finds and clinical impact, currently it is highly repetitive to the result section. Along these lines it would be good to discuss mechanisms of 5-FU resistance and under what scenarios would the dimerization inhibitors be used.

*Reviewer #2 (Recommendations for the authors):*

Conventional thymidylate synthase (TS) inhibitor drugs bind to the active site or to the nearby folate cofactor site. Either mechanism leads to elevated TS expression, along with drug resistance, presumably by stabilizing the inactive TS dimer. The authors come up with an alternative strategy intended to disrupt TS dimerization, which would lead to accelerated degradation of the less stable monomeric protein.

For the proposed strategy, the study identifies a putative drug binding pocket at the TS protein-protein interface that is predicted to disrupt the homodimer. The authors then introduce a cysteine residue in this pocket which serves as the target of a cysteine tethering approach to implement fragment-based drug design. After a series of fragment-based and virtual docking screens, a series of molecules and derivatives are identified for testing. While the cysteine tethering approach is a good first step to identify putative small molecules, it is not certain that the compounds are binding preferentially to a drug "pocket" or simply forming a covalent bond with the introduced cysteine. This is an important consideration when examining the mechanism of action for subsequent derivatives in the context of the native enzyme.

The analogs were experimentally tested against TS activity and for an ability to disrupt dimerization. Effects on TS dimer disruption was assayed using a FRET approach and TS expressed with either a red- or green-emitting tag. It is concluded that some of these molecules disrupt TS dimerization. However, a key concern regarding the proposed mechanism of action, is that data supporting drug-protein interactions and TS dimer dissociation are somewhat indirect. Since a primary focus of the investigation centers around the proposed mechanism of action of the new compounds, the study would be enhanced with additional structural and/or biophysical data to support binding of the individual molecules to TS directly, and the subsequent dissociation of TS dimers.

The comprehensive investigation then examines the stability of TS in treated cells and with proteasome inhibitors. The data reveal that administration of compounds correlates with enhanced degradation of TS and the authors hypothesize that this is due to the TS monomer, disrupted by E7 administration, is less stable than the functional dimer. A comparison of treatments of conventional TS inhibitors, such as 5-FU and Pemetrexed, support a different mechanism of action as TS levels are elevated following administration of these established compounds. Finally, the investigation identifies the effects of the lead compound, E7, in mouse models of pancreatic and ovarian cancer models. Administration of E7 is associated with decreased TS expression and tumor burden.

In summary, this is a deep and comprehensive investigation that identifies new molecules that target TS inhibition for the treatment of cancer. The mechanism of action is clearly different than more established TS inhibitors. That said, evidence supporting the proposed mechanism of action is incomplete in the present form.

This manuscript describes a deep and intensive study with an exceptional amount of data and encompasses a range of topics spanning molecular mechanism, drug screening and development, cell biology, and animal model pharmacokinetics. The drug approach is designed to disrupt TS dimers is new and the data generally support the conclusions that are made. However, there are several details surrounding the investigation that should be considered prior to publication.

The primary concern relates to the degree in which the compounds bind to and physically disrupt TS dimers. The FRET data are reported as changes in signal, but the reported signal indicates that an entirely free monomer population is never achieved even at saturating concentrations (40-100 uM) of compound. As such, the proposed mechanism of action would be supported with additional, direct evidence to show binding of the compounds to, and the disruption of, the TS homodimers. Ideally, this could be done with structures of complex-bound TS. However, structural determination is likely to be outside the scope of the current investigation. Another option would be to include SEC profiles showing the disruption of the recombinant TS dimers, along with proper controls (for example the F59A mutant). Similarly, binding of compounds could be quantitatively determined with recombinant TS dimer using ITC. Qualitative binding could be performed with SEC (superdex 75 should resolve the monomer vs dimer). Another qualitative approach to support binding could be a SEC run showing that the protein, monitored at UV280, coelutes with the fluorinated compound (E5-FITC), which could be tracked with fluorescence.

It was not entirely clear how the dual labeled TS dimer was generated for the FRET experiments. The manuscript indicates that these experiments are conducted at 50-500 nM protein concentration. Prior work from the group (Genovese, 2010) reports a dimer equilibrium concentration of ~70 nM and this study reports a similar KD for dimer formation (Figure 5E). Therefore, it is not clear from the kinetic scheme or from the methods how purified green-dimer and red-dimers dissociate to form a red-green dimer used in the FRET studies. Are the two proteins co-expressed and the complex purified by dual affinity using different tags? Is the concentration of individually purified dimers decreased to induce dissociation, then then mixed together to form a heterogeneous mix of red-red, green-green, and red-green dimers concentrated for the FRET studies? Although the prior work (2010) is cited, some additional details for how the bi-labeled dimer was generated would be helpful.

As it is stated in the paper, a fluorescence value of 1 means that the protein is a dimer and a value of 0 correlates to the monomer. It is therefore puzzling that the dimer only partially dissociates when mixed with compounds (with a decrease of no more than ~0.4 units), but does not fully dissociate with increased compound concentration. This observation seems to suggest that the TS dimer is not fully dissociating with addition of compounds and should be addressed in the manuscript.

Similar to the above data, the BS3 crosslinked results reported in Figure 8F appears to be very minimal in disrupting the dimer to monomer. To help define the full degree that the compounds have on disrupting the TS dimer, a similar set of crosslinking experiments with the F59A TS mutant would be helpful.

At the end of pg. 30, it is indicated that E7 treatment leads to a 40-50% reduction in expression of TS mRNA in vivo. However, in cells treated with E7, the mRNA levels of TS remain unchanged (middle pg. 27, Figure 8g, figure supplement 3). This difference in mRNA expression in vitro and in vivo should at least be discussed in the paper.

[Editors' note: further revisions were suggested prior to acceptance, as described below.]

Thank you for resubmitting your work entitled "Disrupters of the thymidylate synthase homodimer accelerate its proteasomal degradation and inhibit cancer growth" for further consideration by *eLife*. Your revised article has been evaluated by Erica Golemis (Senior Editor) and a Reviewing Editor.

The manuscript has been improved but there are some remaining issues that need to be addressed, as outlined below:

In general, the reviewers and handling editors feel that you have done a thorough job addressing many of the concerns raised; however, additional clarification for some of the data presented specifically as outlined and described by reviewer 2 would strengthen the conclusions of the manuscript and make it suitable for publication in *eLife*.

*Reviewer #1 (Recommendations for the authors):*

The authors have addressed the major concerns that were raised in the review.

*Reviewer #2 (Recommendations for the authors):*

The authors have addressed several concerns in the revised manuscript. However, there remain a couple of lingering, but important, issues that should be addressed prior to publication:

It is appreciated that the authors include additional chromatograms analyzing the proposed monomer and dimer peaks (notably in Supporting Figure 3) of the revised manuscript. However, the peak designated as monomer is different in each panel of the figure. In panels A and B the proposed monomer and dimer peaks are barely resolved (~1 rt/min separation), but the same two peaks are drastically separated (~6 rt/min) in panels C-E.

Similarly, the text indicates that dimer and monomer elute at 22 and 24 min, respectively but, since this elution time for each peak varies in each chromatogram, the text should be edited accordingly for accuracy.

Based on the added explanations in the response to reviewers and the cumulative data, the authors should more appropriately refer to the compounds (throughout the text) as dimer destabilizers as opposed to dimer disrupters.

---

## [Author Response]

Essential revisions:The identification of novel chemical inhibitors of thymidylate synthase has the potential to have a significant impact on cancer treatment. The work was viewed as impactful and novel overall. There are a few concerns that need to be addressed prior to this manuscript being suitable for publication including1. Addressing the issue of the utility of these inhibitors compared to 5-FU.

The TS-dimer disrupters here disclosed promise significant advantages over traditional TS-directed anticancer drugs, such as 5-FU. Such advantages stem from a totally different mechanism of action of these compounds. In fact, FdUMP, the active metabolite of 5-FU, inhibits TS by forming a covalent complex containing the co-substrate, mTHF, and the enzyme (Carreras and Santi, 1995). This inhibitory ternary complex is quite stable and leads to prolonged inhibition of the enzyme, depletion of dTMP pools, and thymineless death. However, this engagement of the enzyme dimer by FdUMP induces translational derepression of TS-mRNA and enzyme up-regulation by about 2–4-fold, thus accounting for the occurrence of resistance to TS-targeted compounds such as 5-FU: the acute induction of TS levels following therapy with inhibitors of this enzyme is one of the critical mechanisms of resistance to 5-FU (Chu et al., 1996).

Additionally, enzyme stabilization rather than translational derepression has been indicated as the primary mechanism of TS induction by fluoropyrimidines in human colon cancer cell lines (Kitchens et al., 1999a). TS turnover is carried out by the 26S proteasome with an ubiquitin-independent mechanism (Forsthoefel et al., 2004; Pena et al., 2006). Treatment with 5-FU causes a remarkable slowing of the TS degradation rate in both human colon and ovarian tumor cell lines, most of the protein being involved in the ternary complex formation and thereby stabilized versus degradation (Kitchens et al., 1999b; Marverti et al., 2009).

The TS-dimer disrupters here proposed inhibit TS through a new and different mechanism of action, i.e., by shifting the dimer-monomer equilibrium of the enzyme toward the inactive monomer. Because they do not lead to formation of the inhibitory ternary complex, they are not expected to trigger the mentioned translational derepression and enzyme up-regulation, and are believed to avoid/delay the onset of resistance based on such a mechanism.

Our experiments show clearly that E7 is acting on cells overexpressing hTS where 5FU cannot, thus showing in perspective a higher capacity to counteract the drug resistance status (see effect on pancreatic cancer cells reported in our manuscript). This suggest that E7 can be active after the therapy failure due to 5FU drug resistance.

In principle, even if the dimer disrupters act on the same target, but with a different mechanism, we foresee a potential synergy for their combination with 5FU or, possibly, all those drugs that cross the same metabolic pathways.

In addition, contrary to 5-FU, they induce a decrease in TS half-life due to an enhanced proteasomal degradation, most likely associated with the increase in the fraction of the monomeric protein and its higher vulnerability, with respect to the dimer, to the proteasome. This effect is expected to be crucial in the control of hTS levels in cancer cells and it is maintained in in vivo models of pancreatic cancer in which E7 behaves better than 5FU.

In short, the important biological consequences of the binding of these compounds to TS are the reduction of the intracellular levels of the TS enzyme, and a marked decrease of both in vitro cancer cell growth and in vivo tumor growth in particular observed on pancreatic cancer models. The latter is however expected in other cancer types as well.

5FU is a very effective drug that was conceived in the 60ties, therefore nowadays the expected profile for a new drug shows higher safety and efficacy requirements to be approved. We propose here a new candidate showing in the preclinical phase overall better properties than 5FU, in particular in an orthotopic model of pancreatic cancer, therefore this well predicts for a positive development of this compound as a drug lead.

We have added or strengthened these comments in the Summary and Conclusion of the manuscript improving, we think, its quality. For this, we thank the Editor and the Reviewer.

2. Addressing the degree to which the compounds bind to and physically disrupt TS dimers. This biophysical data is critical to the central conclusions of this manuscript.

See below for specific reviewer comments.

Reviewer #1 (Recommendations for the authors):Thymidylate synthase (TS) is a homodimeric enzyme that is a common anti-cancer target. Currently used TS inhibitors bind to the enzymatic active site which leads to an increase protein expression and subsequent resistance that is observed. The present manuscript through a chemical screening and modeling approach identifies novel inhibitors of TS that bind to the homodimerization interface. These compounds are distinct from the commonly used TS inhibitors. The authors show that their novel compounds do indeed bind, lead to destabilization of protein and decreased cell growth. The work is highly significant and very robustly done.Only a few concerns revolving around the utility of the inhibitors compared to 5-FU are required to address this reviewer's concern:– In the panel of cell lines assessing the efficacy of their novel inhibitors in vitro, comparisons to 5-FU should be shown.

According to the suggestion of this Reviewer, as for Figure 6A, we have added the doseresponse curves of 5FU that we have obtained in all cell lines: TOV112D, C13, 2008, IGROV1. The dose-response curves for TOV112D, C13, 2008, IGROV1, are reported in Figure 6—figure supplement 3. All data are added to Figure 6—figure supplement 3-source data 1. Ovarian cancer cell_growth_inhibition_all_compounds.

In those experiments we observed that cisplatin-resistant cells, such as C13* and A2780/CP, are actually also cross-resistant to 5-FU. These cell lines show a moderate or no drug resistance or cross-resistant to these novel TS-dimer-disrupting inhibitors. In fact, the same resistant cell lines are equally or even more responsive to E5 and E6 than the sensitive parental counterparts.

– The major concern is that the effects of E7 are not dramatically better than 5-FU, and in most cases are the same or even worse. Please assess if E7 can inhibit viability in cells that are resistant to 5-FU in vitro and in vivo.

About viability studies in vitro, as shown in Figure 6 and in Figure 6—figure supplement 3 and in the table of IC_50_ values in Supplementary File 7, E7 was actually able to inhibit cell viability of C13* and A2780/CP cell lines, the cisplatin-resistant counterparts of the parental sensitive 2008 and A2780 cells, respectively. These resistant cell lines, selected for resistance to cisplatin, are also cross-resistant to 5-FU.

In addition, as shown in Figure 6—figure supplement 4 and in the table of IC_50_ values in Supplementary File 7, E7 was also able to inhibit cell viability of three ColoRectalCancer (CRC) cells, considered resistant to 5-FU among a wide panel of CRC cell lines, divided into sensitive, intermediate and resistant to 5-FU (Bracht K, et al., British Journal of Cancer 103: 340-346).

Noteworthy, E7 was almost as equally active against both PDAC2 and PDAC5 cells, which express low and intermediate TS levels (Firuzi et al., Cancers, 2019), and were sensitive to 5-FU (Figure 6 – supplement 1, Figure 6 – supplement 1-source data 1). The LPc167 primary culture, which had high expression levels of TS was resistant to 5-FU but still sensitive to E7, as reported in the Supplementary file 6A and Figure 6—figure supplement 2, Figure 6—figure supplement 2-Source data 1.

– Data on cell death and proliferation are needed for the in vivo tumor studies.

We thank the reviewer for the very relevant suggestion. We have performed additional analyses to clarify the type of cell death as well as the proliferation. Therefore, we have added the following information:

In the methods: “Sections of 3 µm pancreatic slices were cut from paraffin-embedded specimens. IHC was performed with the anti-hTS antibody described above (dilution 1:100), as well as with the anti-Cleaved Caspase-3 (Asp175) Antibody #9661 (dilution 1:100, Cell Signaling, Danvers, MA, USA). and the anti-Ki67 monoclonal antibody (dilution 1:100, clone MIB1; DBA, Milan, Italy), as described previously (Bianco C et al., Clinical Cancer research 2006; Su JJ, Xu K et al. World J Gastrointest Oncol. 2018).”

In the results: “Additional analyses showed that E7 reduced the proliferation of the cancer cells, as detected by the reduced number of Ki67-positive cells, while increasing apoptotic cell death, as assessed by the staining of cleaved caspase-3 (*Figure 8—figure supplement 3*). Notably, these effects were markedly stronger in tumors treated with E7 compared to tumors treated with 5-FU.”

– The discussion needs to focus on the novel finds and clinical impact, currently it is highly repetitive to the result section. Along these lines it would be good to discuss mechanisms of 5-FU resistance and under what scenarios would the dimerization inhibitors be used.

We thank the Reviewer for the suggestion. Accordingly, we have modified the Summary and Conclusions, discussed the mechanisms of 5-FU resistance and emphasized more the usefulness of these dimerization inhibitors overcoming and/or preventing the onset of resistance. New sentences have been added in main text, section Summary and Conclusions, on page 35.

Reviewer #2 (Recommendations for the authors):Conventional thymidylate synthase (TS) inhibitor drugs bind to the active site or to the nearby folate cofactor site. Either mechanism leads to elevated TS expression, along with drug resistance, presumably by stabilizing the inactive TS dimer. The authors come up with an alternative strategy intended to disrupt TS dimerization, which would lead to accelerated degradation of the less stable monomeric protein.For the proposed strategy, the study identifies a putative drug binding pocket at the TS protein-protein interface that is predicted to disrupt the homodimer. The authors then introduce a cysteine residue in this pocket which serves as the target of a cysteine tethering approach to implement fragment-based drug design. After a series of fragment-based and virtual docking screens, a series of molecules and derivatives are identified for testing. While the cysteine tethering approach is a good first step to identify putative small molecules, it is not certain that the compounds are binding preferentially to a drug "pocket" or simply forming a covalent bond with the introduced cysteine. This is an important consideration when examining the mechanism of action for subsequent derivatives in the context of the native enzyme.

Thank you for noticing that. We would like to point out that in our workflow and process the covalent bond was only a strategic approach adopted in the first stage of the project to capture the fragment with higher affinity for the protein interface. Then, after fragment identification, a medicinal chemistry approach has been developed to generate non-covalent ligands.

We specified this with a sentence in the text under the section: “From low-affinity compounds to TS homodimer disrupters.”

The analogs were experimentally tested against TS activity and for an ability to disrupt dimerization. Effects on TS dimer disruption was assayed using a FRET approach and TS expressed with either a red- or green-emitting tag. It is concluded that some of these molecules disrupt TS dimerization. However, a key concern regarding the proposed mechanism of action, is that data supporting drug-protein interactions and TS dimer dissociation are somewhat indirect. Since a primary focus of the investigation centers around the proposed mechanism of action of the new compounds, the study would be enhanced with additional structural and/or biophysical data to support binding of the individual molecules to TS directly, and the subsequent dissociation of TS dimers.The comprehensive investigation then examines the stability of TS in treated cells and with proteasome inhibitors. The data reveal that administration of compounds correlates with enhanced degradation of TS and the authors hypothesize that this is due to the TS monomer, disrupted by E7 administration, is less stable than the functional dimer. A comparison of treatments of conventional TS inhibitors, such as 5-FU and Pemetrexed, support a different mechanism of action as TS levels are elevated following administration of these established compounds. Finally, the investigation identifies the effects of the lead compound, E7, in mouse models of pancreatic and ovarian cancer models. Administration of E7 is associated with decreased TS expression and tumor burden.In summary, this is a deep and comprehensive investigation that identifies new molecules that target TS inhibition for the treatment of cancer. The mechanism of action is clearly different than more established TS inhibitors. That said, evidence supporting the proposed mechanism of action is incomplete in the present form.This manuscript describes a deep and intensive study with an exceptional amount of data and encompasses a range of topics spanning molecular mechanism, drug screening and development, cell biology, and animal model pharmacokinetics. The drug approach is designed to disrupt TS dimers is new and the data generally support the conclusions that are made. However, there are several details surrounding the investigation that should be considered prior to publication.The primary concern relates to the degree in which the compounds bind to and physically disrupt TS dimers. The FRET data are reported as changes in signal, but the reported signal indicates that an entirely free monomer population is never achieved even at saturating concentrations (40-100 uM) of compound. As such, the proposed mechanism of action would be supported with additional, direct evidence to show binding of the compounds to, and the disruption of, the TS homodimers. Ideally, this could be done with structures of complex-bound TS. However, structural determination is likely to be outside the scope of the current investigation. Another option would be to include SEC profiles showing the disruption of the recombinant TS dimers, along with proper controls (for example the F59A mutant).

We have performed the experiments using an anion exchange chromatography column and demonstrated that the dimer disrupters bind to both dimer and monomer. In those experiments, accounted for in Figure 4, we have shown that the two dimer disrupters tested co-eluted with both the protein monomers and dimers (see also our following answer).

Similarly, binding of compounds could be quantitatively determined with recombinant TS dimer using ITC. Qualitative binding could be performed with SEC (superdex 75 should resolve the monomer vs dimer). Another qualitative approach to support binding could be a SEC run showing that the protein, monitored at UV280, coelutes with the fluorinated compound (E5-FITC), which could be tracked with fluorescence.

We thank the Reviewer to stimulate an interesting discussion. In spite of the centennial history of molecular spectroscopy, spectroscopic evidence is sometimes considered ‘indirect’, and more ‘direct’ ways to ‘see’ protein–ligand complexes are searched for. A classical and highly convincing answer comes from crystallography. Unfortunately, despite our many efforts, we could not obtain crystals of complexes of hTS with the allosteric inhibitors discussed here.

A frequently adopted strategy makes use of chromatography. In general, because of dilution throughout columns, mass-law action acts against detection of surviving protein/ligand complexes, and tailored experimental solutions must be found in order to avoid extensive dissociation of such complexes (for some literature on this problem, see K.F. Blom, B. S. Larsen, C. N. McEwen, Determining Affinity-Selected Ligands and Estimating Binding Affinities by Online Size Exclusion Chromatography/Liquid Chromatography-Mass Spectrometry, J. Comb. Chem. 1999, 1, 82-90; G. Deng, G. Sanyal, Applications of mass spectrometry in early stages of target-based drug discovery, J. Pharm. Biomed. Anal. 40 (2006) 528–538). In our case the problem was further complicated by the concurrence of the dimer/monomer equilibrium together with the ligand binding equilibria by the protein dimers and monomers. However, after several attempts failed due to complete dimer/monomer re-equilibration in the elution throughout the chromatographic columns, we succeeded in proving the ability of compounds C3 and E5 to bind hTS and favor disruption of its dimer by employing anion-exchange chromatography. We have added a synthetic overview of these new experiments in the final portion of paragraph ‘From low-affinity compounds to TS homodimer disrupters’ and have shown the most significant of these results in some new figures, namely, Figure 3E in the main text and the associated figures Figure 3—figure supplement 1 and Figure 3—figure supplement 2.

It was not entirely clear how the dual labeled TS dimer was generated for the FRET experiments. The manuscript indicates that these experiments are conducted at 50-500 nM protein concentration. Prior work from the group (Genovese, 2010) reports a dimer equilibrium concentration of ~70 nM and this study reports a similar KD for dimer formation (Figure 5E). Therefore, it is not clear from the kinetic scheme or from the methods how purified green-dimer and red-dimers dissociate to form a red-green dimer used in the FRET studies. Are the two proteins co-expressed and the complex purified by dual affinity using different tags? Is the concentration of individually purified dimers decreased to induce dissociation, then then mixed together to form a heterogeneous mix of red-red, green-green, and red-green dimers concentrated for the FRET studies? Although the prior work (2010) is cited, some additional details for how the bi-labeled dimer was generated would be helpful.

Neither one. After protecting all cysteines except 43 and 43’, we tagged the protein dimers (the protein concentrations, 30-80 microM, were much larger than the dimer-monomer equilibrium constant, 80 nM) with the maleimides of the two probes together. Hence, we obtained protein dimers carrying two probes, one per monomer. Properly working on fluorescein- (F, green) and tetramethylrhodammine- (T, red) maleimide concentrations, we succeeded in establishing conditions in which the same amounts of each probe were bound to TS: 1 F: 1T : 1 protein monomer. As a result, 25 % of protein dimers were labeled with two Fs, 25 % with two Ts and 50% with one F and one T. Because the F-T Förster’s critical distance is twice as large as the C43-C43’ distance, the Fto-T FRET efficiency is near unity in such hetero-tagged TS dimers. This assumption and the above population of fluorophores were the bases for developing an expression that relates the fractions of protein dimers with the measured intensities of the F and T emission bands. Because the analysis is rather cumbersome and has been fully given in the quoted reference, we avoided reporting it in this already large and complex manuscript.

As it is stated in the paper, a fluorescence value of 1 means that the protein is a dimer and a value of 0 correlates to the monomer. It is therefore puzzling that the dimer only partially dissociates when mixed with compounds (with a decrease of no more than ~0.4 units), but does not fully dissociate with increased compound concentration. This observation seems to suggest that the TS dimer is not fully dissociating with addition of compounds and should be addressed in the manuscript.

Even at fully saturating concentrations of the ligands, i.e., at concentrations such that all dimers and all monomers of hTS are bound to an inhibitor molecule, inhibitor-bound dimers and monomers still equilibrate. In no case encountered so far is this equilibrium fully shifted towards the monomers. In all favorable cases (many tested compounds caused no effect), relative to the free protein dimer/monomer equilibrium, binding of these ligands favored the protein monomers. The extent of such shift (measurable either in terms of equilibrium constant ratio or of DDG°), measures the ability of each compound to destabilize the dimeric assembly of the protein. In other words, the effect of a given DDI might be seen as resulting from two features: the ability to bind the protein (dimers and monomers) and the ability to destabilize the dimeric assembly.

Similar to the above data, the BS3 crosslinked results reported in Figure 8F appears to be very minimal in disrupting the dimer to monomer. To help define the full degree that the compounds have on disrupting the TS dimer, a similar set of crosslinking experiments with the F59A TS mutant would be helpful.

We thank the reviewer for the comments and suggestions. In Figure 7F the reduction of the dimer/monomer ratio after E7 treatment is 40% after 3 hours at 20 uM. This reduction is the result of a snapshot obtained after 3 hours of treatment, but we should take into account that the E7 efficacy, in living cells, continues for up to 36 hours, as demonstrated by the immunoblot in Figure 7—figure supplement 1c,d-Source data 1. (Effect of dimer disrupters on hTS protein levels in cells_WB).

The immunoblot in the Figure shows a drastic and time dependent reduction of the TS protein levels. The TS protein reduction observed in living cells in steady state conditions after E7 treatment reflects the continuous and dynamic shift of the TS equilibrium from the dimeric towards the monomeric form. The monomeric form of TS, as described in the main text, turns out to be very unstable and more degradable by the proteasome.

F59A TS mutant is essentially monomeric at nM, low μm concentrations. Following the reviewer’s suggestion, we performed a set of crosslinking experiments with the F59A TS mutant. Based on the fast degradation of the F59A TS mutant (see Figure 7G) we were forced to use a proteasome inhibitor MG132 for 6 hours at 20 μm prior to harvesting and lysing the cells for crosslinking (BS3). We ectopically expressed TS-Myc-DDK-WT and TS-Myc-DDK-F59A in HEK293 cells. After 24 hours, MG132 were added, and after 3 hours, the cells were treated with E7 for 3 hours. Next the cells were lysed and crosslinked (BS3). The whole cell lysate was separated by SDS-Page gel in reducing conditions, and western blot performed by using the anti-Flag antibody (Author response image 1).

The immunoblot shown in Author response image 2 highlights a clear reduction of the dimer/monomer ratio of TS-WT after 3 hours of E7 treatment with respect to the control condition. The result is in agreement with the effect we have reported in Figure 7F. Unfortunately, as we expected, the F59A TS mutant did not show any detectable dimeric form, and therefore it was not possible to determine a dimer/monomer ratio and any change of it associated with cell treatment with E7.

**Author response image 1. sa2fig1:** (A) Effects on exogenous hTS dimer/monomer equilibrium in HEK293 cells after a 3 hour exposure to E7 (20 μM), in the presence of MG132 (6 hour). Next the cells have been lysed and crosslinked (BS3), and western blot performed by using the anti-Flag antibody. (B) Quantification of the immunoblot shown in (A) of exogenous hTS dimer/monomer equilibrium (anti-FLAG stain) in HEK293 cells transfected with TS-Myc-DDK (24 hour) or with the F59A mutant (24 hour) tagged vector.

**Author response image 2. sa2fig2:** Original blots (A) and densitometry quantification (B). (A) Effects on exogenous hTS dimer/monomer equilibrium in HEK293 cells after a 3 hours exposure to E7 (20 µM), in the presence of MG132 (6 hour). Next the cells have been lysed and crosslinked (BS3), and western blot performed by using the anti-Flag antibody. (B) Quantification of the immunoblot shown in (A) of exogenous hTS dimer/monomer equilibrium (anti-FLAG stain) in HEK293 cells transfected with TSMyc-DDK (24 hour) or with the F59A mutant (24 hour) tagged vector.

At the end of pg. 30, it is indicated that E7 treatment leads to a 40-50% reduction in expression of TS mRNA in vivo. However, in cells treated with E7, the mRNA levels of TS remain unchanged (middle pg. 27, Figure 8g, figure supplement 3). This difference in mRNA expression in vitro and in vivo should at least be discussed in the paper.

Thank you for noticing this. We clarify here that in the main text of the previous version of the manuscript, in the middle of page 27, the TS mRNA levels described refer to an experiment different from that described at the end of page 30 (Figure 8F in the present version). Figure 8F demonstrates a reduction of endogenous TS mRNA levels after E7 treatment in vivo. Figure 7—figure supplement 3 (present version), instead, shows the quantification of either WT or F59A TS mRNA described in the middle of page 27 corresponding to Figure 7G in the main text. The experiment was performed to demonstrate that the protein levels of mutant TS-F59A (Figure 7G) are related to the instability of the protein itself and not to the reduction of its mRNA (Figure 7—figure supplement 3 present version).

Furthermore, we have performed the quantification of TS mRNA in cells (Figure 8—figure supplement 2 and Figure 8—figure supplement 8-Source data 1) in different cancer cell lines after treatment with E5, E7 and 5FU. The experiments show its reduction following treatment with E7, in agreement with the results obtained in vivo (Figure 8F). We added a comment in the section “ in vivo anticancer activity” section.

[Editors' note: further revisions were suggested prior to acceptance, as described below.]

The manuscript has been improved but there are some remaining issues that need to be addressed, as outlined below:In general, the reviewers and handling editors feel that you have done a thorough job addressing many of the concerns raised; however, additional clarification for some of the data presented specifically as outlined and described by reviewer 2 would strengthen the conclusions of the manuscript and make it suitable for publication in eLife.Reviewer #2 (Recommendations for the authors):The authors have addressed several concerns in the revised manuscript. However, there remain a couple of lingering, but important, issues that should be addressed prior to publication:It is appreciated that the authors include additional chromatograms analyzing the proposed monomer and dimer peaks (notably in Supporting Figure 3) of the revised manuscript. However, the peak designated as monomer is different in each panel of the figure. In panels A and B the proposed monomer and dimer peaks are barely resolved (~1 rt/min separation), but the same two peaks are drastically separated (~6 rt/min) in panels C-E.Similarly, the text indicates that dimer and monomer elute at 22 and 24 min, respectively but, since this elution time for each peak varies in each chromatogram, the text should be edited accordingly for accuracy.

In panels A and B of Figure 3 supplement 1 (the mentioned Supporting Figure 3 corresponds to Figure 3 supplement 1, as it is written in the text, just to avoid misunderstanding), corresponding to AE chromatographic runs with, respectively, hTS alone and in the presence of two active-site ligands, dUMP and RTX, the two bands are separated by 1.5-2 minutes. In panels C-E, reporting chromatograms obtained in the presence of increasing concentrations of C3, the two bands are consistently separated by ca. 5 minutes. We attribute this increased separation to a larger delaying effect of bound C3 on the protein monomer elution than on the dimer elution. Indeed, we have performed another AE chromatographic run, not reported for obvious reasons of brevity, in the same technical chromatographic conditions, with the same volume of hTS at the same concentration and with 40 µM 2-amino benzothiazole, the chemical core of the dimer destabilizers. This compound caused neither inhibition of the enzymatic activity of hTs nor a decrease in FRET efficiency. Consistently, the separation between the dimer and the monomer bands was, again, ca. 2 minutes.

Our mentioning the elution times for the two bands as ca. 22 and 24 minutes refers specifically to panels A and B in the figure. In the runs shown in panels C-E the retention time of the dimer band is less reproducible. We attribute our limited control on the zero-elution time to the fact that we perform the injection into the column manually. We have corrected the text at p. 16 (paragraph ‘Additional evidence of the ability of compounds…) by replacing the retention times with the separation between the two bands, that is, quite consistently, around 2 minutes in the absence of compound C3 and 5-6 minutes in its presence.

Based on the added explanations in the response to reviewers and the cumulative data, the authors should more appropriately refer to the compounds (throughout the text) as dimer destabilizers as opposed to dimer disrupters.

Once more, we thank this Reviewer for his/her thorough work. We have adhered to the suggestion and have modified the manuscript accordingly.